# Learning Causal Abstractions of Linear Structural Causal Models

**Riccardo Massidda**[1]          **Sara Magliacane**[2]          **Davide Bacciu**[1]

[1]Department of Computer Science, Università di Pisa, Pisa, IT
[2]Informatics Institute, University of Amsterdam, Amsterdam, NL

## Abstract

The need for modelling causal knowledge at different levels of granularity arises in several settings. Causal Abstraction provides a framework for formalizing this problem by relating two Structural Causal Models at different levels of detail. Despite increasing interest in applying causal abstraction, e.g. in the interpretability of large machine learning models, the graphical and parametrical conditions under which a causal model can abstract another are not known. Furthermore, learning causal abstractions from data is still an open problem. In this work, we tackle both issues for linear causal models with linear abstraction functions. First, we characterize how the low-level coefficients and the abstraction function determine the high-level coefficients and how the high-level model constrains the causal ordering of low-level variables. Then, we apply our theoretical results to learn high-level and low-level causal models and their abstraction function from observational data. In particular, we introduce Abs-LiNGAM, a method that leverages the constraints induced by the learned high-level model and the abstraction function to speedup the recovery of the larger low-level model, under the assumption of non-Gaussian noise terms. In simulated settings, we show the effectiveness of learning causal abstractions from data and the potential of our method in improving scalability of causal discovery.

## 1 INTRODUCTION

Causal Abstraction formalizes the property of distinct causal models to describe the same phenomenon with different levels of detail [Beckers and Halpern, 2019]. Despite having different variables and mechanisms, whenever two Structural Causal Models (SCMs) are in an abstraction relation, there must always exist at least one implementation on the low-level *concrete* model of any property of the high-level *abstract* one — such as values, interventions, mechanisms, and endogenous or exogenous distributions.

Abstract causal models allow the interpretation of causal models with large number of variables, such as in climate phenomena [Chalupka et al., 2016] or brain activation patterns [Dubois et al., 2020]. Causal Abstraction has also found wide interest in explainable AI to align machine representations with human-interpretable concepts in feedforward neural networks [Geiger et al., 2021, 2023], concept-based neural networks [Marconato et al., 2023], and Large Language Models [Wu et al., 2024, Geiger et al., 2024].

Previous works on the definition of Causal Abstraction do not focus on the graphical and parametrical conditions for two models to be in an abstraction relation. Furthermore, the problem of learning abstractions from data, when the high-level model is not known, is still open. In this context, Zennaro [2022] and Geiger et al. [2023] propose methods to learn an abstraction function assuming to know both the low-level and the abstract model, while Chalupka et al. [2016], Kekić et al. [2023] and Felekis et al. [2024] assume to have at least the graphical structure of the abstract model.

In this paper, we tackle these issues by focusing on the scenario where two linear SCMs are abstracted by a linear transformation, as shown in Figure 1. In particular, we study necessary and sufficient conditions for abstraction in terms of the edges and the coefficients of the models. We then propose Abs-LiNGAM, a strategy to learn from data the abstract model, the concrete model, and their abstraction function under the further assumption of non-Gaussian exogenous noise. We summarize our contributions as follows:

1. We first prove that abstract edges necessarily require edges in the low-level model to connect relevant variables, i.e., variables on which the abstraction function directly depends (Theorem 1). Then, we show that the abstraction necessarily arranges concrete variables in

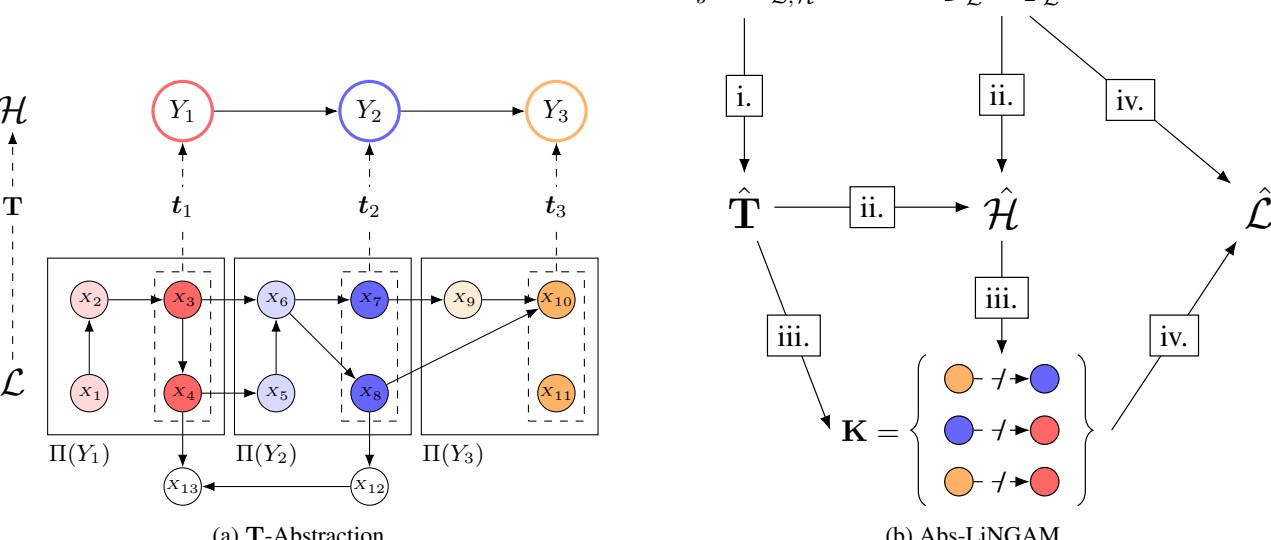

(a) $\mathbf{T}$-Abstraction

(b) Abs-LiNGAM

Figure 1: An overview of our contributions: (a.) A linear SCM $\mathcal{H}$, representing the *abstract* causal model, is a $\mathbf{T}$-abstraction of a linear SCM $\mathcal{L}$, representing the *concrete* causal model, whenever the linear transformation $\mathbf{T}$ from concrete to abstract variables is interventionally consistent, i.e., whenever it relates both values and interventions on the abstract model and the concrete model. We prove that, for each abstract variable $Y$, the transformation $\mathbf{T}$ induces a block $\Pi(Y)$ of concrete causal variables that necessarily follows the causal ordering of the abstract model and whose parameters are constrained by the abstract coefficients. For each block, the abstraction function depends on a possibly smaller subset of *relevant* variables, which we portray as dashed. (b.) We propose Abs-LiNGAM, a method to speedup the causal discovery of the concrete model $\mathcal{L}$ given an additional dataset $\mathcal{D}_J$ sampled from the joint distribution of the abstract and the concrete model. In order, Abs-LiNGAM (i.) reconstructs the transformation $\mathbf{T}$, (ii.) fits the abstract model by abstracting the concrete dataset $\mathcal{D}_{\mathcal{L}}$, (iii.) infers a set of constraints $\mathbf{K}$ on which paths cannot exist in the concrete graph, and finally (iv.) discovers the concrete model in a search space reduced by the constraints.

adjacent and disjoint blocks that must follow the abstract causal ordering (Theorem 2).

2. We then prove a necessary and sufficient condition for causal abstraction that relates the coefficients of the linear models and the abstraction function (Theorem 3). In this way, we can characterize the set of all concrete models that are abstracted by a given abstract SCM and design a complete and correct algorithm to sample any model from this set (Algorithm 1).

3. We introduce Abs-LiNGAM, a method to speedup the causal discovery of large linear non-Gaussian models given an additional and small dataset sampled from the observational joint distribution of the model and one of its abstractions. Abs-LiNGAM recovers the abstraction function, learns the abstract model using low-level data, and finally constrains the recovery of the concrete model by ensuring that the necessary conditions we introduced are satisfied (Algorithm 2).

4. As we report in Section 5, experiments in simulated settings show that Abs-LiNGAM substantially reduces the search space, and thus the execution time, compared to directly solving the problem on the low-level dataset with DirectLiNGAM [Shimizu et al., 2011].

We also publicly release online the code of Abs-LiNGAM and the experimental settings[1].

## 2 BACKGROUND

Given a set of variables $\boldsymbol{X}$, we denote the domain of each variable $X \in \boldsymbol{X}$ as $\mathcal{D}(X)$ and of any subset $\boldsymbol{V} \subseteq \boldsymbol{X}$ as $\mathcal{D}(\boldsymbol{V})$. We define a Structural Causal Model [SCM; Pearl, 2009] as a tuple $\mathcal{M} = \big(\boldsymbol{X}, \boldsymbol{E}, \{f_X\}_{X \in \boldsymbol{X}}, \mathbb{P}_{\boldsymbol{E}}\big)$, where

1. $\boldsymbol{X}$ is a set containing $d$ distinct *endogenous* variables,
2. $\boldsymbol{E}$ is a set containing $d$ distinct *exogenous* variables,
3. $f_X \colon \mathcal{D}(\mathrm{Pa}(X) \cup \{E_X\}) \to \mathcal{D}(X)$ is a *causal mechanism*, i.e. a function that determines the value of the variable $X \in \boldsymbol{X}$ given its parents $\mathrm{Pa}(X)$ and the exogenous noise term $E_X \in \boldsymbol{E}$,
4. $\mathbb{P}_{\boldsymbol{E}}$ is the joint distribution over $\boldsymbol{E}$.

We assume that parental relations define a directed acyclic graph $\mathcal{G}_{\mathcal{M}}$ and, consequently, that the reduced form of the model always has a unique solution [Bongers et al., 2021]. By slightly abusing the notation, we denote as $\mathcal{M}$ both

---

[1] https://github.com/rmassidda/causabs

the SCM and its reduced form $\mathcal{M} \colon \mathcal{D}(\boldsymbol{E}) \to \mathcal{D}(\boldsymbol{X})$ mapping exogenous to endogenous values. A hard intervention is an assignment $i = (\boldsymbol{V} \leftarrow \boldsymbol{v})$ on a subset of variables $\boldsymbol{V} \subseteq \boldsymbol{X}$ that replaces each mechanism of the variables $\boldsymbol{V}$ with a constant value $\boldsymbol{v} \in \mathcal{D}(\boldsymbol{V})$. We denote as $\boldsymbol{I}^*$ the set of all hard interventions on an SCM, containing all possible assignments to any subset of endogenous variables, also including the empty intervention. Formally, an intervention $i$ results in a different SCM $\mathcal{M}^i$ defined by the tuple $\left(\boldsymbol{X}, \boldsymbol{E}, \{f_X^i\}_{X \in \boldsymbol{X}}, \mathbb{P}_{\boldsymbol{E}}\right)$, where $f_X^i = f_X$ if $X \notin \boldsymbol{V}$ and $f_X^i(\cdot) = v_X$ otherwise. We then define the restriction of an intervened causal model as the set of values that the model can take after the intervention, which we denote as

$$\mathrm{Rst}\left(\mathcal{M}^{\boldsymbol{V} \leftarrow \boldsymbol{v}}\right) = \{\boldsymbol{x} \in \mathcal{D}(\boldsymbol{X}) \mid \boldsymbol{x}_{\boldsymbol{V}} = \boldsymbol{v}\}, \quad (1)$$

where $\mathrm{Rst}\left(\mathcal{M}\right) = \mathcal{D}(\boldsymbol{X})$ for a non-intervened SCM.

We assume faithfulness and causal sufficiency, i.e., the absence of hidden confounding and selection bias [Spirtes et al., 2000]. In particular, faithfulness implies the absence of canceling paths across variables, while causal sufficiency implies mutual independence of exogenous terms, as in $E_1 \perp\!\!\!\perp E_2$ for any $E_1, E_2 \in \boldsymbol{E}$.

A linear SCM $\mathcal{M} = (\boldsymbol{X}, \boldsymbol{E}, \mathbf{W}, \mathbb{P}_{\boldsymbol{E}})$, also known as a linear Additive Noise Model (ANM) [Peters et al., 2017], is an SCM whose structural equations are linear and represented by an upper-triangular adjacency matrix $\mathbf{W} \in \mathbb{R}^{d \times d}$, as in

$$\mathbf{X} = \mathbf{W}^\top \mathbf{X} + \mathbf{E}. \quad (2)$$

We can compute the reduced form of the model in closed form as,

$$\mathcal{M}(\boldsymbol{e}) = \mathbf{F}^\top \boldsymbol{e}, \quad (3)$$

where $\mathbf{F} = (\mathbf{I} - \mathbf{W})^{-1}$ is a $d \times d$ linear transformation.

Causal Abstraction theory relates variables across different SCMs to determine whether they represent in a *consistent* way the same system at different levels of detail [Beckers and Halpern, 2019]. Overall, we refer to *concrete*, or low-level, causal models as $\mathcal{L} = (\boldsymbol{X}, \boldsymbol{E}, \boldsymbol{f}, \mathbb{P}_{\boldsymbol{E}})$ and to *abstract*, or high-level, causal models as $\mathcal{H} = (\boldsymbol{Y}, \boldsymbol{U}, \boldsymbol{g}, \mathbb{P}_{\boldsymbol{U}})$, where $|\boldsymbol{X}| \geq |\boldsymbol{Y}|$. An abstraction requires to consider two subsets of allowed interventions $\boldsymbol{I} \subseteq \boldsymbol{I}^*$ and $\boldsymbol{J} \subseteq \boldsymbol{J}^*$ respectively on the concrete and the abstract model.

In this work, we focus on *strong* abstractions, where any concrete or abstract intervention is allowed, i.e., $\boldsymbol{I} = \boldsymbol{I}^*$ and $\boldsymbol{J} = \boldsymbol{J}^*$. Then, given a surjective function $\tau \colon \mathcal{D}(\boldsymbol{X}) \to \mathcal{D}(\boldsymbol{Y})$, $\mathcal{H}$ is a $\tau$-abstraction of $\mathcal{L}$ if and only if there exists a surjective function $\gamma \colon \mathcal{D}(\boldsymbol{E}) \to \mathcal{D}(\boldsymbol{U})$ on the exogenous variables such that, for any low-level intervention $i \in \boldsymbol{I}$ and any exogenous configuration $\boldsymbol{e} \in \mathcal{D}(\boldsymbol{E})$, it holds

$$\tau(\mathcal{L}^i(\boldsymbol{e})) = \mathcal{H}^{\omega(i)}(\gamma(\boldsymbol{e})), \quad (4)$$

where the intervention map $\omega \colon \boldsymbol{I} \to \boldsymbol{J}$ is uniquely induced by the value abstraction function $\tau$ [Massidda et al., 2023]. Formally, $\omega(i) = j$ if and only if $\mathrm{Rst}\left(\mathcal{H}^j\right) = \tau(\mathrm{Rst}\left(\mathcal{L}^i\right))$ and $j \in \boldsymbol{J}$, otherwise $\omega(i)$ is undefined. We refer to Equation (4) as the *interventional consistency* property and to the function $\gamma$ as the *exogenous abstraction function*. Similarly, we refer to the equation

$$\tau(\mathcal{L}(\boldsymbol{e})) = \mathcal{H}(\gamma(\boldsymbol{e})) \quad (5)$$

on non-intervened models as *observational consistency*. Since the empty intervention is necessarily a fixed point of the intervention map [Massidda et al., 2023], interventional consistency implies observational consistency.

Finally, as for the causal mechanisms, we assume that the abstraction function does not yield cancelling paths towards abstract variables. Formally, the composition of the abstraction function $\tau$, with the concrete model $\mathcal{L}$ must not cancel the effect of concrete variables $\boldsymbol{X}$ on abstract variables $\boldsymbol{Y}$.

# 3 THEORY OF LINEAR CAUSAL ABSTRACTION

In this section, we study graphical and structural properties of linear causal models in a linear abstraction relation. First, we prove that the set of concrete *relevant* variables on which each abstract variable depends are necessarily disjoint (Section 3.1). Further, we prove necessary and sufficient conditions on the existence of an abstract edge in terms of the directed paths between relevant variables in the concrete graph (Section 3.2). We then show that the abstraction function constrains the causal ordering of *concrete blocks* composed of both relevant and non-relevant variables (Section 3.3). Finally, given the notion of concrete block, we provide an equivalent formulation of abstraction based on the model parameters, which also characterizes the set of possible concretizations of an abstract model (Section 3.4).

## 3.1 LINEAR CAUSAL ABSTRACTION

We focus on the scenario where the value abstraction function $\tau$ between an abstract and a concrete causal model is a linear transformation represented by a matrix $\mathbf{T}$. We then define such linear relation between SCMs as $\mathbf{T}$-abstraction.

**Definition 1** (T-Abstraction)**.** *Let $\mathcal{H}$ be a strong $\tau$-abstraction of $\mathcal{L}$, where $\mathcal{H}$ and $\mathcal{L}$ are two SCMs respectively on variables $\boldsymbol{Y}$ and $\boldsymbol{X}$. Then, $\mathcal{H}$ is a $\mathbf{T}$-abstraction of $\mathcal{L}$ whenever there exists a linear transformation $\mathbf{T} \in \mathbb{R}^{d \times b}$, where $d = |\boldsymbol{X}|$ and $b = |\boldsymbol{Y}|$, such that $\tau(\boldsymbol{x}) = \mathbf{T}^\top \boldsymbol{x}$.*

One of the common aspects of causal abstraction consists of reducing the dimensionality of a causal model by selecting *relevant* and discarding *irrelevant* variables [Zennaro, 2022]. Therefore, for each abstract variable $Y$, we define its set of

*relevant* variables $\Pi_R(Y)$ as the set of concrete variables on which it directly depends according to the **T**-abstraction. Overall, we refer to the set of relevant variables in the concrete model as the union of the relevant variables of each abstract variable and to all the remaining as *irrelevant*.

**Definition 2** (Relevant Variables). *Let $\mathcal{H}$ be a **T**-abstraction of $\mathcal{L}$, where $\mathcal{H}$ and $\mathcal{L}$ are two SCMs respectively on variables $\boldsymbol{Y}$ and $\boldsymbol{X}$. We define the set of relevant concrete variables of an abstract variable $Y_j \in \boldsymbol{Y}$ as the subset*

$$\Pi_R(Y_j) = \{X_i \in \boldsymbol{X} \mid t_{ij} \neq 0\}, \tag{6}$$

*where $t_{ij}$ is the $i$-th element on the $j$-th column of **T**. Moreover, we define the set of relevant variables $\Pi_R(\boldsymbol{Y})$ of the abstract model $\mathcal{H}$ as the union of all relevant sets for each variable $Y \in \boldsymbol{Y}$. Formally,*

$$\Pi_R(\boldsymbol{Y}) = \bigcup_{Y \in \boldsymbol{Y}} \Pi_R(Y). \tag{7}$$

*We define as irrelevant the remaining variables in the concrete model $\mathcal{L}$, i.e. $\boldsymbol{X} \setminus \Pi_R(\boldsymbol{Y})$.*

To guarantee surjectivity, the transformation **T** must have full column-rank and, consequently, the set of relevant variables for each abstract variable must be non empty. Since we require consistency to hold on the set of all possible abstract hard interventions, we can easily prove that this implies that the sets of relevant variables must be mutually disjoint.

**Lemma 1** (Disjoint Relevant). *Let $\mathcal{H}$ be a **T**-abstraction of $\mathcal{L}$, where $\mathcal{H}$ and $\mathcal{L}$ are two linear SCMs respectively on variables $\boldsymbol{Y}$ and $\boldsymbol{X}$. Then, for any pair of distinct abstract variables $Y_1, Y_2 \in \boldsymbol{Y}$, it holds that $\Pi_R(Y_1) \cap \Pi_R(Y_2) = \emptyset$, where $\Pi_R(Y_1) \neq \emptyset$ and $\Pi_R(Y_2) \neq \emptyset$.*

*Proof.* We report the proof in Appendix B.1. $\square$

Beckers and Halpern [2019] define *constructive* abstraction as a special case where abstract variables depend on disjoint sets of low-level variables and conjectures that, under further assumptions, strong abstraction might entail constructive abstraction. Notably, with our linearity assumptions, such conjecture immediately derives from Lemma 1.

**Corollary 1** (Constructive Abstraction). *Let $\mathcal{H}$ be a strong $\tau$-abstraction of $\mathcal{L}$ where $\mathcal{H}$ and $\mathcal{L}$ are linear SCMs and $\tau$ is a linear transformation. Then, $\mathcal{H}$ is a constructive $\tau$-abstraction of $\mathcal{L}$.*

*Proof.* We report the proof in Appendix B.2. $\square$

## 3.2 GRAPHICAL CHARACTERIZATION OF T-ABSTRACT LINEAR CAUSAL MODELS

To characterize the relation between abstract edges and the concrete graph in a **T**-abstraction, we must take into account that directed paths between *relevant* variables in the concrete graph might be mediated by *irrelevant* variables. Therefore, to study how causal effect propagates, we say that a directed path between two relevant variables is **T**-direct whenever it is mediated by irrelevant variables only. We denote edges and directed paths between two variables $X_1, X_2$ respectively as $X_1 \rightarrow X_2$ and $X_1 \dashrightarrow X_2$.

**Definition 3** (**T**-direct Path). *Let $\mathcal{H}$ be a **T**-abstraction of $\mathcal{L}$, where $\mathcal{H}$ and $\mathcal{L}$ are two SCMs respectively on variables $\boldsymbol{Y}$ and $\boldsymbol{X}$ with graphs $\mathcal{G}_{\mathcal{H}}$ and $\mathcal{G}_{\mathcal{L}}$. Given two concrete variables $X_1, X_2 \in \boldsymbol{X}$, we say that there exists **T**-direct path in $\mathcal{G}_{\mathcal{L}}$, denoted as $X_1 \xrightarrow{\mathbf{T}} X_2$, if and only if there exists a directed path $X_1 \dashrightarrow X_2$ in $\mathcal{G}_{\mathcal{L}}$ such that any other variable $X_3$ in the path is irrelevant, i.e., for any abstract variable $Y$, it holds $X_3 \notin \Pi_R(Y)$.*

First, we show that a **T**-direct path between relevant variables in the concrete graph is a sufficient condition for the presence of an edge between their corresponding abstract variables. Further, as an immediate corollary, a direct path between relevant variables entails an abstract direct path.

**Lemma 2** (Sufficient Abstract Connectivity). *Let $\mathcal{H}$ be a **T**-abstraction of $\mathcal{L}$, where $\mathcal{H}$ and $\mathcal{L}$ are two linear SCMs respectively on variables $\boldsymbol{Y}$ and $\boldsymbol{X}$ with graphs $\mathcal{G}_{\mathcal{H}}$ and $\mathcal{G}_{\mathcal{L}}$. Then, for any pair of relevant variables $X_1, X_2 \in \Pi_R(\boldsymbol{Y})$, such that $X_1 \in \Pi_R(Y_1)$ and $X_2 \in \Pi_R(Y_2)$ with $Y_1 \neq Y_2 \in \boldsymbol{Y}$, it holds*

$$X_1 \xrightarrow{\mathbf{T}} X_2 \ \text{in} \ \mathcal{G}_{\mathcal{L}} \implies Y_1 \rightarrow Y_2 \ \text{in} \ \mathcal{G}_{\mathcal{H}}. \tag{8}$$

*Proof.* We report the proof in Appendix B.3 $\square$

**Corollary 2** (Sufficient Directed Paths). *Let $\mathcal{H}$ be a **T**-abstraction of $\mathcal{L}$, where $\mathcal{H}$ and $\mathcal{L}$ are two linear SCMs respectively on variables $\boldsymbol{Y}$ and $\boldsymbol{X}$ with graphs $\mathcal{G}_{\mathcal{H}}$ and $\mathcal{G}_{\mathcal{L}}$. Then, for any pair of relevant variables $X_1, X_2 \in \Pi_R(\boldsymbol{Y})$, such that $X_1 \in \Pi_R(Y_1)$ and $X_2 \in \Pi_R(Y_2)$ with $Y_1 \neq Y_2 \in \boldsymbol{Y}$, it holds that*

$$X_1 \dashrightarrow X_2 \ \text{in} \ \mathcal{G}_{\mathcal{L}} \implies Y_1 \dashrightarrow Y_2 \ \text{in} \ \mathcal{G}_{\mathcal{H}}. \tag{9}$$

*Proof.* We report the proof in Appendix B.4 $\square$

In our previous results, the faithfulness assumption plays a fundamental role to ensure that causal effect is not canceled out and thus propagates through **T**-direct paths. As we show in Example 1, whenever we allow for cancelling paths we can construct a **T**-abstraction where two abstract variables are not connected despite the presence of a **T**-direct path between their relevant variables.

**Example 1** (Unfaithful Concrete Model). *Consider the following unfaithful linear SCM $\mathcal{L}$ where, given the weights as reported on the edges, the causal effect of $X_1$ on $X_4$ is canceled out. On the right, we show a linear SCM $\mathcal{H}$.*

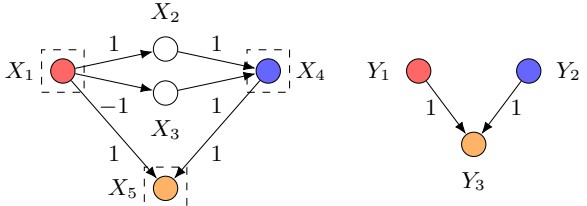

*Consider the linear abstraction function*

$$\mathbf{T} = \begin{bmatrix} 1 & 0 & 0 & 0 & 0 \\ 0 & 0 & 0 & 1 & 0 \\ 0 & 0 & 0 & 0 & 1 \end{bmatrix}^{\top} \quad (10)$$

*that maps each variable in $\boldsymbol{X}$ in $\mathcal{L}$ to the corresponding variable in $\boldsymbol{Y}$ in $\mathcal{H}$, e.g. the first column assigns $X_1$ to $Y_1$, the second column assigns $X_2$ to no high-level variable etc. We visualize the assignments by having the same color for the variables in the two models. Given this abstraction function, $\mathcal{H}$ is a $\mathbf{T}$-abstraction of $\mathcal{L}$, despite the $\mathbf{T}$-direct path $X_1 \xrightarrow{\mathbf{T}} X_4$ between $X_1 \in \Pi_R(Y_1)$ and $X_4 \in \Pi_R(Y_2)$, not having a corresponding path $Y_1 \nrightarrow Y_2$ (Appendix B.5).*

While the presence of a $\mathbf{T}$-direct path between relevant variables is a sufficient condition for the presence of an abstract edge, the converse entails a stronger requirement. It is in fact necessary, for an abstract edge $Y_1 \to Y_2$ to exist, that for each variable in the relevant set of the source node $\Pi_R(Y_1)$ there exists a $\mathbf{T}$-direct path to at least one relevant variable of the target $\Pi_R(Y_2)$. Intuitively, any manipulation on a concrete variable impacts its own abstract variable and, consequently, its descendants in the abstract model. To ensure consistency, it is therefore necessary that the manipulation has an effect on the relevant variables of the descendants (Example 2).

**Theorem 1** (Abstract Connectivity). *Let $\mathcal{H}$ be a $\mathbf{T}$-abstraction of $\mathcal{L}$, where $\mathcal{H}$ and $\mathcal{L}$ are two linear SCMs respectively on variables $\boldsymbol{Y}$ and $\boldsymbol{X}$ with graphs $\mathcal{G}_{\mathcal{H}}$ and $\mathcal{G}_{\mathcal{L}}$. Then, there exists an edge $Y_1 \to Y_2$ in $\mathcal{G}_{\mathcal{H}}$ if and only if for each $X_1 \in \Pi_R(Y_1)$ there exists $X_2 \in \Pi_R(Y_2)$ such that $X_1 \xrightarrow{\mathbf{T}} X_2$ in $\mathcal{G}_{\mathcal{L}}$.*

*Proof.* We report the proof in Appendix B.6 □

By combining the sufficient condition in Lemma 2 and the stronger necessary condition in Theorem 1, we can derive a graphical condition to show whether a model does *not* $\mathbf{T}$-abstract another according to their graphs and the set of relevant variables. We formalize this condition in the following corollary, which we also describe in Example 2.

**Corollary 3** (Connectivity Violation). *Let $\mathcal{H}$ and $\mathcal{L}$ be two linear SCMs respectively on variables $\boldsymbol{Y}$ and $\boldsymbol{X}$ with graphs $\mathcal{G}_{\mathcal{H}}$ and $\mathcal{G}_{\mathcal{L}}$. Consider a linear transformation $\mathbf{T}$ between them leading to the sets of relevant variables $\Pi_R(\boldsymbol{Y})$. If there exists three variables $X_1 \in \Pi_R(Y_1)$, $X_2 \in \Pi_R(Y_2)$, and $X_3 \in \Pi_R(Y_1)$, such that both conditions hold*

- *$X_1 \xrightarrow{\mathbf{T}} X_2$ in $\mathcal{G}_{\mathcal{L}}$, and*
- *for any $X_4 \in \Pi_R(Y_2)$, $X_3 \stackrel{\mathbf{T}}{\nrightarrow} X_4$ is not in $\mathcal{G}_{\mathcal{L}}$,*

*then $\mathcal{H}$ is not a $\mathbf{T}$-abstraction of $\mathcal{L}$.*

*Proof.* We report the proof in Appendix B.7. □

**Example 2** (Abstract Connectivity Violation). *Consider the following linear SCM $\mathcal{L}$ and a linear abstraction transformation $\mathbf{T}$ leading to the reported sets of relevant variables.*

$$\Pi_R(Y_1) \underbrace{\begin{array}{ccc} X_1 & X_2 & X_3 \\ \bullet \to \bullet & \to \bullet \end{array}}_{} \Pi_R(Y_2)$$

*Given the concrete edge $X_2 \xrightarrow{\mathbf{T}} X_3$, there must exist an abstract edge $Y_1 \to Y_2$ (Lemma 2). However, since for $X_3 \in \Pi_R(Y_1)$, the only path $X_1 \dashrightarrow X_3$ to a variable in $\Pi_R(Y_2)$ is mediated by the relevant variable $X_2$, it is not $\mathbf{T}$-direct and thus breaks the conditions of Theorem 1, implying that there should be no abstract edge $Y_1 \to Y_2$ and leading to a contradiction. Intuitively, any two interventions $i = (\Pi_R(Y_1) \leftarrow [a, b])$ and $i' = (\Pi_R(Y_1) \leftarrow [a', b])$ where $a \neq a'$, have the sam ecausal effect on $X_3$, since there is not a $\mathbf{T}$-direct path $X_1 \xrightarrow{\mathbf{T}} X_3$. This breaks interventional consistency as $i, i'$ lead to distinct abstract interventions on $Y_1$ and, thus, to different valus of $Y_2$. However, in the concrete model, the value of $X_3$ is the same regardless of $i, i'$ and, consequently, the value of $Y_2$ is constant. Therefore, given the portrayed graphs and relevant sets, for any choice of both structural and abstraction parameters any abstract model $\mathcal{H}$ does not $\mathbf{T}$-abstract $\mathcal{L}$.*

## 3.3 ORDERING OF CONCRETE BLOCKS INDUCED BY THE ABSTRACT MODEL

Given our definition of $\mathbf{T}$-direct path, we characterized the edges of the abstract graph in terms of the connectivity of the relevant variables. However, despite not influencing directly the abstraction function, we can show that irrelevant variables still contribute to abstract variables and thus have constraints on their causal ordering. In particular, we identify the set of concrete variables whose corresponding exogenous variable contributes to the noise term of the abstract variable. We call this subset of variables the *concrete block* $\Pi(Y)$ of an abstract variable $Y$. To define the concrete block $\Pi(Y)$, we exploit the following corollary, which proves that, whenever the endogenous abstraction function

and the causal models are linear, the exogenous abstraction function $\gamma$ is necessarily a unique linear transformation.

**Corollary 4** (Exogenous Abstraction). *Let $\mathcal{H} = (\boldsymbol{Y}, \boldsymbol{U}, \boldsymbol{g}, \mathbb{P}_{\boldsymbol{U}})$ be a $\mathbf{T}$-abstraction of $\mathcal{L} = (\boldsymbol{X}, \boldsymbol{E}, \boldsymbol{f}, \mathbb{P}_{\boldsymbol{E}})$, where $\mathcal{H}$ and $\mathcal{L}$ are two linear SCMs. Then, the exogenous abstraction function $\gamma \colon \mathcal{D}(\boldsymbol{E}) \to \mathcal{D}(\boldsymbol{U})$, has form*

$$\gamma(\boldsymbol{e}) = \mathbf{S}^{\top}\boldsymbol{e}, \tag{11}$$

*where $\mathbf{S} = \mathbf{FTG}^{-1}$ and $\mathbf{F}, \mathbf{G}$ are the linear transformations of respectively the reduced forms of $\mathcal{L}$ and $\mathcal{H}$, i.e., $\mathcal{L}(\boldsymbol{e}) = \mathbf{F}^{T}\boldsymbol{e}$ and $\mathcal{H}(\boldsymbol{u}) = \mathbf{G}^{T}\boldsymbol{u}$.*

*Proof.* We report the proof in Appendix B.8. $\qquad\square$

**Definition 4** (Concrete Block). *Let $\mathcal{H} = (\boldsymbol{Y}, \boldsymbol{U}, \boldsymbol{g}, \mathbb{P}_{\boldsymbol{U}})$ be a $\mathbf{T}$-abstraction of $\mathcal{L} = (\boldsymbol{X}, \boldsymbol{E}, \boldsymbol{f}, \mathbb{P}_{\boldsymbol{E}})$, where $\mathcal{H}$ and $\mathcal{L}$ are two linear SCMs. We define the concrete block of each abstract variable $Y_j \in \boldsymbol{Y}$ as*

$$\Pi(Y_j) = \{X_i \in \boldsymbol{X} \mid s_{ij} \neq 0\}, \tag{12}$$

*where $s_{ij}$ is the $i$-th element on the $j$-th column of the matrix of the exogenous abstraction function $\mathbf{S}$. Moreover, we define the set of block variables $\Pi(\boldsymbol{Y})$ of the abstract model $\mathcal{H}$ as the union of the blocks of each $Y \in \boldsymbol{Y}$. Formally,*

$$\Pi(\boldsymbol{Y}) = \bigcup_{Y \in \boldsymbol{Y}} \Pi(Y). \tag{13}$$

We prove that the concrete block of an abstract variable contains the set of corresponding relevant variables. In addition, it also contains all the irrelevant variables that are connected to one of these relevant variables through a $\mathbf{T}$-direct path.

**Lemma 3** (Block Composition). *Let $\mathcal{H}$ be a $\mathbf{T}$-abstraction of $\mathcal{L}$, where $\mathcal{H}$ and $\mathcal{L}$ are two linear SCMs respectively on variables $\boldsymbol{Y}$ and $\boldsymbol{X}$. Then, for any abstract variable $Y \in \boldsymbol{Y}$, it holds $X \in \Pi(Y)$ if and only if*

- $X \in \Pi_R(Y)$, *or*
- $X \notin \Pi_R(Y)$, *i.e., $X$ is irrelevant, and there exists $X' \in \Pi_R(Y)$ s.t. $X \xrightarrow{\mathbf{T}} X'$.*

*Proof.* We report the proof in Appendix B.9. $\qquad\square$

Intuitively, this result proves that the irrelevant part of a block lies between the relevant variables of the abstract variable and those of the abstract parents (Example 3).

**Example 3** (Block Composition). *Given a concrete model $\mathcal{L}$ with six variables and an abstract model $\mathcal{H}$ with three variables such that $Y_1 \to Y_3 \leftarrow Y_2$ that is a $\mathbf{T}$-abstraction of $\mathcal{L}$, we visualize a partition of concrete blocks induced by $\mathbf{T}$, where dashed lines denote sets of relevant variables.*

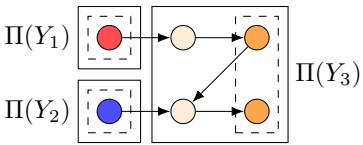

Here block $\Pi(Y_3)$ does not coincide with the set of relevant variables. The irrelevant variables in $\Pi(Y_3)$ have $\mathbf{T}$-direct paths to at least one of the relevant variables $\Pi_R(Y_3)$.

In principle, while sets of relevant variables are mutually disjoint, the rest of the block could be shared without breaking the consistency of the abstraction, as we show in Example 4.

**Example 4** (Block Overlap). *Let $\mathcal{L}$ be a linear SCM represented in the figure below, where the variable $X_2$ is in the block of both $Y_2$ and $Y_3$. Then, any abstract linear SCM $\mathcal{H}$ that is a $\mathbf{T}$-abstraction of $\mathcal{L}$ that induces these concrete blocks is not causally sufficient, since the exogenous terms $U_2, U_3$ in $\mathcal{H}$ are a linear function of respectively $E_2, E_3$ and $E_2, E_4$ in $\mathcal{L}$, and hence they are not independent. Consequently, $Y_2$ and $Y_3$ are confounded in any of these $\mathcal{H}$.*

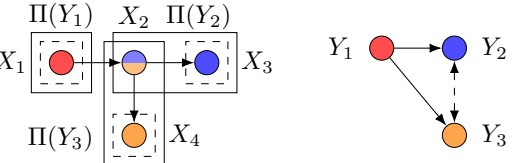

We prove that disjointness of irrelevant variables in a block is a necessary condition to ensure abstract causal sufficiency.

**Lemma 4** (Disjoint Block). *Let $\mathcal{H}$ be a $\mathbf{T}$-abstraction of $\mathcal{L}$, where $\mathcal{H}$ and $\mathcal{L}$ are two linear SCMs respectively on variables $\boldsymbol{Y}$ and $\boldsymbol{X}$. If for any two distinct endogenous variables $Y_1, Y_2$ it holds that $\Pi(Y_1) \cap \Pi(Y_2) \neq \emptyset$, then the abstract model is not causally sufficient.*

*Proof.* We report the proof in Appendix B.10. $\qquad\square$

We now prove our main result of this section, which shows that the causal ordering of the concrete blocks must be consistent with the abstract graph. Intuitively, all relevant variables must follow the abstract order (Theorem 1) and any irrelevant variable in a block must precede at least one relevant variable (Lemma 3). Then, given the causal sufficiency of the abstract model, which ensures that blocks are disjoint (Lemma 4), we can sort concrete blocks according to the causal ordering of the abstract model. Further, we can ignore variables that are not in any block as they must be last in the ordering and thus do not impact abstract variables.

**Theorem 2** (Block Ordering). *Let $\mathcal{H}$ be a $\mathbf{T}$-abstraction of $\mathcal{L}$, where $\mathcal{H}$ and $\mathcal{L}$ are two linear SCMs respectively on variables $\boldsymbol{Y}$ and $\boldsymbol{X}$ with graphs $\mathcal{G}_{\mathcal{H}}$ and $\mathcal{G}_{\mathcal{L}}$. Then, for any valid topological ordering $\prec_{\mathcal{H}}$ of $\mathcal{G}_{\mathcal{H}}$ there exists a valid ordering $\prec_{\mathcal{L}}$ of $\mathcal{G}_{\mathcal{L}}$ such that for any $Y_1, Y_2, Y \in \boldsymbol{Y}$:*

- $Y_1 \prec_{\mathcal{H}} Y_2 \iff \Pi(Y_1) \prec_{\mathcal{L}} \Pi(Y_2)$, and
- $\Pi(Y) \prec_{\mathcal{L}} (X \setminus \Pi(Y))$.

*Proof.* We report the proof in Appendix B.11. □

Given that the ordering of concrete variables depend on the abstract model, we can show that adding or removing variables outside of the blocks still preserves **T**-abstraction.

**Lemma 5** (Submodel Abstraction). *Let $\mathcal{H}$ and $\mathcal{L}$ be two linear SCMs respectively on variables $Y$ and $X$. Then, $\mathcal{H}$ is a **T**-abstraction of $\mathcal{L}$ if and only if $\mathcal{H}$ is a **T**-abstraction of $\mathcal{L}'$, where $\mathcal{L}'$ is a submodel of $\mathcal{L}$ defined on the subset of variables $X' = \Pi(Y)$, i.e., all of the variables in the concrete blocks.*

*Proof.* We report the proof in Appendix B.12. □

### 3.4 CLASS OF T-CONCRETIZATIONS OF AN ABSTRACT MODEL

After having characterized the graphical structure of two linear SCMs in a **T**-abstraction relation, we now focus on how abstraction constraints the parameters of the two models. As we detailed in Lemma 5, variables that are not in any block never cause, either directly or indirectly, relevant variables and thus can be ignored. Therefore, without loss of generality, we consider only variables within the blocks $\Pi(Y)$ of the abstract model. Furthermore, we permute the weights of the concrete model according to the abstract causal ordering with a permutation $\pi_{\mathcal{H}}$, derived from the valid ordering in Theorem 2, as in the following upper-diagonal block matrix

$$\mathbf{W} = \begin{bmatrix} \mathbf{W}_{11} & \mathbf{W}_{12} & \cdots & \mathbf{W}_{1b} \\ 0 & \mathbf{W}_{22} & \cdots & \mathbf{W}_{2b} \\ \vdots & \vdots & \ddots & \vdots \\ 0 & 0 & \cdots & \mathbf{W}_{bb} \end{bmatrix}, \quad (14)$$

where we denote by $\mathbf{W}_{hk} \in \mathbb{R}^{N_h \times N_k}$ the submatrix containing the edges from the concrete block $\Pi(Y_h)$ to $\Pi(Y_k)$. Under the same permutation $\pi_{\mathcal{H}}$, we can also block-wise define the linear abstraction transformation as follows

$$\mathbf{T} = \begin{bmatrix} t_1 & 0 & \cdots & 0 \\ 0 & t_2 & \cdots & 0 \\ \vdots & \vdots & \ddots & \vdots \\ 0 & 0 & \cdots & t_{b.} \end{bmatrix} \quad (15)$$

where each $t_k$ is a vector of size $N_k$. Each of these vectors can still have zero entries for the irrelevant variables. Notably, due to the fact that no irrelevant variable follows a relevant one in the same block, the last component of each vector is non-zero.

Given the same permutation $\pi_{\mathcal{H}}$, the exogenous transformation $\mathbf{S}$ necessarily follows the same structure and is defined

by the endogenous abstraction function and the causal relations among variables in the same block. As a direct consequence, the exogenous and the endogenous transformations coincide whenever a block lacks internal causal relations and, consequently, all variables in the block are relevant.

**Lemma 6** (Exogenous Abstraction). *Let $\mathcal{H} = (Y, U, M, \mathbb{P}_U)$ and $\mathcal{L} = (X, E, W, \mathbb{P}_E)$ be two linear SCMs such that $\mathcal{H}$ is a **T**-abstraction of $\mathcal{L}$, such that $W$ follows permutation $\pi_{\mathcal{H}}$. Then, the exogenous abstraction function $\gamma \colon \mathcal{D}(E) \to \mathcal{D}(U)$ is unique and has form $\gamma(e) = \mathbf{S}^\top e$ for a linear transformation $\mathbf{S} \in \mathbb{R}^{d \times b}$ defined as the upper-diagonal block matrix*

$$\mathbf{S} = \begin{bmatrix} s_1 & 0 & \cdots & 0 \\ 0 & s_2 & \cdots & 0 \\ \vdots & \vdots & \ddots & \vdots \\ 0 & 0 & \cdots & s_{b,} \end{bmatrix} \quad (16)$$

*where $s_k = \mathbf{F}_{kk} t_k = (\mathbf{I} - \mathbf{W}_{kk})^{-1} t_k$ for any $Y_k \in Y$.*

*Proof.* We report the proof in Appendix B.13. □

Given the structure and the ordering induced by the abstraction function, we introduce a provably equivalent formulation of **T**-abstraction entirely based on the model parameters. In this way, we guarantee interventional consistency on all possible abstract hard interventions as a property of the weights of the two linear SCMs. Further, by assessing abstraction in closed-form, we can characterize the set of **T**-concretizations of an abstract model (Example 5).

**Theorem 3** (Block Abstraction). *Let $\mathcal{H} = (Y, U, M, \mathbb{P}_U)$ and $\mathcal{L} = (X, E, W, \mathbb{P}_E)$ be two linear SCMs with graphs $\mathcal{G}_{\mathcal{H}}$ and $\mathcal{G}_{\mathcal{L}}$ respectively. Then $\mathcal{H}$ is a linear **T**-abstraction of $\mathcal{L}$ if and only if for any valid topological ordering $\prec_{\mathcal{H}}$ of $\mathcal{G}_{\mathcal{H}}$ there exists a valid ordering $\prec_{\mathcal{L}}$ of $\mathcal{G}_{\mathcal{L}}$ such that, for any $Y_i, Y_j \in Y$ it holds*

$$Y_i \prec_{\mathcal{H}} Y_j \iff \Pi(Y_i) \prec_{\mathcal{L}} \Pi(Y_j), \text{ and} \quad (17)$$
$$\mathbf{W}_{ij} s_j = m_{ij} t_i, \quad (18)$$

*where $\mathbf{W}_{ij}$ is the $i$-th element on the $j$-th column of $\mathbf{W}$, and $m_{ij}$ is the $i$-th element on the $j$-th column of $\mathbf{M}$.*

*Proof.* We report the proof in Appendix B.14. □

**Example 5** (T-Concretization Class). *Let $\mathcal{H}$ be an abstract causal model with two variables such that $Y_1 \to Y_2$ with unitary weight, and let $\mathbf{T}$ be the following transformation*

$$\mathbf{T} = \begin{bmatrix} 1 & 1 & 0 & 0 \\ 0 & 0 & 1 & 1 \end{bmatrix}^\top, \quad (19)$$

*Then, of the three following linear SCMs, we can easily verify that only the first two models are **T**-abstracted by $\mathcal{H}$.*

**Algorithm 1: T-Concretization Sampling**

**Input:** Abstract adjacency matrix $\mathbf{M} \in \mathbb{R}^{b \times b}$
Abstraction function $\mathbf{T} \in \mathbb{R}^{d \times b}$

**Result:** Concrete adjacency matrix $\mathbf{W} \in \mathbb{R}^{d \times d}$

$\mathbf{W} \leftarrow \mathbf{0}$ ▷ Init Concrete Weights
**for** $Y_j \in \mathbf{Y}$ **do** ▷ Abstract Target Node
  $N_j \leftarrow |\Pi(Y_j)|$
  $\mathbf{W}_{jj} \leftarrow \text{RandomDAG}(N_j)$ ▷ Target Block Weights
  $\mathbf{s}_j \leftarrow (\mathbf{I} - \mathbf{W}_{jj})^{-1}\mathbf{t}_j$
  **for** $Y_i \in \mathbf{Y}$ **do** ▷ Abstract Source Node
    **for** $X_k \in \Pi(Y_i)$ **do** ▷ Source Block
      $\mathbf{v} \sim \{\mathbf{v} \in \mathbb{R}^{N_j} \mid \sum_{h=1}^{N_j} v_h = 1\}$
      $\mathbf{c} \leftarrow \mathbf{v}/\mathbf{s}_j$ ▷ Right-Inverse of $\mathbf{s}_j$
      $[\mathbf{W}_{ij}]_{k,:} \leftarrow m_{ij}[\mathbf{t}_i]_k \mathbf{c}^\top$ ▷ Assign $k$-th row
    **end**
  **end**
**end**

---

**Algorithm 2: Abs-LiNGAM**

**Input:** Concrete Observational Dataset $\mathcal{D}_\mathcal{L}$,
Joint Observational Dataset $\mathcal{D}_J$.

**Result:** Abstraction function $\hat{\mathbf{T}} \in \mathbb{R}^{d \times b}$,
Abstract adjacency matrix $\hat{\mathbf{M}} \in \mathbb{R}^{b \times b}$,
Concrete adjacency matrix $\hat{\mathbf{W}} \in \mathbb{R}^{d \times d}$.

$\hat{\mathbf{T}} \leftarrow \arg\min_{\mathbf{T} \in \mathbb{R}^{b \times d}} \sum_{(\mathbf{x},\mathbf{y}) \in \mathcal{D}_J} \|\mathbf{x}^\top \mathbf{T} - \mathbf{y}^\top\|_2^2$;
**for** $Y_i \in \mathbf{Y}$ **do** ▷ Select Relevant Variables
  $\hat{\Pi}_R(Y_i) \leftarrow \{X_k \in \mathbf{X} \mid [\hat{\mathbf{t}}_i]_k \neq 0\}$
**end**

$\mathcal{D}_{\hat{\mathcal{H}}} \leftarrow \{\hat{\mathbf{T}}^\top \mathbf{x} \mid \mathbf{x} \in \mathcal{D}_\mathcal{L}\}$ ▷ Create Abstract Dataset
$\hat{\mathbf{M}} \leftarrow \text{DirectLiNGAM}(\mathcal{D}_{\hat{\mathcal{H}}}, \emptyset)$ ▷ Abstract Discovery
$\mathbf{K} \leftarrow \emptyset$
**for** $Y_i, Y_j \in \mathbf{Y}$ **do** ▷ Collect Prior Knowledge
  **if** $Y_i \not\dashrightarrow Y_j$ **then** ▷ Check Ancestorship in $\hat{\mathbf{M}}$
    **for** $X_k \in \hat{\Pi}_R(Y_i), X_h \in \hat{\Pi}_R(Y_j)$ **do**
      $\mathbf{K} \leftarrow \mathbf{K} \cup \{X_k \not\dashrightarrow X_h\}$
    **end**
  **end**
**end**
$\hat{\mathbf{W}} \leftarrow \text{DirectLiNGAM}(\mathcal{D}_\mathcal{L}, \mathbf{K})$ ▷ Concrete Discovery

---

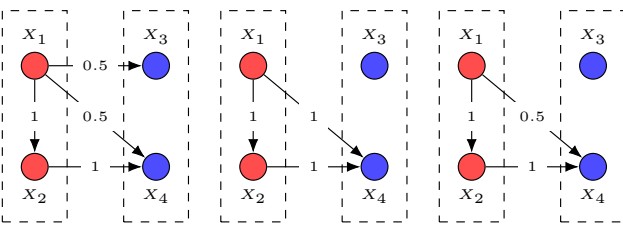

*Given the identical inner-block connections, the exogenous abstraction function is the same for all three models, as in*

$$\mathbf{s}_1 = \begin{bmatrix} 2 \\ 1 \end{bmatrix}, \quad \mathbf{s}_2 = \begin{bmatrix} 1 \\ 1 \end{bmatrix}. \tag{20}$$

*Then, only for the first two models it holds $\mathbf{W}_{12}\mathbf{s}_2 = \mathbf{t}_1$.*

By building on our novel formulation, we define a complete and sound procedure to sample concrete models from an abstract adjacency matrix and a linear abstraction function (Algorithm 1). First, for each abstract target variable $Y_j$, the algorithm samples the inner-block weights $\mathbf{W}_{jj}$, where we assume that any irrelevant variable has at least a relevant variable as a descendant. Consequently, all variables are members of the block. Then, for each source variable $Y_i$, the algorithm samples consistent coefficients $\mathbf{W}_{ij}$ respecting Theorem 3 by first sampling a right-inverses of the exogenous abstraction function $\mathbf{s}_j$. Since the generated model follows the abstract causal ordering and Theorem 3 by construction, it is a valid concretization.

# 4 ABSTRACT INFORMATION FOR NON-GAUSSIAN LINEAR DISCOVERY

In this section, we introduce Abs-LiNGAM (Algorithm 2), a strategy to exploit our results on $\mathbf{T}$-abstraction to speedup observational causal discovery of linear non-Gaussian models, e.g. LiNGAM [Shimizu et al., 2011]. The intuition is that whenever we have a $\mathbf{T}$-abstraction of an unknown model to learn, we can exclude all the candidate solutions not satisfying the graphical conditions we presented in the previous sections. Furthermore, in Abs-LiNGAM, we demonstrate how to infer prior knowledge for the concrete model from a small number of paired concrete-abstract samples, even when the abstract model and the abstraction function are unknown, and an abstract dataset is not directly available. In the following, we formalize the data-generation process and the steps of Abs-LiNGAM.

## 4.1 DATA-GENERATION PROCESS

As in many real-world applications, where observations are produced by sensors or other data-collecting devices, we assume that samples from the low-level concrete model have a significantly larger availability than high-level abstract samples. We formalize this intuition by defining two datasets

$$\mathcal{D}_\mathcal{L} \sim \mathbb{P}_\mathcal{L} \tag{21}$$
$$\mathcal{D}_J \sim \mathbb{P}_{\mathcal{L},\mathcal{H}}, \tag{22}$$

where the former contains concrete samples only and the latter paired observations from the joint observational distribution of both models, such that $|\mathcal{D}_J| \ll |\mathcal{D}_\mathcal{L}|$. Therefore, we define the following data-generating process, where we

produce a significantly lower number of abstract samples.

$$\boldsymbol{e}^{(i)} \sim \text{Exponential} \qquad \text{for } i = 1, \ldots, |\mathcal{D}_{\mathcal{L}}|, \qquad (23)$$

$$\boldsymbol{x}^{(i)} = \mathcal{L}(\boldsymbol{e}^{(i)}) \qquad \text{for } i = 1, \ldots, |\mathcal{D}_{\mathcal{L}}|, \qquad (24)$$

$$\boldsymbol{y}^{(i)} = \mathcal{H}(\gamma(\boldsymbol{e}^{(i)})) \qquad \text{for } i = 1, \ldots, |\mathcal{D}_J|. \qquad (25)$$

Since we assume linear and non-Gaussian data, the models are identifiable in the limit of infinite data [Shimizu et al., 2006]. In Appendix D, we discuss preliminary results to tackle an additional scenario where we consider abstract observations to be perturbed by random noise.

## 4.2 ABS-LINGAM

**T-Reconstruction.** Since we assume a linear transformation, we can fit the abstraction function from the joint dataset $\mathcal{D}_J$ by solving a least-squares problem [Trefethen and Bau, 2022]. Then, for each abstract variable $Y_i \in \boldsymbol{Y}$, we can identify its set of relevant variables $\hat{\Pi}_R(Y)$, as

$$\hat{\Pi}_R(Y_i) = \{X_k \mid [\hat{\boldsymbol{t}}_i]_k \neq 0\}. \qquad (26)$$

In practice, we mask the coefficients of the fitted abstraction transformation $\hat{\mathbf{T}}$ with a small threshold to handle numerical instability, which, whenever a sufficient number of joint samples $|\mathcal{D}_{\mathcal{J}}|$ is available, ensures that each relevant block pertains to a single abstract variable.

**Abstract Causal Discovery.** Then, we focus on learning the abstract causal structure from data. Since we assume abstract samples to be scarce, even in our simplified setting of linear and non-Gaussian models, the abstract model might not be discoverable by the high-level samples in the joint dataset $\mathcal{D}_J$ alone. However, after having identified the abstraction function, we can use it on the concrete dataset to abstract each sample as in

$$\mathcal{D}_{\hat{\mathcal{H}}} = \{\hat{\mathbf{T}}^{\top} \boldsymbol{x} \mid \boldsymbol{x} \in \mathcal{D}_{\mathcal{L}}\}. \qquad (27)$$

In fact, whenever the target model is a $\mathbf{T}$-abstraction, the observational consistency property ensures that abstracting concrete samples is equivalent to directly sampling from the abstract distribution, as in the data-generating process. Then, we can use the newly generated abstract samples with any causal discovery algorithm for linear non-Gaussian models.

**Concrete Causal Discovery** Finally, we can use the constraints induced by the abstract model to speedup discovery of the concrete causal model. As an immediate consequence of Theorem 1, the existence of an abstract directed path $Y_i \dashrightarrow Y_j$ entails the existence of at least a concrete directed path between variables in the corresponding relevant blocks $\Pi_R(Y_i)$ and $\Pi_R(Y_j)$. We cannot, however, directly infer which of the possibly many ancestral relations the concrete model contains. On the other hand, whenever an

abstract path does *not* exist, we can infer that any variable in the source block does not cause, neither directly or indirectly, any variable in the target block. We can therefore restrict the search space of the concrete causal discovery problem by excluding all solutions that do not satisfy the following set of constraints

$$\boldsymbol{K} = \{X_k \not\dashrightarrow X_h \mid X_k \in \Pi_R(Y_i) \\ \wedge X_h \in \Pi_R(Y_j) \wedge Y_i \not\dashrightarrow Y_j\}. \qquad (28)$$

We use the DirectLiNGAM algorithm [Shimizu et al., 2011] to solve the concrete causal discovery problem, as it can integrate prior knowledge in the form of forbidden direct paths and thus restrict the set of candidate solutions.

## 5 EXPERIMENTAL RESULTS

In this section, we discuss our analysis on the performance of Abs-LiNGAM (Algorithm 2) on simulated data. In particular, we validate whether a small amount of paired concrete-abstract observations can reduce the search space, and thus the execution time, of DirectLiNGAM [Shimizu et al., 2011], without compromising the quality of the retrieved concrete causal structure. As baseline, we compare against applying DirectLiNGAM to the concrete dataset without any abstract-induced prior knowledge.

For each run, we sample the parameters of an abstraction function and of an abstract linear SCM. We then generate a concrete causal model by sampling one of the possible $\mathbf{T}$-concretizations of the abstract model with Algorithm 1. We provide details on our experimental setup and additional results respectively in Appendix C and in Appendix E.

We study the performance of Abs-LiNGAM for an increasing number of paired samples (Figure 2a). Since Abs-LiNGAM is a multi-step algorithm, the quality of the retrieved concrete causal model strictly depends on the correctness of the abstraction function, the consequent generated abstract data and abstract causal discovery. As expected, whenever the size of the paired dataset $|\mathcal{D}_J|$ is too small, Abs-LiNGAM wrongly identifies concrete paths as forbidden and, compared to the baseline, fails to retrieve the correct concrete causal model. However, whenever the number of paired samples approaches the number of concrete nodes $|\boldsymbol{X}|$, Abs-LiNGAM performs similarly to the baseline and correctly retrieves the concrete causal model. We observe the same trend for concrete graphs of increasing size (Figure 2b), highlighting how prior knowledge induced from the abstract model significantly reduces the execution time compared to the baseline.

Furthermore, we found that bootstrapping abstract causal discovery, i.e., aggregating several iterations on randomly extracted sub-datasets, improves the performance on the downstream concrete discovery task without noticeably affecting the execution time, which is still dominated by the

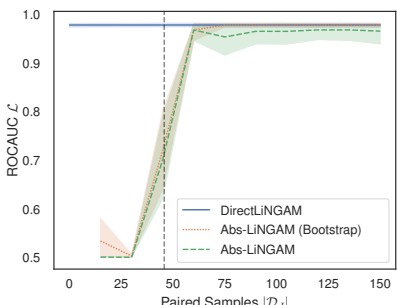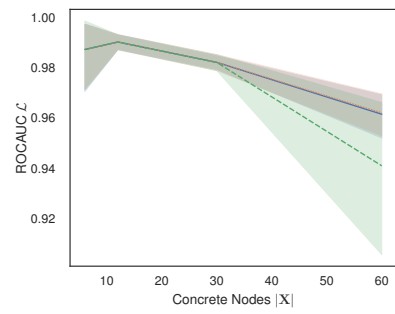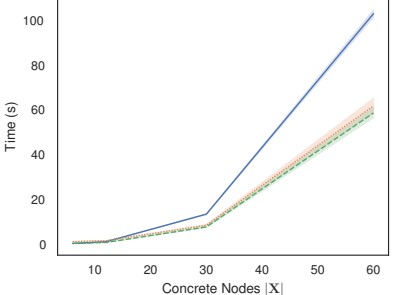

(a) Performance over Paired Samples $|\mathcal{D}_J|$      (b) Performance and Execution Time (s) over Concrete Graph Size $|\boldsymbol{X}|$

Figure 2: We report the performance of Abs-LiNGAM for (a) an increasing number of paired samples $|\mathcal{D}_J|$ and (b) an increasing number of concrete nodes $|\boldsymbol{X}|$ . We plot a variant of Abs-LiNGAM where we bootstrap the abstract causal discovery step with five repetitions. We report the area under the ROC curve and the execution time over 30 runs on randomly generated Erdős-Rényi abstract graphs with $b = 5$ nodes and 8 edges. In the first experiment, we sample for each abstract graph a concrete model with random size $|\boldsymbol{X}| \in [25, 50]$. In the second experiment, we also vary the number of paired samples to always be twice the number of concrete nodes.

final concrete causal discovery run.

# 6   RELATED WORKS

Several works addressed the problem of clustering together variables to reduce dimensionality and maintain the identifiability of causal effect. Both Anand et al. [2023] and Wahl et al. [2023] deal with the problem of partitioning a causal graph into clusters where causal relations at the micro-level are translated as causal edges at the macro-level. Tikka et al. [2023] study instead a particular class of groups, which they define as *transit clusters*, where only part of the variables are allowed to have ingoing or outgoing edgs from the cluster.

Differently from previous works, our work focuses instead on the necessary conditions for causal abstraction and results in different definitions for the grouping of micro-variables. It is however an interesting direction to assess whether different assumptions, for instance on the intervention map, might lead to comparable definitions.

In parallel, several recent papers explored the problem of fitting an abstraction function from data by focusing on either discrete [Zennaro et al., 2023, Felekis et al., 2024] or linear [Kekić et al., 2023, Geiger et al., 2024] SCMs. Notably, apart from interventional samples, all these works assume to have at least partial knowledge of the graphs, the intervention map, or the set of concrete relevant variables corresponding to each abstract one.

Based on our theoretical results on the graphical and parametric conditions of linear abstraction for linear causal models, we instead propose to learn both the abstract and the concrete model, and their abstraction function directly from observational data and without any prior knowledge or any constraint on the graphical structure of the two models.

# 7   CONCLUSION

In this paper, we studied the necessary and sufficient conditions on the causal ordering and the parameters for two linear Structural Causal Models to be in a linear abstraction. Furthermore, we introduced the first procedure to sample from the set of all possible concretizations of an abstract SCM, which can be used in other abstraction applications.

We also proposed Abs-LiNGAM, a strategy to speedup causal discovery of a linear non-Gaussian concrete causal model given an additional dataset of paired observations on concrete and abstract variables. Finally, we empirically highlighted how Abs-LiNGAM leverages abstract information to reduce the search space and improve execution time without sacrificing on the quality of the discovered structure.

An interesting direction for future work is to extend our results to non-linear models and non-linear abstraction functions, and to tackle the causal sufficiency assumption, which requires full-observability of the concrete realizations.

### Acknowledgements

This work has been supported by EU-EIC EMERGE (Grant No. 101070918), by H2020 TAILOR (Grant No. 952215) and by the EU NextGenerationEU programme under the funding schemes PNRR-PE-AI (PE00000013) FAIR — Future Artificial Intelligence Research.

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

# APPENDIX

We organize the Appendix as follows. In Appendix A, we report further information on our notation by summarizing it in a glossary. Then, in Appendix B, we report all the proof for the theoretical results discussed in the main body. Finally, in Appendix C, we present further details on the generative process of the synthetic datasets used for our empirical study, of which we report additional results in Appendix E.

## A GLOSSARY

| Notation | Definition |
|---|---|
| $X$ | Set of endogenous concrete variables |
| $E$ | Set of exogenous concrete variables |
| $Y$ | Set of endogenous abstract variables |
| $U$ | Set of exogenous abstract variables |
| $d$ | Number of concrete variables |
| $b$ | Number of abstract variables |
| $\mathcal{L}$ | Concrete Causal Model |
| $\mathcal{H}$ | Abstract Causal Model |
| $\mathbf{W}$ | Weighted Adjacencies of $\mathcal{L}$ |
| $\mathbf{M}$ | Weighted Adjacencies of $\mathcal{H}$ |
| $\mathbf{F}$ | Reduced Form of $\mathcal{L}$ |
| $\mathbf{G}$ | Reduced Form of $\mathcal{H}$ |
| $\tau$ | Endogenous Abstraction Function |
| $\gamma$ | Exogenous Abstraction Function |
| $\mathbf{T}$ | Linear Endog. Abstraction Transformation |
| $\mathbf{S}$ | Linear Exog. Abstraction Transformation |
| $t_j$ | Vector in $\mathbf{T}$ abstracting $Y_j$ from $\Pi_R(Y_j)$ |
| $s_j$ | Vector in $\mathbf{S}$ abstracting $U_j$ from $e_{\Pi(Y_j)}$ |
| $\Pi_R(Y)$ | Set of relevant variables for $Y$ |
| $\Pi(Y)$ | Block of $Y$ |
| $N_i$ | Number of variables in $\Pi(Y_i)$ |
| $\mathbf{W}_{ij}$ | Submatrix of weights from $\Pi(Y_i)$ to $\Pi(Y_j)$ |
| $\mathbf{F}_{ij}$ | Submatrix of sub-model from $\Pi(Y_i)$ to $\Pi(Y_j)$ |
| $\mathcal{D}_{\mathcal{L}}$ | Dataset sampled from $\mathbb{P}_X$ |
| $\mathcal{D}_J$ | Dataset sampled from the joint $\mathbb{P}_{X,Y}$ |

## B PROOFS

### B.1 LEMMA 1

**Lemma 1 (Disjoint Relevant).** Let $\mathcal{H}$ be a $\mathbf{T}$-abstraction of $\mathcal{L}$, where $\mathcal{H}$ and $\mathcal{L}$ are two linear SCMs respectively on variables $Y$ and $X$. Then, for any pair of distinct abstract variables $Y_1, Y_2 \in Y$, it holds that $\Pi_R(Y_1) \cap \Pi_R(Y_2) = \emptyset$, where $\Pi_R(Y_1) \neq \emptyset$ and $\Pi_R(Y_2) \neq \emptyset$.

*Proof.* Firstly, we show that given an abstract intervention $j = (Y_1 \leftarrow k)$ on $Y_1$, any concrete intervention $i$ such that $\omega(i) = j$ must fix all relevant variables $\Pi_R(Y_1)$. Otherwise,

if we assume the existence of a non-intervened variable $X_s \in \Pi_R(Y_1)$ the function $\mathcal{L}^i_{\Pi_R(Y_1)}$ would be non-constant. Therefore, since $\tau_{Y_1}$ depends on $X_s$ by definition of relevant variable, interventional consistency would not hold, as in

$$\tau_{Y_1} \circ \mathcal{L}^i_{\Pi_R(Y_1)} \neq \mathcal{H}^j_{Y_1} \circ \gamma = k. \tag{29}$$

Therefore, for any abstract intervention $j = (Y_1 \leftarrow k)$, the corresponding concrete interventions must have form

$$i = (\Pi_R(Y_1) \leftarrow v), \tag{30}$$

for any vector $v$ such that $\tau_{Y_1}(v) = k$, without intervening on further relevant variables.

We firstly prove Lemma 1 whenever $Y_1 \not\dashrightarrow Y_2$. Then, we assume the existence of a non-empty subset $V = \Pi_R(Y_1) \cap \Pi_R(Y_2)$ of shared variables. Since $Y_1$ has no causal effect on $Y_2$, given an high-level intervention $j = (Y_1 \leftarrow k)$, it must hold that

$$\mathcal{H}^j_{Y_2} = \mathcal{H}_{Y_2}. \tag{31}$$

However, by intervening on $Y_1$, any concretization must also fix $V$. Therefore, we prove the property by contradiction, as

$$\mathcal{H}^j_{Y_2} \circ \gamma = \tau_{Y_2} \circ \mathcal{L}^i_{\Pi_R(Y_2)} \tag{32}$$

$$\neq \tau_{Y_2} \circ \mathcal{L}_{\Pi_R(Y_2)} \tag{33}$$

$$= \mathcal{H}_{Y_2} \circ \gamma \tag{34}$$

$$\implies \mathcal{H}^j_{Y_2} \neq \mathcal{H}_{Y_2}, \tag{35}$$

given the surjectivity of $\gamma$ and the lack of cancelling paths.

Finally, we can tackle the last scenario, where $Y_1 \dashrightarrow Y_2$, by showing that $\Pi_R(Y) \cap \Pi_R(\text{An}(Y)) = \emptyset$, where $\text{An}(Y)$ is the set of ancestors of $Y$. Given the model acyclicity, for any abstract intervention $j = (Y \leftarrow k)$, it must hold

$$\mathcal{H}^j_{\text{An}(Y)} = \mathcal{H}_{\text{An}(Y)}. \tag{36}$$

However, if the relevant variables of $Y$ were to overlap with the relevant variables of its ancestors, we could show that

$$\mathcal{H}^j_{\text{An}(Y)} \circ \gamma = \tau_{\text{An}(Y)} \circ \mathcal{L}^i_{\text{An}(Y)} \tag{37}$$

$$\neq \tau_{\text{An}(Y)} \circ \mathcal{L}_{\text{An}(Y)} \tag{38}$$

$$= \mathcal{H}_{\text{An}(Y)} \circ \gamma \tag{39}$$

$$\implies \mathcal{H}^j_{\text{An}(Y)} \neq \mathcal{H}_{\text{An}(Y)}. \tag{40}$$

Therefore, since interventional consistency does not hold, $\mathcal{H}$ is not a $\mathbf{T}$-abstraction of $\mathcal{L}$, which contradicts the hypothesis and concludes the proof. $\square$

### B.2 COROLLARY 1

**Corollary 1 (Constructive Abstraction).** Let $\mathcal{H}$ be a strong $\tau$-abstraction of $\mathcal{L}$ where $\mathcal{H}$ and $\mathcal{L}$ are linear SCMs and $\tau$ is a linear transformation. Then, $\mathcal{H}$ is a constructive $\tau$-abstraction of $\mathcal{L}$.

*Proof.* By definition of linear transformation, the set of low-level variables on which an abstract variable $Y \in \boldsymbol{Y}$ depends through the linear abstraction function $\tau$ coincides with its set of relevant variables $\Pi_R(Y) \subseteq \boldsymbol{X}$. Therefore, by showing that the relevant sets are disjoint whenever the SCMs $\mathcal{H}$ and $\mathcal{L}$ are linear, a **T**-abstraction on linear SCMs is also a constructive abstraction. By definition of **T**-abstraction, this is equivalent to state that a linear $\tau$-abstraction on linear SCMs is a constructive abstraction under our assumption on the absence of cancelling paths. $\square$

## B.3 LEMMA 2

**Lemma 2 (Sufficient Directed Paths)** Let $\mathcal{H}$ be a **T**-abstraction of $\mathcal{L}$, where $\mathcal{H}$ and $\mathcal{L}$ are two linear SCMs respectively on variables $\boldsymbol{Y}$ and $\boldsymbol{X}$ with graphs $\mathcal{G}_\mathcal{H}$ and $\mathcal{G}_\mathcal{L}$. Then, for any pair of relevant variables $X_1, X_2 \in \Pi_R(\boldsymbol{Y})$, such that $X_1 \in \Pi_R(Y_1)$ and $X_2 \in \Pi_R(Y_2)$ with $Y_1 \neq Y_2 \in \boldsymbol{Y}$, it holds

$$X_1 \xrightarrow{\mathbf{T}} X_2 \text{ in } \mathcal{G}_\mathcal{L} \implies Y_1 \to Y_2 \text{ in } \mathcal{G}_\mathcal{H}. \quad (41)$$

*Proof.* Let $Y_1, Y_2$ be two distinct abstract variables and let $i, i'$ be two concrete interventions that fix any *relevant* variable except for those in the relevant set $\Pi_R(Y_2)$, and whose assignments differ only in $X_1 \in \Pi_R(Y_1)$. Formally,

$$i = (\boldsymbol{V} \leftarrow \boldsymbol{v}, \Pi_R(Y_1) \leftarrow \boldsymbol{c}) \quad (42)$$
$$i' = (\boldsymbol{V} \leftarrow \boldsymbol{v}, \Pi_R(Y_1) \leftarrow \boldsymbol{c}'), \quad (43)$$

where

$$\boldsymbol{V} = \bigcup_{Y \in \boldsymbol{Y} \setminus \{Y_1, Y_2\}} \Pi_R(Y). \quad (44)$$

Given $X_1 \xrightarrow{\mathbf{T}} X_2$, there exists at least a directed path composed only of non-relevant variables, that are therefore non-intervened. Consequently, due to the *faithfulness* assumption, the concrete model does not have cancelling paths and, therefore, an intervention on a variable always has an effect on its descendants. In particular, since $i, i'$ constrain $X_1$ to two different values, it holds that

$$\mathcal{L}^i_{X_2} \neq \mathcal{L}^{i'}_{X_2} \quad (45)$$
$$\tau_{Y_2} \circ \mathcal{L}^i \neq \tau_{Y_2} \circ \mathcal{L}^{i'} \quad (46)$$
$$\mathcal{H}^j_{Y_2} \circ \gamma \neq \mathcal{H}^{j'}_{Y_2} \circ \gamma, \quad (47)$$

where, given the intervention map, the concrete interventions correspond to the following abstract interventions

$$j = (\boldsymbol{Y} \setminus \{Y_1, Y_2\} \leftarrow \tau(\boldsymbol{v}), Y_1 \leftarrow \tau_{Y_1}(\boldsymbol{c})) \quad (48)$$
$$j' = (\boldsymbol{Y} \setminus \{Y_1, Y_2\} \leftarrow \tau(\boldsymbol{v}), Y_1 \leftarrow \tau_{Y_1}(\boldsymbol{c}')). \quad (49)$$

Therefore, due to the surjectivity of $\gamma$, it also holds

$$\mathcal{H}^j_{Y_2} \neq \mathcal{H}^{j'}_{Y_2}. \quad (50)$$

Consequently, since $j$ and $j'$ differ only in $Y_1$ and fix everything but $Y_2$, $Y_1$ has a direct effect on $Y_2$, i.e., $Y_1 \to Y_2$. $\square$

## B.4 COROLLARY 2

**Corollary 2 (Sufficient Directed Paths)** Let $\mathcal{H}$ be a **T**-abstraction of $\mathcal{L}$, where $\mathcal{H}$ and $\mathcal{L}$ are two linear SCMs respectively on variables $\boldsymbol{Y}$ and $\boldsymbol{X}$ with graphs $\mathcal{G}_\mathcal{H}$ and $\mathcal{G}_\mathcal{L}$. Then, for any pair of relevant variables $X_1, X_2 \in \Pi_R(\boldsymbol{Y})$, such that $X_1 \in \Pi_R(Y_1)$ and $X_2 \in \Pi_R(Y_2)$ with $Y_1 \neq Y_2 \in \boldsymbol{Y}$, it holds that

$$X_1 \dashrightarrow X_2 \text{ in } \mathcal{G}_\mathcal{L} \implies Y_1 \dashrightarrow Y_2 \text{ in } \mathcal{G}_\mathcal{H}. \quad (51)$$

*Proof.* Given Lemma 2, whenever there exists a **T**-direct path between relevant variables $X_1 \in \Pi_R(Y_1)$ and $X_2 \in \Pi_R(Y_2)$ there must exist an abstract edge $Y_1 \to Y_2$. However, if the path is not **T**-direct, then there must exists some relevant variable $X_3 \in \Pi_R(Y_3)$ for another abstract variable $Y_3$ along the path. We firstly consider the case where $Y_3 \neq Y_1$ and $Y_3 \neq Y_2$. Consequently, there must exist an edge $Y_1 \to Y_3$ and, by applying the same argument on the path $X_3 \dashrightarrow X_2$, the corollary holds for $Y_1 \dashrightarrow Y_2$. Due to the acyclicity of the abstract graph, the case where $Y_3 = Y_1$ or $Y_3 = Y_2$ can arise only at the beginning (resp. the end) of the path. In this case, we could consider the successive variable until we get one different from $Y_1, Y_2$, if any. If there is none, then there exists a **T**-direct path between the relevant variables of $Y_1, Y_2$ and we fallback to the scenario of Lemma 2, which directly entails $Y_1 \dashrightarrow Y_2$. $\square$

## B.5 EXAMPLE 1

**Example 1 (Unfaithful Concrete Model)**

*Proof.* To prove **T**-abstraction of the example, we anticipate the parametrical characterization of linear abstraction which we introduce in Section 3.4. In particular, given the adjacencies of the model,

$$\mathbf{W} = \begin{bmatrix} 0 & 1 & -1 & 0 & 1 \\ 0 & 0 & 0 & 1 & 0 \\ 0 & 0 & 0 & 1 & 0 \\ 0 & 0 & 0 & 0 & 1 \\ 0 & 0 & 0 & 0 & 0 \end{bmatrix} \quad (52)$$

$$\mathbf{M} = \begin{bmatrix} 0 & 0 & 1 \\ 0 & 0 & 1 \\ 0 & 0 & 0 \end{bmatrix} \quad (53)$$

the necessary form for the exogenous abstraction function, which we will introduce in Lemma 6, is

$$\mathbf{S} = \begin{bmatrix} 1 & 0 & 0 \\ 0 & 1 & 0 \\ 0 & 1 & 0 \\ 0 & 1 & 0 \\ 0 & 0 & 1 \end{bmatrix}. \quad (54)$$

Consequently, we can prove abstraction by showing that for any $Y_i, Y_j$ it holds that

$$\mathbf{W}_{ij}\boldsymbol{s}_j = m_{ij}\boldsymbol{t}_i. \tag{55}$$

For this example, of particular interest is the case $Y_1 \to Y_2$, where it holds that

$$\mathbf{W}_{1,2}\boldsymbol{s}_2 = m_{1,2}\boldsymbol{t}_1 \tag{56}$$

$$\begin{bmatrix} 1 & -1 & 0 \end{bmatrix} \begin{bmatrix} 1 \\ 1 \\ 1 \end{bmatrix} = 0 \cdot \begin{bmatrix} 1 \end{bmatrix} \tag{57}$$

$$0 = 0, \tag{58}$$

and thus $\mathcal{H}$ **T**-abstracts $\mathcal{L}$. $\quad\square$

## B.6 THEOREM 1

**Theorem 1 (Abstract Connectivity)** Let $\mathcal{H}$ be a **T**-abstraction of $\mathcal{L}$, where $\mathcal{H}$ and $\mathcal{L}$ are two linear SCMs respectively on variables $\boldsymbol{Y}$ and $\boldsymbol{X}$ with graphs $\mathcal{G}_{\mathcal{H}}$ and $\mathcal{G}_{\mathcal{L}}$. Then, there exists an edge $Y_1 \to Y_2$ in $\mathcal{G}_{\mathcal{H}}$ if and only if for each $X_1 \in \Pi_R(Y_1)$ there exists $X_2 \in \Pi_R(Y_2)$ such that $X_1 \xrightarrow{\mathbf{T}} X_2$ in $\mathcal{G}_{\mathcal{L}}$.

*Proof.* The sufficient condition follows immediately from Lemma 2, where we already proved that any **T**-direct path between relevant variables entails an abstract edge.

To prove the necessary condition, we consider instead two abstract interventions $j, j'$ which differ only in $Y_1$ and fix everything but $Y_2$. Formally,

$$j = (Y_1 \leftarrow k, \boldsymbol{V} \leftarrow \boldsymbol{v}) \tag{59}$$
$$j' = (Y_1 \leftarrow k', \boldsymbol{V} \leftarrow \boldsymbol{v}), \tag{60}$$

where $\boldsymbol{V} = \boldsymbol{Y} \setminus \{Y_1, Y_2\}$. Consequently, since $Y_1$ has a direct linear effect on $Y_2$, it holds that

$$\mathcal{H}_{Y_2}^j \neq \mathcal{H}_{Y_2}^{j'} \tag{61}$$
$$\mathcal{H}_{Y_2}^j \circ \gamma \neq \mathcal{H}_{Y_2}^{j'} \circ \gamma \tag{62}$$
$$\tau_{Y_2} \circ \mathcal{L}^i \neq \tau_{Y_2} \circ \mathcal{L}^{i'}, \tag{63}$$

for any intervention $i, i'$ such that $\omega(i) = j$ and $\omega(i') = j'$.

Let now $X_1 \in \Pi_R(Y_1)$ be a relevant concrete variable for $Y_1$, and $t_{11}$ be the non-zero coefficient from $X_1$ to $Y_1$ in the linear abstraction transformation $\mathbf{T}$. We can then build two concrete interventions $i, i'$ by setting all relevant variables of $Y_1$ to zero, except for $X_1$. Formally, the interventions have the following form

$$i = \left(X_1 \leftarrow \frac{k}{t_{11}}, \Pi(Y_1) \setminus \{X_1\} \leftarrow \boldsymbol{0}, \dots\right) \tag{64}$$

$$i' = \left(X_1 \leftarrow \frac{k'}{t_{11}}, \Pi(Y_1) \setminus \{X_1\} \leftarrow \boldsymbol{0}, \dots\right). \tag{65}$$

If we suppose that it does not exist a variable $X_2 \in \Pi_R(Y_2)$ such that $X_1 \xrightarrow{\mathbf{T}} X_2$, all directed paths $X_1 \dashrightarrow X_2$, if any, are mediated by a relevant variable of any abstract variable $Y \in \boldsymbol{Y} \setminus \{Y_2\}$. Consequently, given our construction of $j, j'$ and consequently $i, i'$, any path is mediated by an intervened variable and, therefore, it holds

$$\tau_{Y_2} \circ \mathcal{L}^i = \tau_{Y_2} \circ \mathcal{L}^{i'}, \tag{66}$$

which however breaks interventional consistency and implies that $\mathcal{H}$ is not a **T**-abstraction of $\mathcal{L}$, proving the necessary condition by contradiction. $\quad\square$

## B.7 COROLLARY 3

**Corollary 3 (Connectivity Violation)** Let $\mathcal{H}$ and $\mathcal{L}$ be two linear SCMs respectively on variables $\boldsymbol{Y}$ and $\boldsymbol{X}$ with graphs $\mathcal{G}_{\mathcal{H}}$ and $\mathcal{G}_{\mathcal{L}}$. Consider a linear transformation $\mathbf{T}$ between them leading to the sets of relevant variables $\Pi_R(\boldsymbol{Y})$. If there exists three variables $X_1 \in \Pi_R(Y_1)$, $X_2 \in \Pi_R(Y_2)$, and $X_3 \in \Pi_R(Y_1)$, such that both conditions hold

- $X_1 \xrightarrow{\mathbf{T}} X_2$ in $\mathcal{G}_{\mathcal{L}}$, and
- for any $X_4 \in \Pi_R(Y_2)$, $X_3 \xnrightarrow{\mathbf{T}} X_4$ is not in $\mathcal{G}_{\mathcal{L}}$,

then $\mathcal{H}$ is not a **T**-abstraction of $\mathcal{L}$.

*Proof.* Follows directly from Lemma 2 which applied to the first item implies that $Y_1 \to Y_2$, and from Theorem 1, which applied to the second item implies that $Y_1 \nrightarrow Y_2$, hence providing a contradiction to the assumption that $\mathcal{H}$ is a **T**-abstraction of $\mathcal{L}$. $\quad\square$

## B.8 COROLLARY 4

**Corollary 4 (Exogenous Abstraction)** Let $\mathcal{H} = (\boldsymbol{Y}, \boldsymbol{U}, \boldsymbol{g}, \mathbb{P}_{\boldsymbol{U}})$ be a **T**-abstraction of $\mathcal{L} = (\boldsymbol{X}, \boldsymbol{E}, \boldsymbol{f}, \mathbb{P}_{\boldsymbol{E}})$, where $\mathcal{H}$ and $\mathcal{L}$ are two linear SCMs. Then, the exogenous abstraction function $\gamma \colon \mathcal{D}(\boldsymbol{E}) \to \mathcal{D}(\boldsymbol{U})$, has form

$$\gamma(\boldsymbol{e}) = \mathbf{S}^{\top}\boldsymbol{e}, \tag{67}$$

where $\mathbf{S} = \mathbf{F}\mathbf{T}\mathbf{G}^{-1}$ and $\mathbf{F}, \mathbf{G}$ are the linear transformations of respectively the reduced forms of $\mathcal{L}$ and $\mathcal{H}$, i.e., $\mathcal{L}(\boldsymbol{e}) = \mathbf{F}^T \boldsymbol{e}$ and $\mathcal{H}(\boldsymbol{u}) = \mathbf{G}^T \boldsymbol{u}$.

*Proof.* Since $\mathcal{H}$ **T**-abstracts $\mathcal{L}$, it must hold $\tau \circ \mathcal{L} = \mathcal{H} \circ \gamma$. Consequently, due to the invertibility of the reduced form $\mathcal{H}$ of linear SCMs, it holds that

$$\gamma = \mathcal{H}^{-1} \circ \tau \circ \mathcal{L}. \tag{68}$$

Since, $\mathcal{L}$, $\tau$, and $\mathcal{H}^{-1}$ are linear transformations, their composition coincides with a linear transformation $\mathbf{S} = \mathbf{F}\mathbf{T}\mathbf{G}^{-1}$. $\quad\square$

## B.9   LEMMA 3

**Lemma 3 (Block Composition)**   Let $\mathcal{H}$ be a **T**-abstraction of $\mathcal{L}$, where $\mathcal{H}$ and $\mathcal{L}$ are two linear SCMs respectively on variables $\boldsymbol{Y}$ and $\boldsymbol{X}$. Then, for any abstract variable $Y \in \boldsymbol{Y}$, it holds $X \in \Pi(Y)$ if and only if

- $X \in \Pi_R(Y)$, or
- $X \notin \Pi_R(Y)$, i.e., $X$ is irrelevant, and there exists $X' \in \Pi_R(Y)$ s.t. $X \xrightarrow{\mathbf{T}} X'$.

*Proof.* Let $Y$ be an abstract variable and $j = (\mathrm{Pa}(Y) \leftarrow \boldsymbol{k})$ be a hard intervention fixing all of its endogenous parents. Consequently, the value of the abstract variable, $\mathcal{H}_Y^j(\boldsymbol{u})$ depends only on its exogenous term $U_Y$. Further, given the definition of concrete block, the formulation

$$\mathcal{H}_Y^j(\gamma(\boldsymbol{e})) = \mathcal{H}_Y^j(\mathbf{S}^\top \boldsymbol{e}), \qquad (69)$$

depends only on the exogenous terms $\boldsymbol{e}_{\Pi(Y)}$. Therefore, given the interventional consistency property

$$\mathcal{H}_Y^j(\gamma(\boldsymbol{e})) = \tau_Y(\mathcal{L}_{\Pi_R(Y)}^i(\boldsymbol{e})), \qquad (70)$$

and the lack of cancelling paths, $\mathcal{L}_{\Pi_R(Y)}^i$ also depends only on the exogenous terms $\boldsymbol{e}_{\Pi(Y)}$, for any concrete intervention

$$i = (\Pi_R(\mathrm{Pa}(Y)) \leftarrow \boldsymbol{c}), \qquad (71)$$

where $\tau_{\mathrm{Pa}(Y)}(\boldsymbol{c}) = \boldsymbol{k}$. Notably, given the intervention $i$, the structural mechanisms of $\Pi_R(Y)$ depend only on the exogenous noise of the relevant variables and on those variables whose direct path is non-mediated by another relevant variable. Given Lemma 2, any of such relevant variables must be in the relevant set of a parent, and thus be constrained by the intervention $i$. Consequently, $\mathcal{L}_{\Pi_R(Y)}^i$ depends only on its relevant variables and the *irrelevant* variables with a **T**-direct path towards the former. □

## B.10   LEMMA 4

**Lemma 4 (Disjoint Block)**   Let $\mathcal{H}$ be a **T**-abstraction of $\mathcal{L}$, where $\mathcal{H}$ and $\mathcal{L}$ are two linear SCMs respectively on variables $\boldsymbol{Y}$ and $\boldsymbol{X}$. If for any two distinct endogenous variables $Y_1, Y_2$ it holds that $\Pi(Y_1) \cap \Pi(Y_2) \neq \emptyset$, then the abstract model is not causally sufficient.

*Proof.* By definition of concrete block (Definition 4), each abstract exogenous term $U_Y$ is a function $\gamma$ of the noise terms of the block $\Pi(Y)$. Therefore, given two variables $Y_1, Y_2 \in \boldsymbol{Y}$, we can write

$$U_1 = \gamma_1(E_{\Pi(Y_1)}) \qquad (72)$$
$$U_2 = \gamma_2(E_{\Pi(Y_2)}). \qquad (73)$$

Therefore, whenever the blocks share a subset of variables $\boldsymbol{S} = \Pi(Y_1) \cap \Pi(Y_2)$, both $U_1$ and $U_2$ are a function of the exogenous terms

$$\boldsymbol{V} = \{E_X \in \boldsymbol{E} \mid X \in \boldsymbol{S}\}. \qquad (74)$$

Consequently, the exogenous terms $U_1, U_2$ are not independent and the variables $Y_1, Y_2$ are then confounded. □

## B.11   THEOREM 2

**Theorem 2 (Block Ordering)**   Let $\mathcal{H}$ be a **T**-abstraction of $\mathcal{L}$, where $\mathcal{H}$ and $\mathcal{L}$ are two linear SCMs respectively on variables $\boldsymbol{Y}$ and $\boldsymbol{X}$ with graphs $\mathcal{G}_{\mathcal{H}}$ and $\mathcal{G}_{\mathcal{L}}$. Then, for any valid topological ordering $\prec_{\mathcal{H}}$ of $\mathcal{G}_{\mathcal{H}}$ there exists a valid ordering $\prec_{\mathcal{L}}$ of $\mathcal{G}_{\mathcal{L}}$ such that for any $Y_1, Y_2, Y \in \boldsymbol{Y}$:

- $Y_1 \prec_{\mathcal{H}} Y_2 \iff \Pi(Y_1) \prec_{\mathcal{L}} \Pi(Y_2)$, and
- $\Pi(Y) \prec_{\mathcal{L}} (\boldsymbol{X} \setminus \Pi(\boldsymbol{Y}))$.

*Proof.* Firstly, we recall that in a valid topological order, a variable precedes another only if there is a directed path from the former to the latter [Bondy and Murty, 2008].

$$X_1 \dashrightarrow X_2 \implies X_1 \prec X_2 \qquad (75)$$

Since we always compare abstract variables with abstract variables and concrete variables with concrete variables, in the following we ease the notation by avoiding the subscript on the precedence operator $\prec$.

We show the existence of a valid topological ordering on the concrete model by construction. Given the topological ordering on the abstract model, we assign to each abstract node $Y \in \boldsymbol{Y}$ an integer $\rho_{\boldsymbol{Y}}(Y) \in \{1, \ldots, |\boldsymbol{Y}|\}$ such that

$$Y_1 \prec Y_2 \iff \rho_{\boldsymbol{Y}}(Y_1) < \rho_{\boldsymbol{Y}}(Y_2). \qquad (76)$$

Then, we can take any valid topological ordering within any concrete block $\Pi(Y)$ and assign in the same way $\rho_{\Pi(Y)}(X)$ for any $Y \in \boldsymbol{Y}$ and $X \in \Pi(Y)$. We do the same for the set $\boldsymbol{Q}$ of concrete variables outside of any block, which we formally define as follows

$$\boldsymbol{Q} = \boldsymbol{X} \setminus \bigcup_{Y \in \boldsymbol{Y}} \Pi(Y). \qquad (77)$$

We then assign the "position" of each concrete variable $X \in \boldsymbol{X}$ through a further integer defined as follows,

$$\rho_{\boldsymbol{X}} = \begin{cases} \sum_{Y' \prec Y} |\Pi(Y')| + \rho_{\Pi(Y)}(X) & \exists Y. X \in \Pi(X) \\ \sum_{Y \in \boldsymbol{Y}} |\Pi(Y)| + \rho_{\boldsymbol{Q}}(X) & X \in \boldsymbol{Q}. \end{cases} \qquad (78)$$

Notably, since the blocks do not overlap (Lemma 4), the assignment is unique. We finally define the concrete topological ordering for any $X_1, X_2 \in \boldsymbol{X}$ as

$$X_1 \prec X_2 \iff \rho_{\boldsymbol{X}}(X_1) < \rho_{\boldsymbol{X}}(X_2). \qquad (79)$$

Given this ordering, it holds by construction that

$$\forall Y_1, Y_2 \in \boldsymbol{Y}. Y_1 \prec_{\mathcal{H}} Y_2 \iff \Pi(Y_1) \prec_{\mathcal{L}} \Pi(Y_2) \quad (80)$$

$$\forall Y \in \boldsymbol{Y}. \Pi(Y) \prec_{\mathcal{L}} \{X \in \boldsymbol{X} \mid X \notin \bigcup_{Y \in \boldsymbol{Y}} \Pi(Y)\}. \quad (81)$$

Therefore, to finally prove the Theorem we have to show that the ordering we defined is valid for the concrete graph. Formally, we have to show that, for any $X_1, X_2 \in \boldsymbol{X}$,

$$X_1 \to X_2 \implies X_1 \prec X_2 \quad (82)$$
$$\implies \rho_{\boldsymbol{X}}(X_1) < \rho_{\boldsymbol{X}}(X_2). \quad (83)$$

*Case* $\{X_1, X_2\} \subset \Pi(Y) \vee \{X_1, X_2\} \subset \boldsymbol{Q}$. Whenever $X_1 \to X_2$ and $X_1, X_2$ are in the same block $\Pi(Y)$ for some $Y \in \boldsymbol{Y}$ or are both in $\boldsymbol{Q}$, then $\rho_{\boldsymbol{X}}(X_1) < \rho_{\boldsymbol{X}}(X_2)$ by definition.
*Case* $X_1 \in \Pi(Y_1), X_2 \in \boldsymbol{Q}$. Also holds by definition.
*Case* $X_1 \in \boldsymbol{Q}, X_2 \in \Pi(Y)$. By definition of block, this case never occurs, since otherwise $X_1$ would be in $\Pi$ (Lemma 3).
*Case* $X_1 \in \Pi(Y_1), X_2 \in \Pi(Y_2)$. Further, whenever $X_1 \to X_2$ such that $X_1 \in \Pi(Y_1)$ for some $Y_1$ and $X_2 \in \Pi(Y_2)$ for some $Y_2$, then $X_1$ is relevant, otherwise it would have also been in the block $\Pi(Y_2)$, which are necessarily disjoint (Lemma 4). Therefore, given the sufficient condition on the existence of an abstract edge (Lemma 2), it must hold

$$Y_1 \to Y_2 \quad (84)$$
$$\implies Y_1 \prec Y_2 \quad (85)$$
$$\implies \Pi(Y_1) \prec \Pi(Y_2) \quad (86)$$
$$\implies X_1 \prec X_2. \quad (87)$$

$\square$

## B.12 LEMMA 5

**Lemma 5 (Submodel Abstraction)** Let $\mathcal{H}$ and $\mathcal{L}$ be two linear SCMs respectively on variables $\boldsymbol{Y}$ and $\boldsymbol{X}$. Then, $\mathcal{H}$ is a $\mathbf{T}$-abstraction of $\mathcal{L}$ if and only if $\mathcal{H}$ is a $\mathbf{T}$-abstraction of $\mathcal{L}'$, where $\mathcal{L}'$ is a submodel of $\mathcal{L}$ defined on the subset of variables $\boldsymbol{X}' = \Pi(\boldsymbol{Y})$, i.e., all of the variables in the concrete blocks.

*Proof.* The Lemma directly follows from Theorem 2, where the variables not in any block always follow in the topological ordering the remaining. Therefore, by removing them, for any intervention $i$ the interventional consistency $\tau \circ \mathcal{L}'^i = \tau \circ \mathcal{L}^i$ still holds since they do not influence any relevant variable, hence the abstraction function $\tau$, nor any block, hence the exogenous abstraction function $\gamma$. Similarly, we could add as many variables and mechanism not influencing the blocks and interventional consistency would still hold. $\square$

## B.13 LEMMA 6

**Lemma 6 (Exogenous Abstraction)** Let $\mathcal{H} = (\boldsymbol{Y}, \boldsymbol{U}, \mathbf{M}, \mathbb{P}_{\boldsymbol{U}})$ and $\mathcal{L} = (\boldsymbol{X}, \boldsymbol{E}, \mathbf{W}, \mathbb{P}_{\boldsymbol{E}})$ be two linear SCMs such that $\mathcal{H}$ is a $\mathbf{T}$-abstraction of $\mathcal{L}$, such that $\mathbf{W}$ follows permutation $\pi_{\mathcal{H}}$. Then, the exogenous abstraction function $\gamma \colon \mathcal{D}(\boldsymbol{E}) \to \mathcal{D}(\boldsymbol{U})$ is unique and has form $\gamma(\boldsymbol{e}) = \mathbf{S}^\top \boldsymbol{e}$ for a linear transformation $\mathbf{S} \in \mathbb{R}^{d \times b}$ defined as the upper-diagonal block matrix

$$\mathbf{S} = \begin{bmatrix} \boldsymbol{s}_1 & \boldsymbol{0} & \cdots & \boldsymbol{0} \\ \boldsymbol{0} & \boldsymbol{s}_2 & \cdots & \boldsymbol{0} \\ \vdots & \vdots & \ddots & \vdots \\ \boldsymbol{0} & \boldsymbol{0} & \cdots & \boldsymbol{s}_{b,} \end{bmatrix} \quad (88)$$

where $\boldsymbol{s}_k = \mathbf{F}_{kk} \boldsymbol{t}_k = (\mathbf{I} - \mathbf{W}_{kk})^{-1} \boldsymbol{t}_k$ for any $Y_k \in \boldsymbol{Y}$.

*Proof.* Given the definition of $\mathbf{T}$-abstraction, we can rephrase observational consistency as

$$\tau \circ \mathcal{L} = \mathcal{H} \circ \gamma \quad (89)$$
$$\mathbf{F}\mathbf{T} = \mathbf{S}\mathbf{G} \quad (90)$$

where $\mathbf{F}$ and $\mathbf{G}$ are respectively the reduced forms of the concrete and the abstract SCM. Consequently, by exploiting the block-definition of $\mathbf{T}$, we can reformulate the left side of the equation as

$$\begin{bmatrix} \mathbf{F}_{11} & \mathbf{F}_{12} & \cdots & \mathbf{F}_{1b} \\ \boldsymbol{0} & \mathbf{F}_{22} & \cdots & \mathbf{F}_{2b} \\ \vdots & \vdots & \ddots & \vdots \\ \boldsymbol{0} & \boldsymbol{0} & \cdots & \mathbf{F}_{bb} \end{bmatrix} \begin{bmatrix} \boldsymbol{t}_1 & \boldsymbol{0} & \cdots & \boldsymbol{0} \\ \boldsymbol{0} & \boldsymbol{t}_2 & \cdots & \boldsymbol{0} \\ \vdots & \vdots & \ddots & \vdots \\ \boldsymbol{0} & \boldsymbol{0} & \cdots & \boldsymbol{t}_b \end{bmatrix} \quad (91)$$

$$= \begin{bmatrix} \mathbf{F}_{11}\boldsymbol{t}_1 & \mathbf{F}_{12}\boldsymbol{t}_2 & \cdots & \mathbf{F}_{1b}\boldsymbol{t}_b \\ \boldsymbol{0} & \mathbf{F}_{22}\boldsymbol{t}_2 & \cdots & \mathbf{F}_{2b}\boldsymbol{t}_b \\ \vdots & \vdots & \ddots & \vdots \\ \boldsymbol{0} & \boldsymbol{0} & \cdots & \mathbf{F}_{bb}\boldsymbol{t}_b \end{bmatrix} \quad (92)$$

Given that block variables are not shared (Lemma 4) and follow the same topological order of $\mathbf{T}$, the exogenous transformation must also have form

$$\mathbf{S} = \begin{bmatrix} \boldsymbol{s}_1 & \boldsymbol{0} & \cdots & \boldsymbol{0} \\ \boldsymbol{0} & \boldsymbol{s}_2 & \cdots & \boldsymbol{0} \\ \vdots & \vdots & \ddots & \vdots \\ \boldsymbol{0} & \boldsymbol{0} & \cdots & \boldsymbol{s}_{b,} \end{bmatrix}. \quad (93)$$

We can therefore reformulate the right side $\mathbf{S}\mathbf{G}$ of the ob-

servational consistency equation as

$$\begin{bmatrix} \boldsymbol{s}_1 & \boldsymbol{0} & \cdots & \boldsymbol{0} \\ \boldsymbol{0} & \boldsymbol{s}_2 & \cdots & \boldsymbol{0} \\ \vdots & \vdots & \ddots & \vdots \\ \boldsymbol{0} & \boldsymbol{0} & \cdots & \boldsymbol{s}_{b,} \end{bmatrix} \begin{bmatrix} 1 & g_{12} & \cdots & g_{1b} \\ 0 & 1 & \cdots & g_{2b} \\ \vdots & \vdots & \ddots & \vdots \\ 0 & 0 & \cdots & 1 \end{bmatrix} \quad (94)$$

$$= \begin{bmatrix} \boldsymbol{s}_1 & g_{12}\boldsymbol{s}_1 & \cdots & g_{1b}\boldsymbol{s}_1 \\ \boldsymbol{0} & \boldsymbol{s}_2 & \cdots & g_{2b}\boldsymbol{s}_2 \\ \vdots & \vdots & \ddots & \vdots \\ \boldsymbol{0} & \boldsymbol{0} & \cdots & \boldsymbol{s}_{b.} \end{bmatrix} \quad (95)$$

Consequently, for any $Y_i \in \boldsymbol{Y}$, it holds $\boldsymbol{s}_i = \mathbf{F}_{ii}\boldsymbol{t}_i$.  $\square$

## B.14  THEOREM 3

**Theorem 3 (Block Abstraction)** Let $\mathcal{H} = (\boldsymbol{Y}, \boldsymbol{U}, \mathbf{M}, \mathbb{P}_{\boldsymbol{U}})$ and $\mathcal{L} = (\boldsymbol{X}, \boldsymbol{E}, \mathbf{W}, \mathbb{P}_{\boldsymbol{E}})$ be two linear SCMs with graphs $\mathcal{G}_{\mathcal{H}}$ and $\mathcal{G}_{\mathcal{L}}$ respectively. Then $\mathcal{H}$ is a linear $\mathbf{T}$-abstraction of $\mathcal{L}$ if and only if for any valid topological ordering $\prec_{\mathcal{H}}$ of $\mathcal{G}_{\mathcal{H}}$ there exists a valid ordering $\prec_{\mathcal{L}}$ of $\mathcal{G}_{\mathcal{L}}$ such that, for any $Y_i, Y_j \in \boldsymbol{Y}$ it holds

$$Y_i \prec_{\mathcal{H}} Y_j \iff \Pi(Y_i) \prec_{\mathcal{L}} \Pi(Y_j), \text{ and} \quad (96)$$
$$\mathbf{W}_{ij}\boldsymbol{s}_j = m_{ij}\boldsymbol{t}_i, \quad (97)$$

where $\mathbf{W}_{ij}$ is the $i$-th element on the $j$-th column of $\mathbf{W}$, and $m_{ij}$ is the $i$-th element on the $j$-th column of $\mathbf{M}$.

*Proof.* Firstly, we introduce the following decomposition of the reduced forms of the concrete and the abstract model, which we separately prove in Appendix B.15.

$$\mathbf{F}_{ij} = \begin{cases} (\mathbf{I} - \mathbf{W}_{ii})^{-1} & \text{if } i = j \\ \mathbf{F}_{ii}(\mathbf{W}_{ij} + \mathbf{R}_{ij})\mathbf{F}_{jj} & \text{if } i < j \\ \mathbf{0} & \text{otherwise,} \end{cases} \quad (98)$$

$$\mathbf{R}_{ij} = \sum_{i<k<j} \mathbf{W}_{ik}\mathbf{F}_{kk}(\mathbf{W}_{kj} + \mathbf{R}_{kj}) \quad (99)$$

$$g_{ij} = \begin{cases} 1 & \text{if } i = j \\ m_{ij} + \rho_{ij} & \text{if } i < j \\ 0 & \text{otherwise,} \end{cases} \quad (100)$$

$$\rho_{ij} = \sum_{i<k<j} m_{ik}(m_{kj} + \rho_{kj}) \quad (101)$$

*Necessary Condition.* We show that $\mathbf{T}$-abstraction implies both conditions. For the existence of a valid concrete ordering, we invite the reader to consult the proof of Theorem 2. Therefore, we focus on proving that $\mathbf{T}$-abstraction entails $\mathbf{W}_{ij}\boldsymbol{s}_j = m_{ij}\boldsymbol{t}_i$ for any $Y_i, Y_j \in \boldsymbol{Y}$. Given the decomposition consistency condition $\mathbf{FT} = \mathbf{SG}$ from the proof of

Lemma 6, for each $i < j$, it must hold that

$$\mathbf{F}_{ij}\boldsymbol{t}_j = \boldsymbol{s}_i g_{ij} \quad (102)$$
$$\mathbf{F}_{ii}(\mathbf{W}_{ij} + \mathbf{R}_{ij})\mathbf{F}_{jj}\boldsymbol{t}_j = \mathbf{F}_{ii}\boldsymbol{t}_i(m_{ij} + \rho_{ij}) \quad (103)$$
$$(\mathbf{W}_{ij} + \mathbf{R}_{ij})\boldsymbol{s}_j = \boldsymbol{t}_i(m_{ij} + \rho_{ij}) \quad (104)$$
$$\mathbf{W}_{ij}\boldsymbol{s}_j = m_{ij}\boldsymbol{t}_i, \quad (105)$$

where the first step comes from the previously introduced decomposition, proved in Appendix B.15. To prove the last step we firstly notice that

$$\mathbf{R}_{ij}\boldsymbol{s}_j = \rho_{ij}\boldsymbol{t}_i \iff \mathbf{W}_{ij}\boldsymbol{s}_j = m_{ij}\boldsymbol{t}_i. \quad (106)$$

We then prove the statement for each row by induction on the columns. We take $j = i + 1$ as base case, where it holds

$$\mathbf{R}_{ij}\boldsymbol{s}_j = \rho_{ij}\boldsymbol{t}_i \quad (107)$$
$$\mathbf{0}\boldsymbol{s}_j = 0 \cdot \boldsymbol{t}_i \quad (108)$$
$$\mathbf{0} = \mathbf{0} \quad (109)$$
$$\implies \mathbf{W}_{ij}\boldsymbol{s}_j = m_{ij}\boldsymbol{t}_i. \quad (110)$$

Consequently, we can show that

$$\mathbf{R}_{ij}\boldsymbol{s}_j = \sum_{i<k<j} \mathbf{W}_{ik}\mathbf{F}_{kk}(\mathbf{W}_{kj} + \mathbf{R}_{kj})\boldsymbol{s}_j \quad (111)$$
$$= \sum_{i<k<j} \mathbf{W}_{ik}\mathbf{F}_{kk}\mathbf{W}_{kj}\boldsymbol{s}_j + \mathbf{W}_{ik}\mathbf{F}_{kk}\mathbf{R}_{kj}\boldsymbol{s}_j \quad (112)$$
$$= \sum_{i<k<j} \mathbf{W}_{ik}\mathbf{F}_{kk}\mathbf{W}_{kj}\boldsymbol{s}_j + \mathbf{W}_{ik}\mathbf{F}_{kk}\rho_{kj}\boldsymbol{t}_k \quad (113)$$
$$= \sum_{i<k<j} \mathbf{W}_{ik}\mathbf{F}_{kk}m_{kj}\boldsymbol{t}_k + \mathbf{W}_{ik}\mathbf{F}_{kk}\rho_{kj}\boldsymbol{t}_k \quad (114)$$
$$= \sum_{i<k<j} \mathbf{W}_{ik}\mathbf{F}_{kk}\boldsymbol{t}_k m_{kj} + \mathbf{W}_{ik}\mathbf{F}_{kk}\boldsymbol{t}_k\rho_{kj} \quad (115)$$
$$= \sum_{i<k<j} m_{ik}\boldsymbol{t}_i m_{kj} + m_{ik}\boldsymbol{t}_i\rho_{kj} \quad (116)$$
$$= \sum_{i<k<j} m_{ik}(m_{kj} + \rho_{kj})\boldsymbol{t}_i \quad (117)$$
$$= \rho_{ij}\boldsymbol{t}_i. \quad (118)$$

*Sufficient Condition.* We now show that the conditions imply interventional consistency of the abstraction. That is, we want to prove that

$$\tau_Y \circ \mathcal{L}^{\iota}_{\Pi(Y)} = \mathcal{H}^{\omega(\iota)}_Y \circ \gamma, \quad (119)$$

for any concrete intervention $\iota$ on the relevant sets defined by the linear abstraction transformation $\mathbf{T}$. Firstly, we notice that the equation is immediately true for any abstract variable $Y \in \boldsymbol{Y}$ whenever the intervention targets its relevant set. Therefore, we focus on the case where the abstract intervention $\omega(\iota)$ does not affect $Y$. Consequently, given that we assume that the topological ordering of the blocks coincides

with that of the abstract variables, we can decompose the concrete model as

$$\mathcal{L}^{\iota}_{\Pi(Y_j)}(\boldsymbol{e}) = \sum_{Y_i \in \mathrm{Pa}(Y_j)} \left( \left[ \mathcal{L}^{\iota}_{\Pi(Y_i)}(\boldsymbol{e}) \right]^{\top} \mathbf{W}_{ij} + \boldsymbol{e}^{\top}_{\Pi(Y_j)} \right) \mathbf{F}_{jj}, \tag{120}$$

where we (i.) compute the linear contribution of the parents, (ii.) sum the exogenos noise of the block, (iii.) and apply the submodel composed of the internal connections in the block. Similarly, we can decompose the abstract model as

$$\mathcal{H}^{\omega(\iota)}_{Y_j}(\boldsymbol{u}) = \sum_{Y_i \in \mathrm{Pa}(Y_j)} \mathcal{H}^{\omega(\iota)}_{Y_i}(\boldsymbol{u}) \cdot m_{ij} + u_j. \tag{121}$$

Abstraction holds whenever interventional consistency is satisfied by at least an exogenous transformation $\gamma$. To continue the proof, we then define it as the linear transformation from Lemma 6, where $\boldsymbol{s}_j = \mathbf{F}_{jj} \boldsymbol{t}_j$ for any $Y_j \in \boldsymbol{Y}$. Therefore, we can reformulate interventional consistency as

$$\sum_{Y_i \in \mathrm{Pa}(Y_j)} \left( \left[ \mathcal{L}^{\iota}_{\Pi(Y_i)}(\boldsymbol{e}) \right]^{\top} \mathbf{W}_{ij} + \boldsymbol{e}^{\top}_{\Pi(Y_j)} \right) \mathbf{F}_{jj} \boldsymbol{t}_j$$
$$= \sum_{Y_i \in \mathrm{Pa}(Y_j)} \mathcal{H}^{\omega(\iota)}_{Y_i}(\mathbf{S}^{\top} \boldsymbol{e}) \cdot m_{ij} + \boldsymbol{e}^{\top}_{\Pi(Y_j)} \boldsymbol{s}_j, \tag{122}$$

which further simplifies to

$$\sum_{Y_i \in \mathrm{Pa}(Y_j)} \left[ \mathcal{L}^{\iota}_{\Pi(Y_i)}(\boldsymbol{e}) \right]^{\top} \mathbf{W}_{ij} \mathbf{F}_{jj} \boldsymbol{t}_j$$
$$= \sum_{Y_i \in \mathrm{Pa}(Y_j)} \mathcal{H}^{\omega(\iota)}_{Y_i}(\mathbf{S}^{\top} \boldsymbol{e}) \cdot m_{ij} \tag{123}$$

given our choice of the exogenous transformation $\mathbf{S}$. We prove this last equation by induction on the topological ordering of the abstract graph. In fact, as a base case, for any root of the graph the equation holds given that the parent set is the empty set. Consequently, we can finally show that $\mathbf{W}_{ij} \boldsymbol{s}_j = m_{ij} \boldsymbol{t}_i$ implies abstraction as follows

$$\sum_{Y_i \in \mathrm{Pa}(Y_j)} \mathcal{H}^{\omega(\iota)}_{Y_i}(\mathbf{S}^{\top} \boldsymbol{e}) \cdot m_{ij} \tag{124}$$

$$= \sum_{Y_i \in \mathrm{Pa}(Y_j)} \left[ \mathcal{L}^{\iota}_{\Pi(Y_i)}(\boldsymbol{e}) \right]^{\top} \boldsymbol{t}_i \cdot m_{ij} \tag{125}$$

$$= \sum_{Y_i \in \mathrm{Pa}(Y_j)} \left[ \mathcal{L}^{\iota}_{\Pi(Y_i)}(\boldsymbol{e}) \right]^{\top} \mathbf{W}_{ij} \boldsymbol{s}_j \tag{126}$$

$$= \sum_{Y_i \in \mathrm{Pa}(Y_j)} \left[ \mathcal{L}^{\iota}_{\Pi(Y_i)}(\boldsymbol{e}) \right]^{\top} \mathbf{W}_{ij} \mathbf{F}_{jj} \boldsymbol{t}_j. \tag{127}$$

$\square$

## B.15 MODEL REDUCTION DECOMPOSITION

In the following, we prove the decomposition of the model reduction matrix $\mathbf{F}$ from the proof in Appendix B.14. To simplifiy the notation, we define the matrix $\mathbf{A} = (\mathbf{I} - \mathbf{W})$.

*Proof.* Back-substituting to solve $\mathbf{FA} = \mathbf{I}$ leads to

$$\mathbf{F}_{ij} = \begin{cases} \mathbf{A}_{ii}^{-1} & i = j \\ -\sum_{i < k \leq j} \mathbf{F}_{ii} \mathbf{A}_{ik} \mathbf{F}_{kj} & i < j \\ 0 & i > j \end{cases} \tag{128}$$

Therefore, we want to prove that whenever $i < j$, it holds

$$-\sum_{i < k \leq j} \mathbf{F}_{ii} \mathbf{A}_{ik} \mathbf{F}_{kj} = \mathbf{F}_{ii}(\mathbf{W}_{ij} + \mathbf{R}_{ij}) \mathbf{F}_{jj}, \tag{129}$$

where

$$\mathbf{R}_{ij} = \sum_{i < k < j} \mathbf{W}_{ik} \mathbf{F}_{kk}(\mathbf{W}_{kj} + \mathbf{R}_{kj}). \tag{130}$$

Overall, we simplify the thesis as follows

$$\mathbf{F}_{ii}(\mathbf{W}_{ij} + \mathbf{R}_{ij}) \mathbf{F}_{jj} = -\sum_{i < k \leq j} \mathbf{F}_{ii} \mathbf{A}_{ik} \mathbf{F}_{kj} \tag{131}$$

$$\mathbf{F}_{ii}(\mathbf{W}_{ij} + \mathbf{R}_{ij}) \mathbf{F}_{jj} = \sum_{i < k \leq j} \mathbf{F}_{ii} \mathbf{W}_{ik} \mathbf{F}_{kj} \tag{132}$$

$$\mathbf{F}_{ii} \mathbf{R}_{ij} \mathbf{F}_{jj} = \sum_{i < k < j} \mathbf{F}_{ii} \mathbf{W}_{ik} \mathbf{F}_{kj} \tag{133}$$

$$\mathbf{R}_{ij} \mathbf{F}_{jj} = \sum_{i < k < j} \mathbf{W}_{ik} \mathbf{F}_{kj}. \tag{134}$$

We finally prove our thesis by induction on the decreasing row component $i$, starting from $i = j - 1$. In the base case, both sides of the equation reduce to zero and thus the statement holds. We then prove the inductive case by showing that if the statement holds for any $k > i$, then it also holds for $i$. Formally,

$$\sum_{i < k < j} \mathbf{W}_{ik} \mathbf{F}_{kj} \tag{135}$$

$$= \sum_{i < k < j} \mathbf{W}_{ik} \mathbf{F}_{kk}(\mathbf{W}_{kj} + \mathbf{R}_{kj}) \mathbf{F}_{jj} \tag{136}$$

$$= \mathbf{R}_{ij} \mathbf{F}_{jj}. \tag{137}$$

$\square$

## C DATASET

In the following, we report further details on the simulation procedure used to generate the dataset for the experiments, which we also visualize in Figure 4.

**Abstract Model.** Given a number of desired nodes and edges, we sample the abstract model by randomly sampling an Erdős-Rényi graph for the given parameters. Then, we sample the weights of the edges from the uniform distribution in the interval $[-2, -0.5] \cup [0.5, 2]$.

**Abstraction Function.** Given the abstract model, we sample the abstraction function by firstly assigning a block size to each node from the uniform distribution, whose minimum and maximum values are given as input. Then, within each block we randomly choose at least half of the nodes to be *relevant* and randomly assign the remaining as relevant or not. We also sample a further block to contain the *ignored* variables, for which the abstraction function maps to zero. We finally sample the abstraction coefficients from the uniform distribution in the interval $[-2, -0.5] \cup [0.5, 2]$.

**Concrete Model.** Given an abstract model and an abstraction function, we sample the concrete model using the algorithm in Algorithm 1. Firstly, we sample the causal relations within each block by randomly sampling an upper triangular matrix with non-zero entries from the standard normal distribution. Then, we employ the Dirichlet distribution to sample each vector $v$ with sum one as requested by the algorithm to explore the right-inverses of the exogenous abstraction function. Finally, we randomly sample from the standard normal distribution. the weights to connect ignored variables.

**Data Generation.** As we detailed in the main body, we sample the data from the concrete model by first sampling the non-Gaussian noise and then by abstracting the noise to sample from the abstract model. In all experiments, we use the Exponential distribution. We then normalize the data to have zero mean and unit variance and permute all the variables in both the concrete and abstract samples.

# D  ADDITIVE NOISE ON ABSTRACT OBSERVATIONS

In this section, we discuss strategies to handle a further scenario where we consider abstract observations to be further perturbed by random noise. We consider the following generative model for the abstract observations:

$$e^{(i)} \sim \text{Exponential} \qquad \text{for } i = 1, \dots, |\mathcal{D}_\mathcal{L}|, \quad (138)$$

$$x^{(i)} = \mathcal{L}(e^{(i)}) \qquad \text{for } i = 1, \dots, |\mathcal{D}_\mathcal{L}|, \quad (139)$$

$$y^{(i)} = \mathcal{H}(\gamma(e^{(i)})) + \epsilon^{(i)} \quad \text{for } i = 1, \dots, |\mathcal{D}_J|, \quad (140)$$

where $\epsilon \sim \mathcal{N}(0, \sigma^2)$ is a Gaussian noise term and the data-generating process is the same of Section 4.1. Due to the presence of noise, minimizing the least-squares error does not ensure to recover the true abstraction function. We thus propose two strategies to identify the concrete blocks of each abstract variable. By exploiting the fact that each concrete

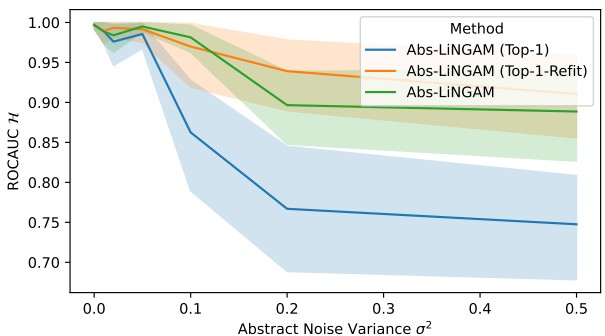

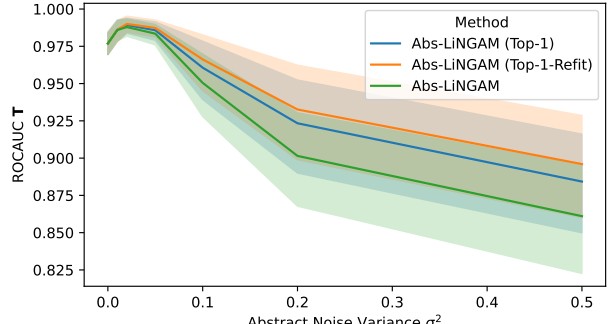

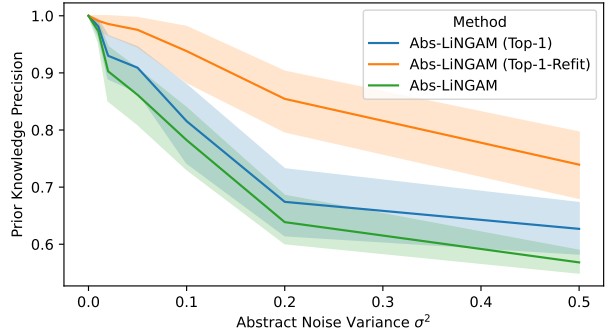

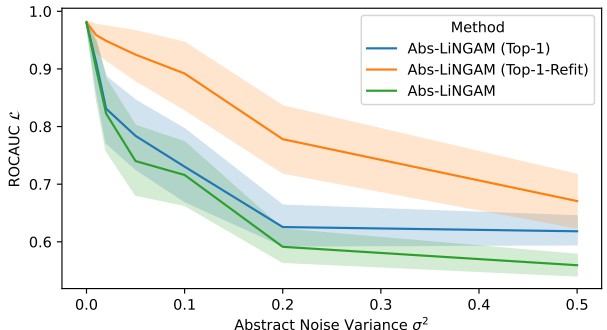

Figure 3: Results of Abs-LiNGAM over pairs of abstract ($b = 5$ nodes) and concrete ($d \in [25, 50]$ nodes) linear SCMs after perturbing the abstract observations with normal noise of increasing variance $\sigma^2$. We denote as "Top-1" the strategy where we force the selection of at most a single abstract variable per concrete one and as "Top-1-Refit" the one where we then refit each abstraction vector. All results are averaged over 30 independent runs with $|\mathcal{D}_\mathcal{L}| = 20000$ concrete samples and $|\mathcal{D}_J| = 150$ paired samples.

variable pertains to a single abstract variable, we can filter the resulting matrix $\hat{\mathbf{T}}$ to select only the largest component per row if it is above the threshold. We find then beneficial to refit the model once we have identified the block in this way, as in

$$\boldsymbol{t}_i = \arg\min_{\boldsymbol{t}_i} \left\| \boldsymbol{x}_{\Pi_R Y_i} - \boldsymbol{t}_i^\top \boldsymbol{y}_i \right\|_2^2. \qquad (141)$$

In Figure 3, we report results for the reconstruction of the blocks from the paired samples for increasing variance $\sigma^2$ of the noise term for these strategies.

# E  ADDITIONAL RESULTS

In this section, we report additional results on our experiments on Abs-LiNGAM (Algorithm 2). We mostly consider three settings: *small*, where the number of nodes in the abstract model is $b = 5$ and the number of nodes in the concrete model is $d \in [25, 50]$; *medium*, where the number of nodes in the abstract model is $b = 10$ and the number of nodes in the concrete model is $d \in [50, 100]$; and *large*, where the number of nodes in the abstract model is $b = 10$ and the number of nodes in the concrete model is $d \in [100, 150]$. We then report results on the sensitivity of Abs-LiNGAM to the number of paired samples $\mathcal{D}_J$ (Figures 5 to 7), the number of concrete samples $\mathcal{D}_{\mathcal{L}}$ (Figures 8 to 10), and the number of nodes in the concrete model $d$ (Figures 11 to 13). Further, we report results on the quality of the retrieved prior knowledge given the threshold used to mask the learned abstraction function $\hat{\mathbf{T}}$ (Figure 14) and the threshold used to mask the learned abstract model $\hat{\mathcal{H}}$ (Figure 15). Similarly, we study the retrieval of the prior knowledge for different number of bootstrap samples to identify the abstract model $\hat{\mathcal{H}}$ (Figure 16). To provide further insights on the performance of Abs-LiNGAM, we also report precision and recall on the three settings (Tables 1 to 3). We finally report additional results on the reconstruction of the abstraction function $\hat{\mathbf{T}}$ in the small (Figure 17), medium (Figure 18), and large (Figure 19) settings.

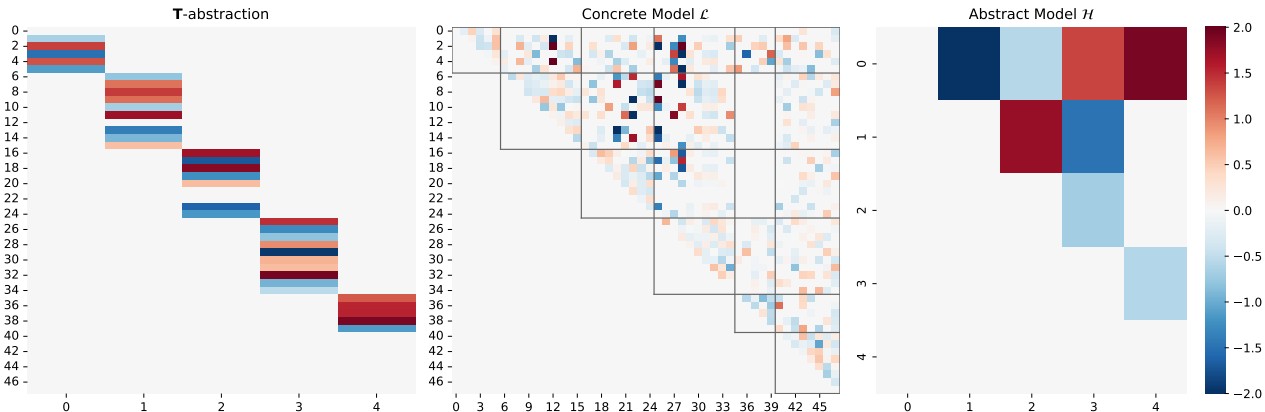

Figure 4: Visualization of a pair of concrete-abstract models and their abstraction function. The abstract graph has 5 nodes and 8 edges while the concrete has 5 blocks of random size from $[5, 10]$, with an additional block for the ignored variables.

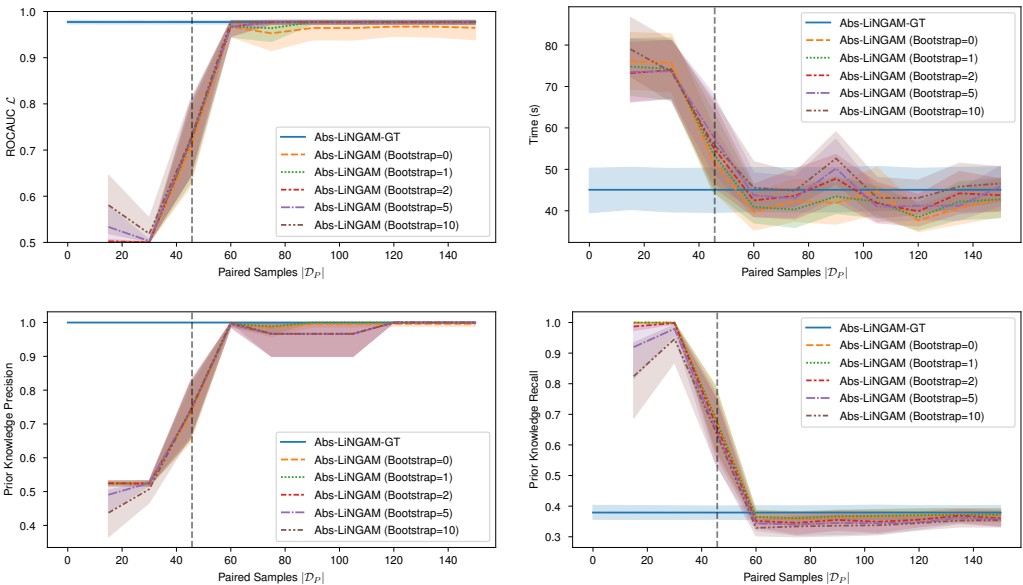

Figure 5: Results of Abs-LiNGAM over pairs of abstract ($b = 5$ nodes) and concrete ($d \in [25, 50]$ nodes) linear SCMs. In all subfigures we plot the results for an increasing number of paired samples $\mathcal{D}_J$ and we report the average size of the concrete graphs as a vertical dashed line. Abs-LiNGAM-GT denotes a ground truth oracle where the abstraction function and the abstract model are given. The first plot (top left) shows the ROC-AUC of the retrieved concrete causal model $\hat{\mathcal{L}}$. The second plot (top right) shows the execution time required to retrieve the concrete causal model. The third and fourth plots (bottom) show the precision and recall of the prior knowledge inferred by the learned abstraction function $\hat{\mathbf{T}}$ and the consequent abstract model $\hat{\mathcal{H}}$. All results are averaged over 30 independent runs with $|\mathcal{D}_{\mathcal{L}}| = 15000$ concrete samples.

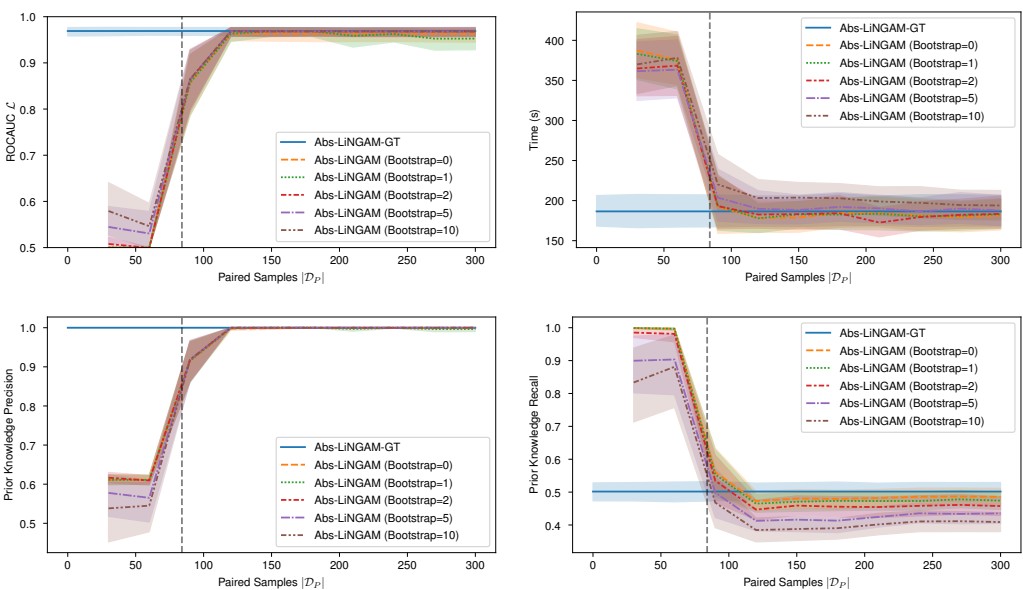

Figure 6: Results of Abs-LiNGAM over pairs of abstract ($b = 10$ nodes) and concrete ($d \in [50, 100]$ nodes) linear SCMs. In all subfigures we plot the results for an increasing number of paired samples $\mathcal{D}_J$ and we report the average size of the concrete graphs as a vertical dashed line. Abs-LiNGAM-GT denotes a ground truth oracle where the abstraction function and the abstract model are given. The first plot (top left) shows the ROC-AUC of the retrieved concrete causal model $\hat{\mathcal{L}}$. The second plot (top right) shows the execution time required to retrieve the concrete causal model. The third and fourth plots (bottom) show the precision and recall of the prior knowledge inferred by the learned abstraction function $\hat{\mathbf{T}}$ and the consequent abstract model $\hat{\mathcal{H}}$. All results are averaged over 30 independent runs with $|\mathcal{D}_{\mathcal{L}}| = 15000$ concrete samples.

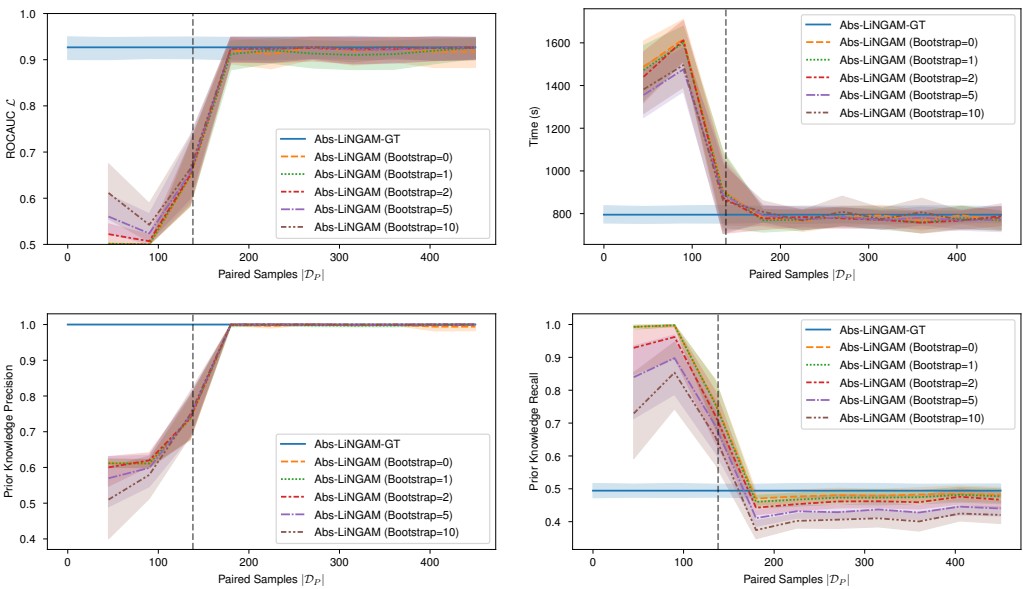

Figure 7: Results of Abs-LiNGAM over pairs of abstract ($b = 10$ nodes) and concrete ($d \in [100, 150]$ nodes) linear SCMs. In all subfigures we plot the results for an increasing number of paired samples $\mathcal{D}_J$ and we report the average size of the concrete graphs as a vertical dashed line. Abs-LiNGAM-GT denotes a ground truth oracle where the abstraction function and the abstract model are given. The first plot (top left) shows the ROC-AUC of the retrieved concrete causal model $\hat{\mathcal{L}}$. The second plot (top right) shows the execution time required to retrieve the concrete causal model. The third and fourth plots (bottom) show the precision and recall of the prior knowledge inferred by the learned abstraction function $\hat{\mathbf{T}}$ and the consequent abstract model $\hat{\mathcal{H}}$. All results are averaged over 30 independent runs with $|\mathcal{D}_{\mathcal{L}}| = 15000$ concrete samples.

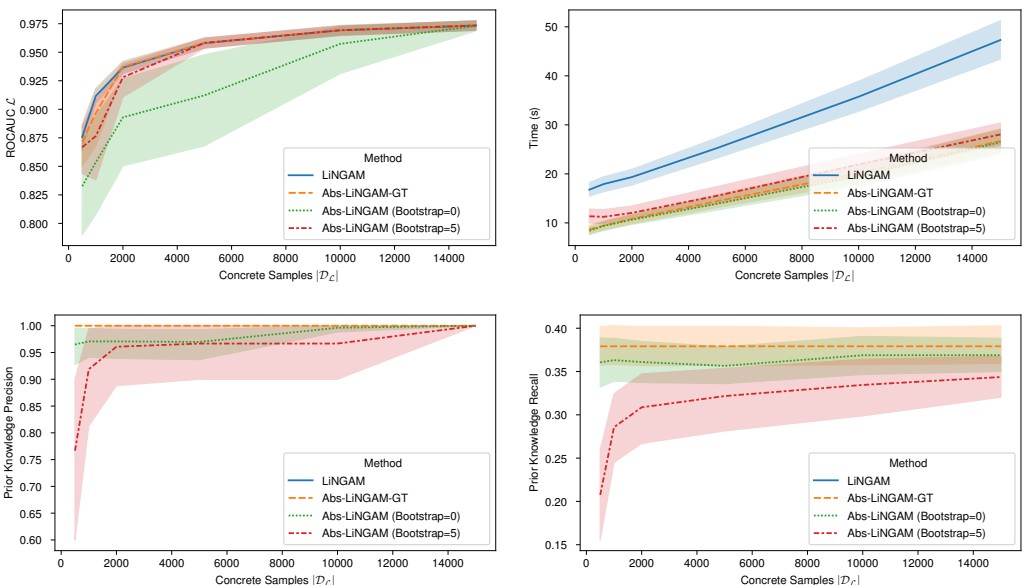

Figure 8: Results of Abs-LiNGAM over pairs of abstract ($b = 5$ nodes) and concrete ($d \in [25, 50]$ nodes) linear SCMs. In all subfigures we plot the results for an increasing number of concrete samples $|\mathcal{D}_\mathcal{L}|$. Abs-LiNGAM-GT denotes a ground truth oracle where the abstraction function and the abstract model are given. The first plot (top left) shows the ROC-AUC of the retrieved concrete causal model $\hat{\mathcal{L}}$. The second plot (top right) shows the execution time required to retrieve the concrete causal model. The third and fourth plots (bottom) show the precision and recall of the prior knowledge inferred by the learned abstraction function $\hat{\mathbf{T}}$ and the consequent abstract model $\hat{\mathcal{H}}$. All results are averaged over 30 independent runs with $|\mathcal{D}_J| = 100$ paired samples.

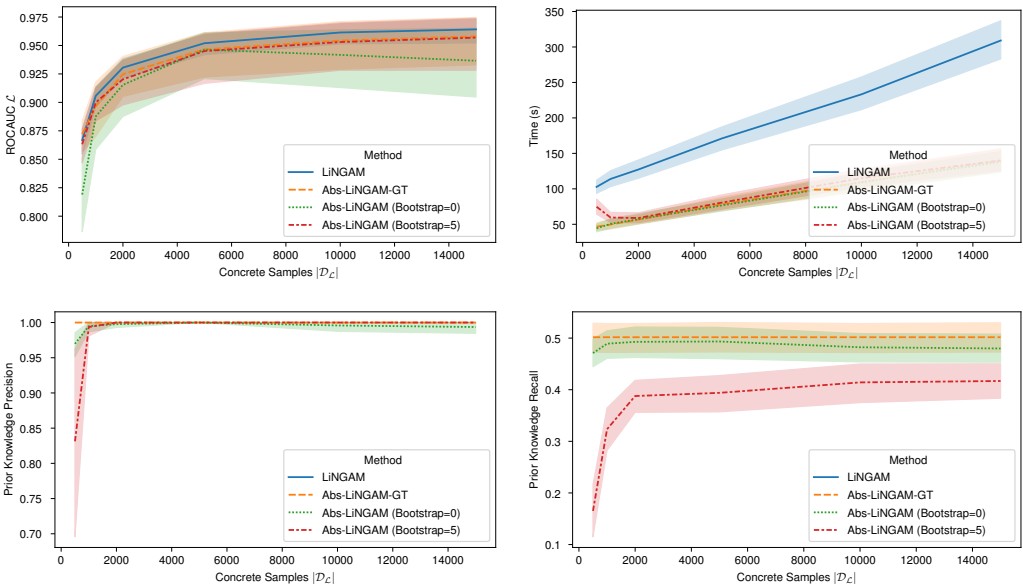

Figure 9: Results of Abs-LiNGAM over pairs of abstract ($b = 10$ nodes) and concrete ($d \in [50, 100]$ nodes) linear SCMs. In all subfigures we plot the results for an increasing number of concrete samples $|\mathcal{D}_\mathcal{L}|$. Abs-LiNGAM-GT denotes a ground truth oracle where the abstraction function and the abstract model are given. The first plot (top left) shows the ROC-AUC of the retrieved concrete causal model $\hat{\mathcal{L}}$. The second plot (top right) shows the execution time required to retrieve the concrete causal model. The third and fourth plots (bottom) show the precision and recall of the prior knowledge inferred by the learned abstraction function $\hat{\mathbf{T}}$ and the consequent abstract model $\hat{\mathcal{H}}$. All results are averaged over 30 independent runs with $|\mathcal{D}_J| = 200$ paired samples.

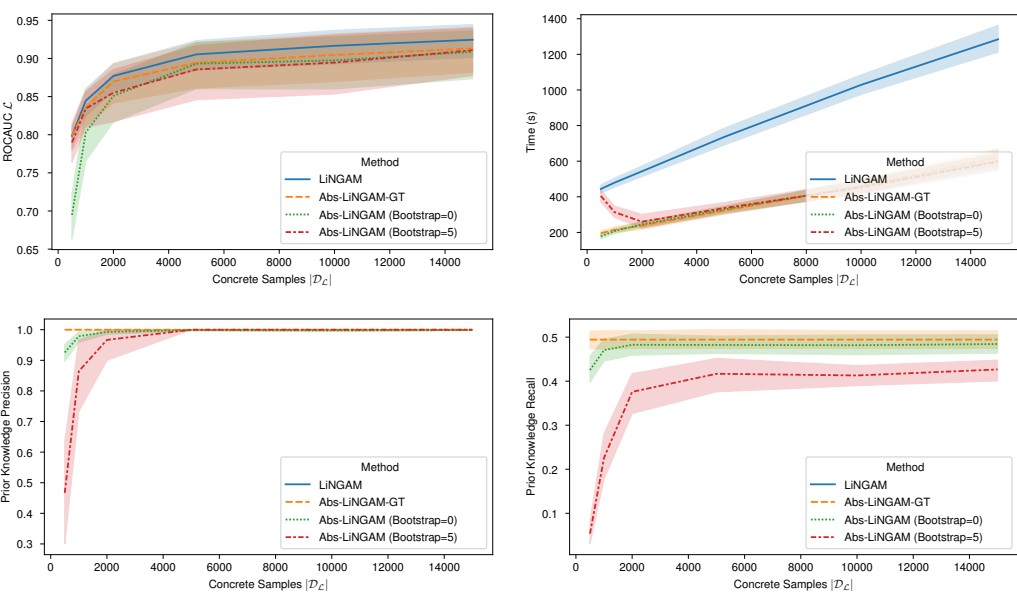

Figure 10: Results of Abs-LiNGAM over pairs of abstract ($b = 10$ nodes) and concrete ($d \in [100, 150]$ nodes) linear SCMs. In all subfigures we plot the results for an increasing number of concrete samples $|\mathcal{D}_\mathcal{L}|$. Abs-LiNGAM-GT denotes a ground truth oracle where the abstraction function and the abstract model are given. The first plot (top left) shows the ROC-AUC of the retrieved concrete causal model $\hat{\mathcal{L}}$. The second plot (top right) shows the execution time required to retrieve the concrete causal model. The third and fourth plots (bottom) show the precision and recall of the prior knowledge inferred by the learned abstraction function $\hat{\mathbf{T}}$ and the consequent abstract model $\hat{\mathcal{H}}$. All results are averaged over 30 independent runs with $|\mathcal{D}_J| = 300$ paired samples.

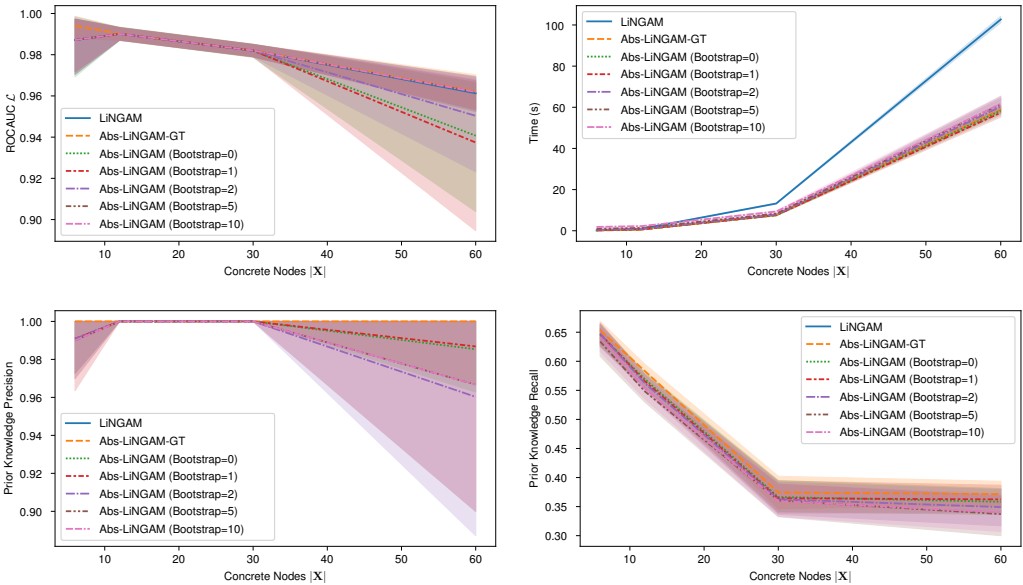

Figure 11: Results of Abs-LiNGAM over pairs of abstract ($b = 5$ nodes) and concrete models with increasing size $d \in [5, 60]$. Abs-LiNGAM-GT denotes a ground truth oracle where the abstraction function and the abstract model are given. The first plot (top left) shows the ROC-AUC of the retrieved concrete causal model $\hat{\mathcal{L}}$. The second plot (top right) shows the execution time required to retrieve the concrete causal model. The third and fourth plots (bottom) show the precision and recall of the prior knowledge inferred by the learned abstraction function $\hat{\mathbf{T}}$ and the consequent abstract model $\hat{\mathcal{H}}$. All results are averaged over 30 independent runs with $|\mathcal{D}_\mathcal{L}| = 1500$ concrete samples and $|dset_J| = 2 \cdot |\boldsymbol{X}|$ paired samples.

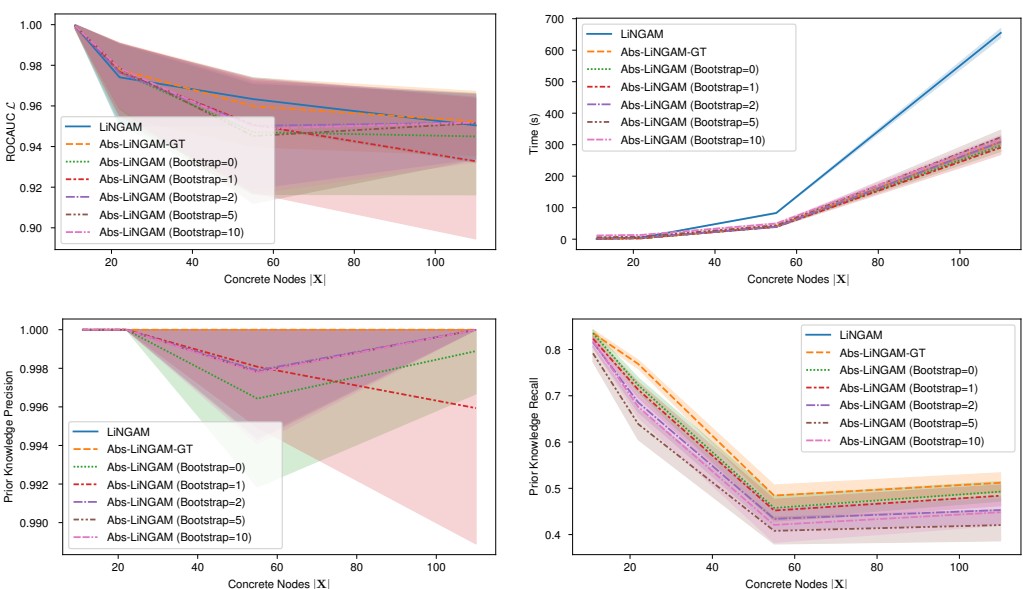

Figure 12: Results of Abs-LiNGAM over pairs of abstract ($b = 10$ nodes) and concrete models with increasing size $d \in [10, 120]$. Abs-LiNGAM-GT denotes a ground truth oracle where the abstraction function and the abstract model are given. The first plot (top left) shows the ROC-AUC of the retrieved concrete causal model $\hat{\mathcal{L}}$. The second plot (top right) shows the execution time required to retrieve the concrete causal model. The third and fourth plots (bottom) show the precision and recall of the prior knowledge inferred by the learned abstraction function $\hat{\mathbf{T}}$ and the consequent abstract model $\hat{\mathcal{H}}$. All results are averaged over 30 independent runs with $|\mathcal{D}_{\mathcal{L}}| = 1500$ concrete samples and $|dset_J| = 2 \cdot |\mathbf{X}|$ paired samples.

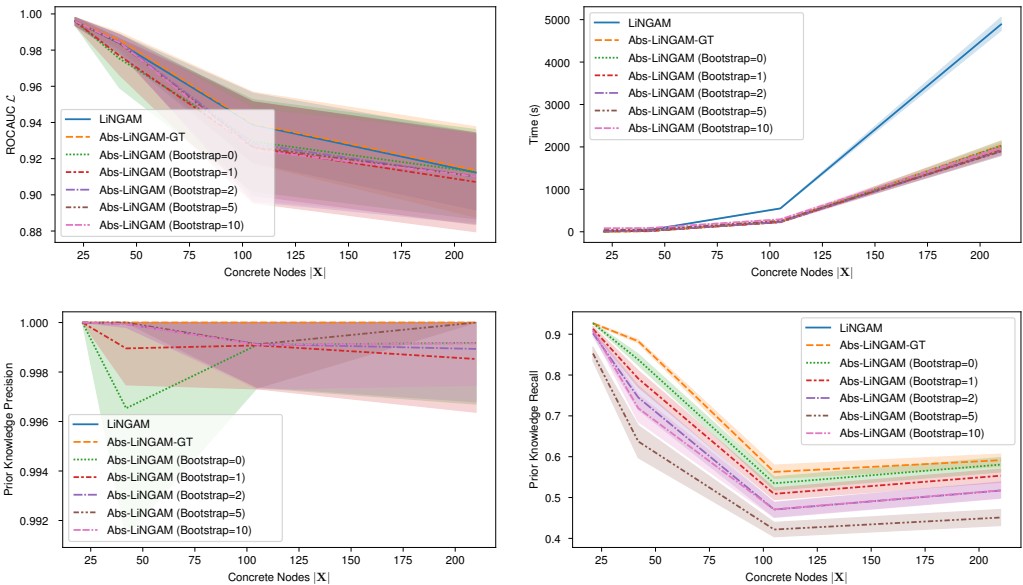

Figure 13: Results of Abs-LiNGAM over pairs of abstract ($b = 10$ nodes) and concrete models with increasing size $d \in [20, 180]$. Abs-LiNGAM-GT denotes a ground truth oracle where the abstraction function and the abstract model are given. The first plot (top left) shows the ROC-AUC of the retrieved concrete causal model $\hat{\mathcal{L}}$. The second plot (top right) shows the execution time required to retrieve the concrete causal model. The third and fourth plots (bottom) show the precision and recall of the prior knowledge inferred by the learned abstraction function $\hat{\mathbf{T}}$ and the consequent abstract model $\hat{\mathcal{H}}$. All results are averaged over 30 independent runs with $|\mathcal{D}_{\mathcal{L}}| = 1500$ concrete samples and $|dset_J| = 2 \cdot |\mathbf{X}|$ paired samples.

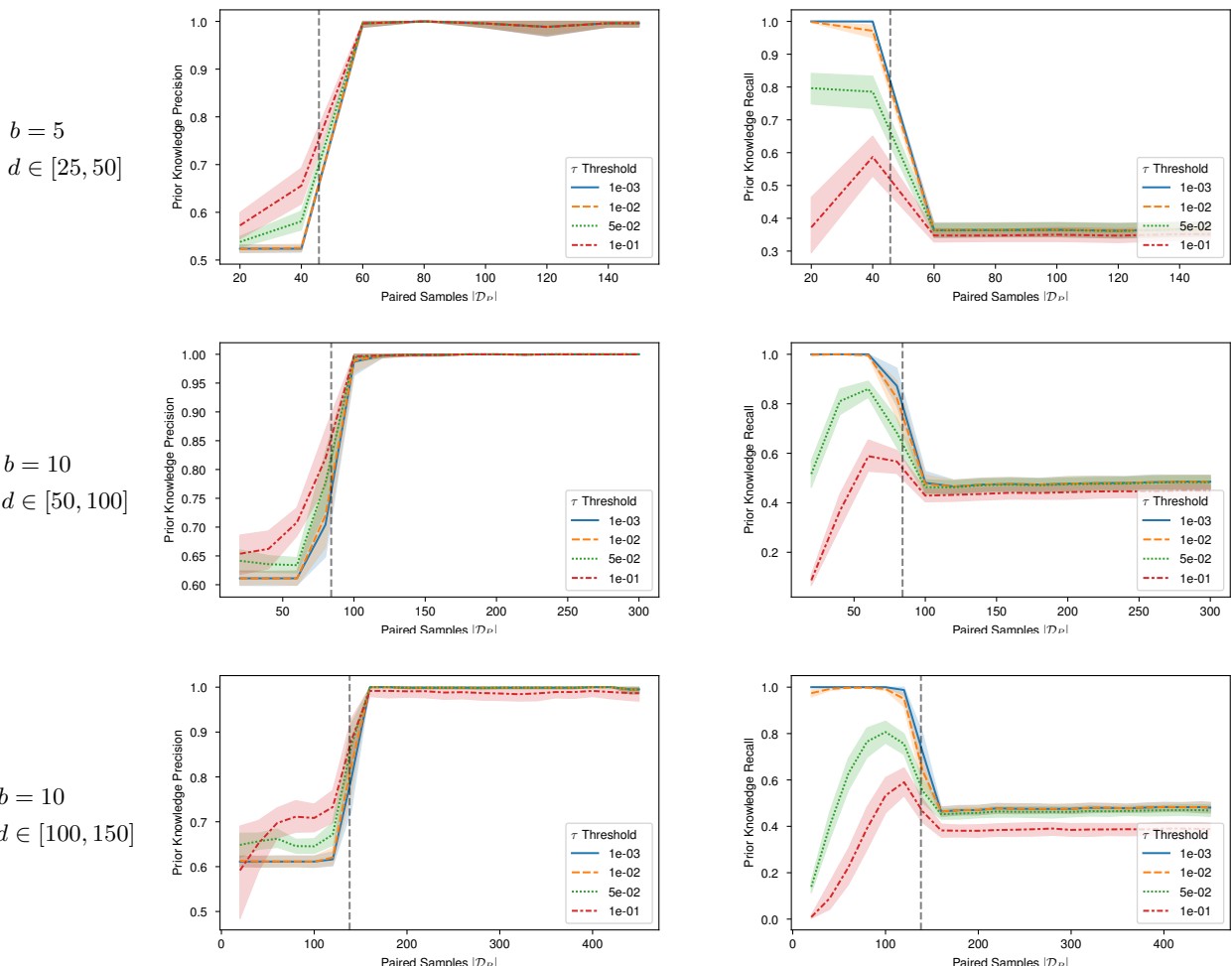

Figure 14: Analysis of the prior knowledge inferred by the learned abstraction function $\hat{\mathbf{T}}$ and the consequent abstract model $\hat{\mathcal{H}}$ on a concrete model ($d \in [25, 50]$ nodes). We report precision (left) and recall (right) of the prior knowledge for different thresholds to mask the learned abstraction function $\hat{\mathbf{T}}$.

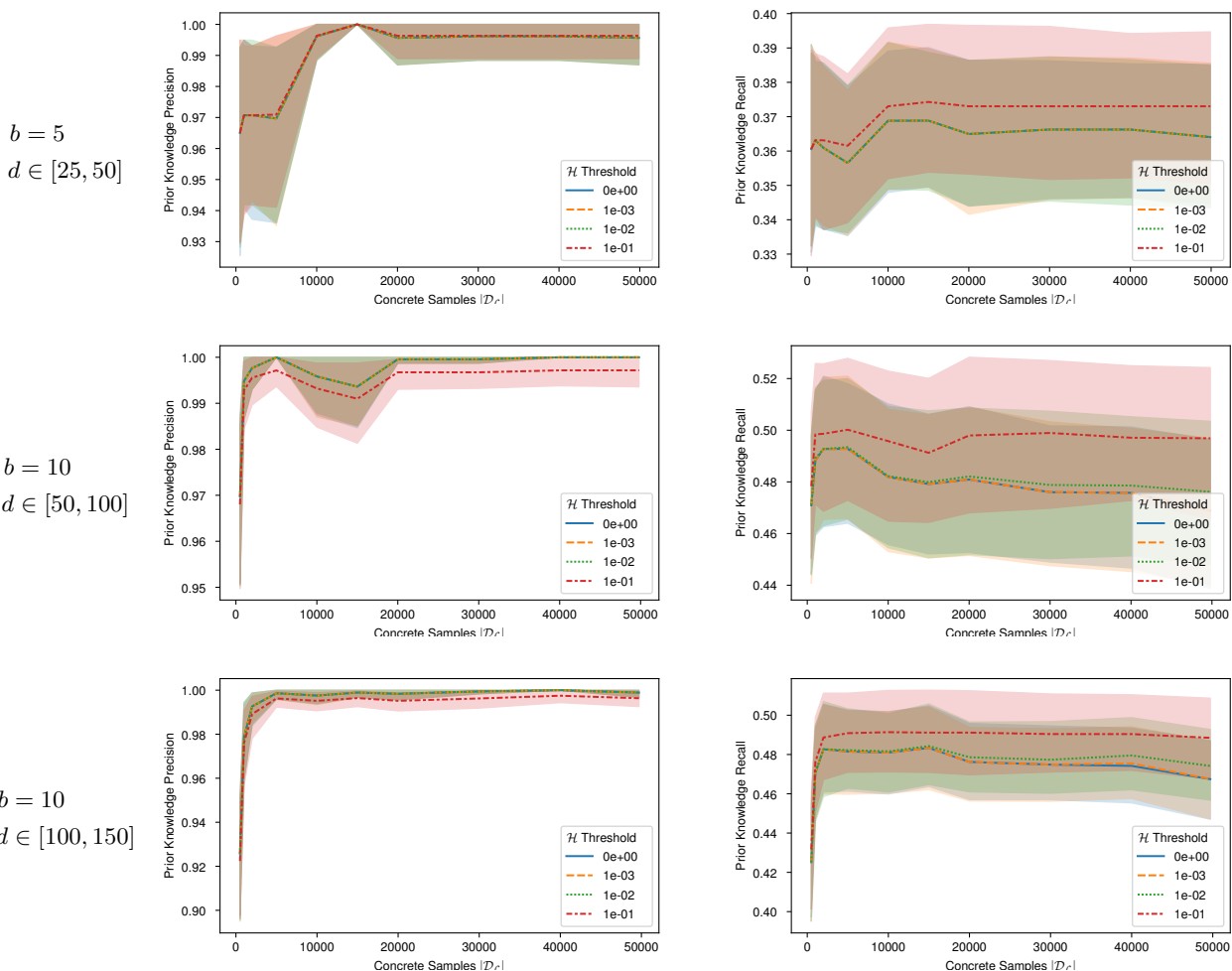

Figure 15: Analysis of the prior knowledge inferred by the learned abstraction function $\hat{\mathbf{T}}$ and the consequent abstract model $\hat{\mathcal{H}}$ on a concrete model. We report precision (left) and recall (right) of the prior knowledge for different thresholds to mask the learned abstract model $\hat{\mathcal{H}}$.

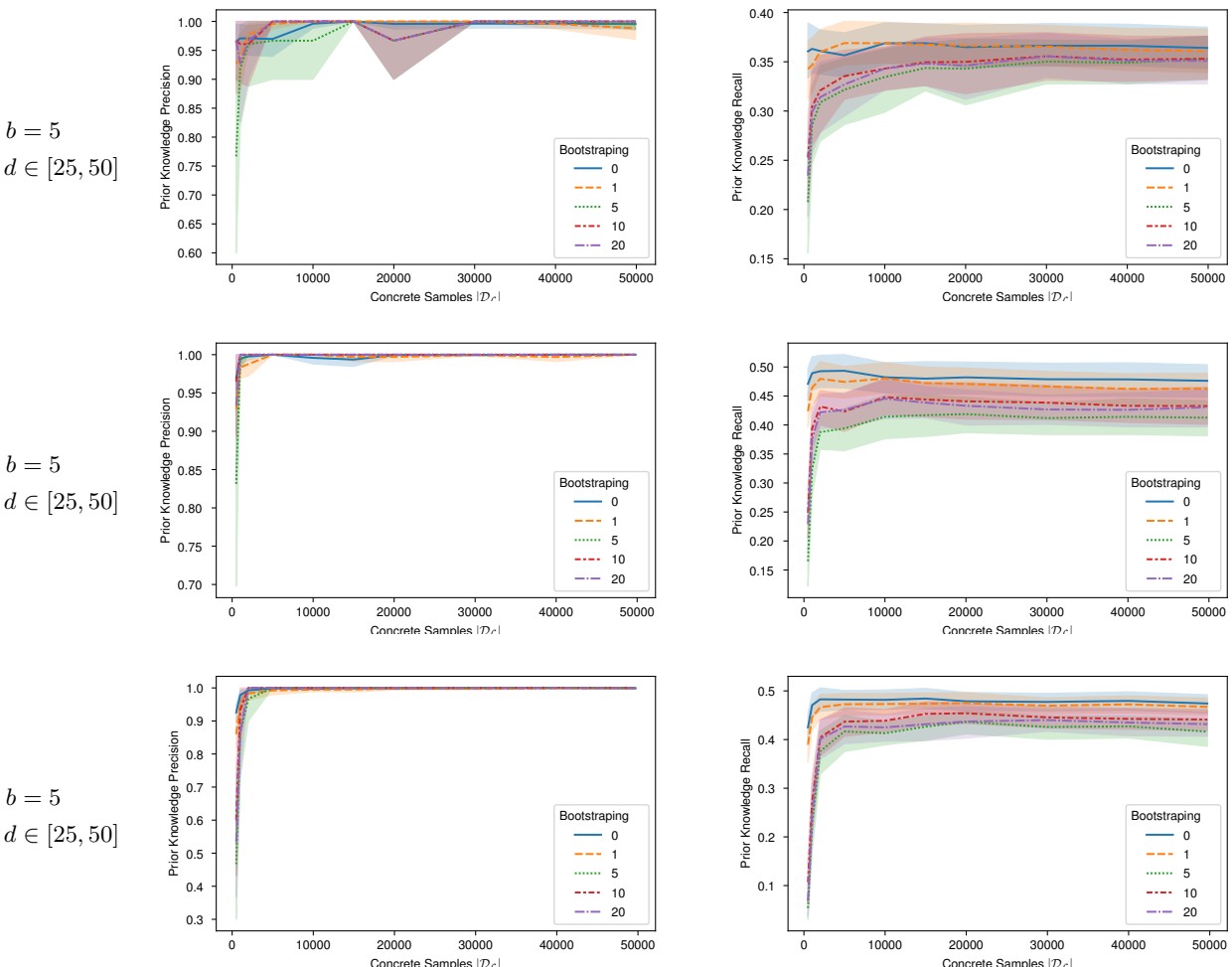

Figure 16: Analysis of the prior knowledge inferred by the learned abstraction function $\hat{\mathbf{T}}$ and the consequent abstract model $\hat{\mathcal{H}}$, with $b$ nodes, on a concrete model with $d$ nodes. We report precision (left) and recall (right) of the prior knowledge for different number of bootstrapped samples to fit the abstract model $\hat{\mathcal{H}}$.

| Method | ROCAUC | Precision | Recall | Time |
|---|---|---|---|---|
| Abs-Fit (Bootstrap=0) | 0.965±0.066 | 0.957±0.123 | 0.940±0.074 | 42±13 |
| Abs-Fit (Bootstrap=1) | 0.977±0.012 | 0.980±0.015 | 0.953±0.026 | 42±14 |
| Abs-Fit (Bootstrap=2) | 0.977±0.012 | 0.980±0.015 | 0.952±0.027 | 43±11 |
| Abs-Fit (Bootstrap=5) | 0.977±0.012 | 0.980±0.015 | 0.952±0.027 | 45±13 |
| Abs-Fit (Bootstrap=10) | 0.977±0.012 | 0.980±0.015 | 0.952±0.027 | 46±12 |
| Abs-LiNGAM-GT | 0.977±0.011 | 0.982±0.013 | 0.953±0.026 | 45±15 |
| DirectLiNGAM | 0.977±0.011 | 0.980±0.013 | 0.953±0.026 | 61±12 |

Table 1: Results of Abs-LiNGAM over pairs of abstract ($b = 5$ nodes) and concrete ($d \in [25, 50]$ nodes) linear SCMs. Abs-LiNGAM-GT denotes a ground truth oracle where the abstraction function and the abstract model are given. All results are averaged over 30 independent runs with $|\mathcal{D}_\mathcal{L}| = 15000$ concrete and $|\mathcal{D}_J| = 150$ paired samples.

| Method | ROCAUC | Precision | Recall | Time |
|---|---|---|---|---|
| Abs-LiNGAM (Bootstrap=0) | 0.963±0.043 | 0.939±0.119 | 0.926±0.067 | 179±53 |
| Abs-LiNGAM (Bootstrap=1) | 0.952±0.066 | 0.914±0.169 | 0.914±0.085 | 181±53 |
| Abs-LiNGAM (Bootstrap=2) | 0.968±0.027 | 0.956±0.041 | 0.930±0.061 | 182±50 |
| Abs-LiNGAM (Bootstrap=5) | 0.968±0.027 | 0.955±0.041 | 0.930±0.061 | 189±51 |
| Abs-LiNGAM (Bootstrap=10) | 0.968±0.027 | 0.954±0.040 | 0.930±0.061 | 194±51 |
| Abs-LiNGAM-GT | 0.969±0.026 | 0.965±0.022 | 0.931±0.060 | 186±54 |
| DirectLiNGAM | 0.968±0.025 | 0.958±0.020 | 0.930±0.061 | 394±94 |

Table 2: Results of Abs-LiNGAM over pairs of abstract ($b = 10$ nodes) and concrete ($d \in [50, 100]$ nodes) linear SCMs. Abs-LiNGAM-GT denotes a ground truth oracle where the abstraction function and the abstract model are given. All results are averaged over 30 independent runs with $|\mathcal{D}_\mathcal{L}| = 15000$ concrete and $|\mathcal{D}_J| = 270$ paired samples.

| Method | ROCAUC | Precision | Recall | Time |
|---|---|---|---|---|
| Abs-LiNGAM | 0.927±0.070 | 0.919±0.119 | 0.845±0.132 | 748±121 |
| Abs-LiNGAM (Bootstrap=1) | 0.913±0.083 | 0.877±0.187 | 0.834±0.136 | 731±116 |
| Abs-LiNGAM (Bootstrap=2) | 0.925±0.072 | 0.912±0.130 | 0.844±0.132 | 738±123 |
| Abs-LiNGAM (Bootstrap=5) | 0.926±0.067 | 0.913±0.109 | 0.844±0.130 | 755±140 |
| Abs-LiNGAM (Bootstrap=10) | 0.927±0.065 | 0.918±0.090 | 0.844±0.130 | 775±183 |
| Abs-LiNGAM-GT | 0.927±0.069 | 0.920±0.117 | 0.845±0.131 | 763±116 |
| DirectLiNGAM | 0.928±0.061 | 0.925±0.047 | 0.844±0.128 | 1608±212 |

Table 3: Results of Abs-LiNGAM over pairs of abstract ($b = 10$ nodes) and concrete ($d \in [100, 150]$ nodes) linear SCMs. Abs-LiNGAM-GT denotes a ground truth oracle where the abstraction function and the abstract model are given. All results are averaged over 30 independent runs with $|\mathcal{D}_\mathcal{L}| = 15000$ concrete and $|\mathcal{D}_J| = 270$ paired samples.

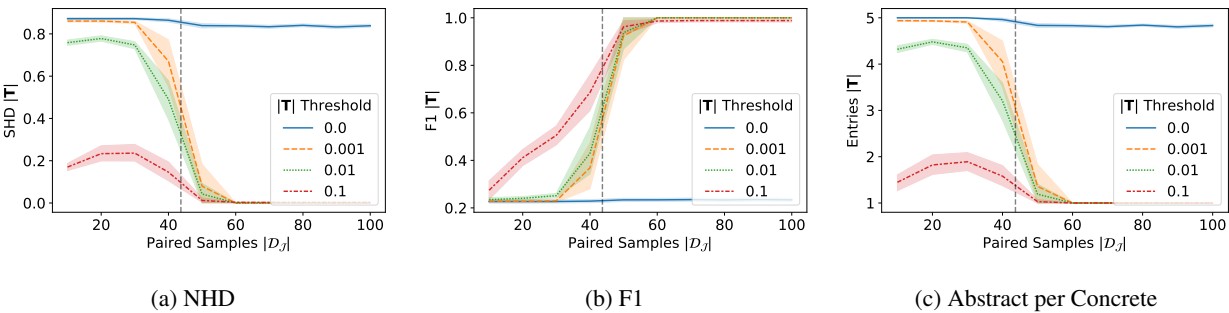

(a) NHD          (b) F1          (c) Abstract per Concrete

Figure 17: Reconstruction metrics of the linear abstraction function $\mathbf{T}$ over pairs of abstract ($b = 5$ nodes) and concrete ($d \in [25, 50]$ nodes) linear SCMs for an increasing number of paired samples $|\mathcal{D}_{\mathcal{J}}|$. For different thresholds, we report the normalized Hamming Distance (*left*), the F1 score (*center*), and the average number of abstract variables assigned to each concrete variable (*right*). All results are averaged over 30 independent runs.

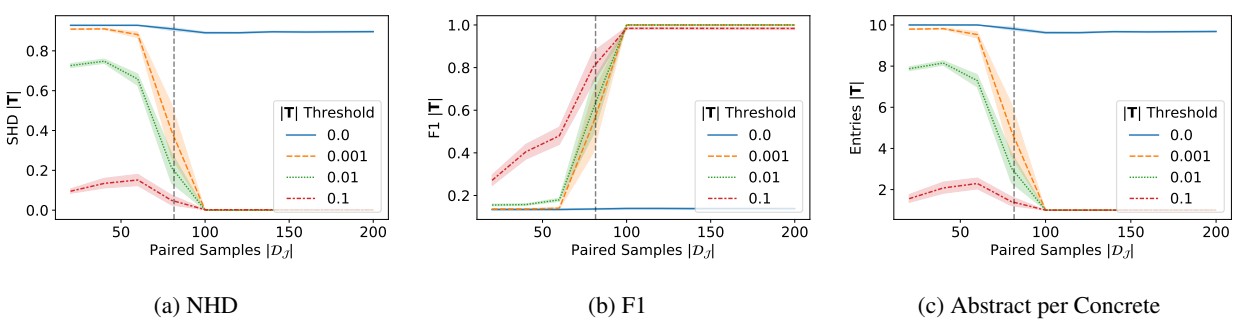

(a) NHD          (b) F1          (c) Abstract per Concrete

Figure 18: Reconstruction metrics of the linear abstraction function $\mathbf{T}$ over pairs of abstract ($b = 10$ nodes) and concrete ($d \in [50, 100]$ nodes) linear SCMs for an increasing number of paired samples $|\mathcal{D}_{\mathcal{J}}|$. For different thresholds, we report the normalized Hamming Distance (*left*), the F1 score (*center*), and the average number of abstract variables assigned to each concrete variable (*right*). All results are averaged over 30 independent runs.

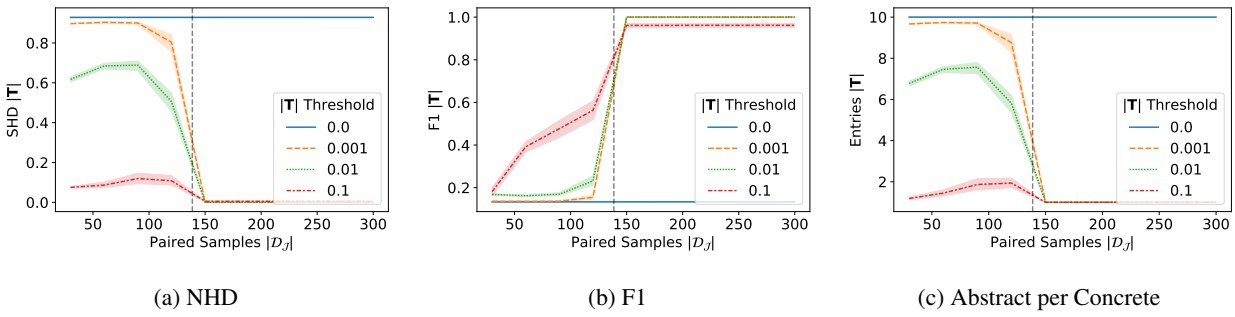

(a) NHD          (b) F1          (c) Abstract per Concrete

Figure 19: Reconstruction metrics of the linear abstraction function $\mathbf{T}$ over pairs of abstract ($b = 10$ nodes) and concrete ($d \in [100, 150]$ nodes) linear SCMs for an increasing number of paired samples $|\mathcal{D}_{\mathcal{J}}|$. For different thresholds, we report the normalized Hamming Distance (*left*), the F1 score (*center*), and the average number of abstract variables assigned to each concrete variable (*right*). All results are averaged over 30 independent runs.