# OpenReview forum: "Learning Causal Abstractions of Linear Structural Causal Models"
_auai.org/UAI/2024/Conference — UAI 2024 poster_

### Official Review · Reviewer_Kv4K · 2024-03-08

**Q2-1 Originality-Novelty:** 3
**Q2-2 Correctness-Technical Quality:** 2
**Q2-5 Clarity Of Writing:** 4

**Q1 Summary And Contributions:**

This paper gives some new results concerning causal abstractions assuming linear SCMs and a linear transformation between the concrete and the abstract model. Firstly it gives necessary and sufficient conditions in the concrete graph to have an edge in an abstract graph. Secondly it provides an algorithm to sample a linear concrete SCM which is compatible with a given abstract SCM. Thirdly, it provide a new causal discovery algorithm called Abs-LiNGAM, which starts by learning the linear trabsformation, then it uses this learned transformation to transform concrete data to abstract data. After that, it learns the abstract model from abstract data + transformed data. Finally, it learns the concrete model using the absract model + concerete data. It was also shown that this strategy outperform the classifical startegy (without using abstract model) on simulated data.

**Q2-3 Extent To Which Claims Are Supported By Evidence:**

3: Good: the main claims are supported by convincing evidence (in the form of adequate experimental evaluation, proofs, (pseudo-)code, references, assumptions).

**Q2-4 Reproducibility:**

2: Fair: key resources (e.g. proofs, code, data) are unavailable but key details (e.g. proof sketches, experimental setup) are sufficiently well-described for an expert to confidently reproduce the main results.

**Q3 Main Strengths:**

The motivations are clear and the contributions are interesting. The paper is well written and very comprehensible (except lemma 3, see Q4). It includes clear pseudo-code, experimental validation, and proofs available in the Appendix. Moreover, the results presented in Section 3 have the potential to impact other areas of causal inference beyond causal discovery.

**Q4 Main Weakness:**

* No proof or sketch in the main paper eventhough every proofs are extensively described in the appendix.
* It is not clear why the faithfulness assumption is needed.
* In 3.3, the Lemma 3 is unclear to me. As I understand the lemma, any relevant variable of an abstract parent of Y is in the concrete block of Y. For example in Figure 1, we can see that $X_7\in\Pi_R(Y_2)$ and there is a T-direct path from $X_3$ to $X_7$ therefore $X_3\in\Pi(Y_2)$ according to lemma 3 and we can also see that $X_3\in\Pi_R(Y_1)$ therefore $X_3\in\Pi(Y_1)$ according to lemma 3.
* No real world applications.

**Q5 Detailed Comments To The Authors:**

*  Isn't the minimality assumption sufficient for semi-parametric causal discovery methods? Can you provide an intuition why the faithfulness assumption is needed in this paper?
* In 1. Introduction, "viceversa" is in 2 words: "vice versa".
* In 2. Background, in the definition of a $\tau$-abstraction, I don't really understand how the intervention map $\omega$ is defined. The reference given [Massidda et al. 2023] seems to hold for soft interventions. Moreover, the set of interventions \textbf{I} is defined as the pre-image after being used multiple times which makes things a bit unclear in this paragraph.
* In 3.1, you claim to have proven that "strong abstraction entails constructive abstraction in the linear case", however no specific proof of this is given and you have not defined or explained what is a "constructive abstraction".
* In 3.2 , in the 1st paragraph I think you forgot the word "it". "whenever is mediated by irrelevant variables only" -> "whenever it is mediated by irrelevant variables only".
* In 5. Experimental Results, some analysis of the results would be nice. For example, how do you explain the drop (compared to DirectLiNGAM and the Bootstrap version) of the performance of Abs-LiNGAM when the number of concrete nodes grows.
* It would have been insightful to observe the algorithm's performance under varying levels of abstraction, both when the abstraction is significantly distant from the concrete graph and when it is closely aligned.
* Would it be possible to illustrate the performance of Abs-LiNGAM on real data? This would also motivate the usage of such strategy in real world applications.
* In the Appendix B.1, the Lemma 1 does not match exactly the one in the paper.
* In the Appendix B.1, (top of the 2nd column) there are small typos where you confuse $L^i_{\Pi_R}(Y)$ and $L^i_{\Pi_R}(Y_1)$ as well as $\tau_1$ and $\tau_{Y_1}$.
* In the Appendix B.1, in equation (25), I don't understand why i cannot be an intervention on a bigger set of concrete variables than $\Pi_R(Y_1)$. In other words, why is equation (25) not of the form i=(V<-v) with $\Pi_R(Y_1)\subseteq V$?
* In the Appendix B.1, there is a typo: "Then, where we assume the existence [...]" -> "Then, we assume the existence [...]".
* In the Appendix B.1, in equation (34), the inequality sign should be an equal sign.
* In the Appendix B.2, I am unsure if c and c' differ *only* on $X_1$ or if they can as well differ on other relevant variable of Y_1. Could you clarify?
* In the Appendix B.3, in the proof you implicitly assume that Y_3 is different from Y_1 or Y_2. What happens if Y_3=Y_1 or Y_3=Y_2?

I did not dive deeply in the Appendix after B.3.

**Q9 Complying With Reviewing Instructions:**

Yes

---

> ### Author Rebuttal · Authors · 2024-04-04
>
> We thank the reviewer for their detailed comments.
>
> > Isn't the minimality assumption sufficient?
>
> The faithfulness assumption enables the graphical characterization of abstract arcs in terms of concrete edges. For instance, in Example 1, we report a concrete causal model that is causally minimal but not faithful and where, despite the presence of T-direct paths between the relevant variables there is no abstract edge. If we were to run Abs-LiNGAM on Example 1, we would learn the correct abstract graph. However, since the abstract model does not contain an edge from $Y_1$ to $Y_2$, Abs-LiNGAM would wrongly infer that $X_1$ is not an ancestor of $X_4$ in the concrete model, thus inhibiting its correct recovery.
>
> > In Lemma 3 any relevant variable of an abstract parent of Y is in the concrete block of Y.
>
> We thank the reviewer for pointing out the typo in the formula that did not explicitly mention this case. In the proof we implicitly considered this case, since we show that the exogenous abstraction function does *not* depend on the exogenous terms of the parents, as they must be intervened to let $Y$ depend only on its exogenous term. We have now revised and clarified Lemma 3, which now correctly states that "$X\in\Pi(Y)$ if and only if
> $X\in\Pi_R(Y)$, or
> $X$ **is not relevant and** exists $X’\in\Pi_R(Y)$ s.t. $X \to^{\mathbf{T}} X’$" as intended.
>
> > In the Appendix B.1, why it cannot be an intervention on a bigger set of concrete variables than $\Pi_R(Y_1)$.
>
> In Appendix B.1, we focused on interventions on relevant variables. While we can intervene on ignored variables without changing the abstract intervention, intervening on other *relevant* variables changes it. We will clarify this point, which does not affect any proofs.
>
> > In the definition of a $\tau$-abstraction how is the intervention map $\omega$ defined?
>
> Massidda et al. (2023) covers hard interventions as a special case where the structural function of a variable is replaced by a constant. As we show in Appendix B.1, concrete interventions must fix the whole set of relevant variables. So, for any intervention $i=(\Pi_R(Y) \gets c)$, the intervention map reduces to $Y \gets \tau_Y(F^i_{\Pi_R(Y)} ( \tau^{-1}(y), \gamma^{-1}(u) ) = \tau_Y(c) = k$.
>
> > The set of interventions $\mathbf{I}$ is defined as the pre-image
>
> The set of concrete interventions $\mathbf{I}$ contains all possible implementations of an abstract hard intervention. Formally, given the explicit form of $\omega$, it contains any intervention $i$ on the concrete model such that $\omega(i)$ is an hard intervention.
>
> > you claim to have proven that "strong abstraction entails constructive abstraction in the linear case"
>
> Beckers et al. (2019) define an abstraction as constructive whenever abstract variables depend on disjoint sets of concrete variables. They also conjecture that strong abstraction implies constructivity, i.e., disjointness. Lemma 1 proves this exact relation in the linear case, which we will clarify by correctly reporting constructive abstraction.
>
> > How do you explain the drop in the performance of Abs-LiNGAM when the number of concrete nodes grows?
> > It would have been insightful to observe the algorithm's performance under varying levels of abstraction
>
> A growing number of concrete nodes increases the reconstruction error of T and thus the correct recovery of the abstract model. This leads to a drop in precision of the inferred knowledge (Figures 10, 11, 12), which is improved by bootstrapping the abstract discovery step. These experiments also showcase what happens when the level of abstraction varies, e.g. when it gets more distant from the concrete graph with a larger ratio of concrete nodes per abstract node.
>
> > No real world applications.
>
> There are exciting applications in interpretable ML that leverage causal abstraction to interpret LLMs (Wu et al. 2024, Geiger et al. 2023b). In that context, most operators are linear and the abstraction functions are linear operators representing “concepts”. These works assume the abstract model is known, while our method learns an abstraction function from a small set of labeled “concepts”, extending their applicability to more realistic cases.
>
> > Appendix B.2, I am unsure if c and c' differ only on $X_1$
>
> Yes, the assignment differs only in $X_1$. The proof shows that any $X_1$ is sufficient to entail the presence of an abstract arc $Y_1 \to Y_2$.
>
> > Appendix B.3, in the proof you implicitly assume $Y_3$ different from $Y_1$ or $Y_2$.
>
> Due to the acyclicity of the abstract graph, the case $Y_3=Y_1$ or $Y_3=Y_2$ can arise only at the beginning (resp. the end) of the path. In this case, we could take the successor until we get one from a different abstract variable, if any. If there is none, then there exists a T-direct path between the relevant variables of $Y_1, Y_2$ and we fallback to Lemma 2, implying that $Y_1 \to Y_2$, which entails by definition $Y_1$ is an ancestor of $Y_2$.

---

### Official Review · Reviewer_Yoqa · 2024-03-11

**Q2-1 Originality-Novelty:** 2
**Q2-2 Correctness-Technical Quality:** 3
**Q2-5 Clarity Of Writing:** 3

**Q10 Ethical Concerns:**

There are no ethical concerns.

**Q1 Summary And Contributions:**

**Short Summary:**

 This paper establishes necessary and sufficient conditions for **linear** abstraction of a **linear** SCM and introduces Abs-LiNGAM which judiciously uses data from the high-level and low-level to do structure learning on a low-level causal model.

**Long Summary/Contributions:**

1. This paper explores the setting when a low-level linear SCM can be abstracted
linearly to a high-level SCM. It establishes necessary and sufficient graphical conditions such that every
intervention at the high-level has a corresponding intervention at the low-level.

2. When a sufficiently
large dataset entailed by the low-level SCM, as well as a smaller dataset of pairs entailed by the
low-level and high-level SCM are available, and linear non-gaussianity of the additive noise model
holds, the paper introduces the Abs-LiNGAM method to learn the causal model at the low-level. Abs-LiNGAM
information of the linear abstraction function and the constraints that the high-level model imposes
on the low-level model to ensure intervention commutativity

**Q2-3 Extent To Which Claims Are Supported By Evidence:**

2: Fair: the main claims are somewhat supported by evidence (but the experimental evaluation may be weak, or does not match entirely with the claims, important baselines may be missing, proofs contain important ideas but lack rigor, algorithmic details are only discussed superficially, references are imprecise, assumptions are not sufficiently motivated or explicated, etc.).

**Q2-4 Reproducibility:**

3: Good: key resources (e.g. proofs, code, data) are available and key details (e.g. proofs, experimental setup) are sufficiently well-described for competent researchers to confidently reproduce the main results.

**Q3 Main Strengths:**

1. This paper is clearly organised with several illustrative examples and accompanying figures to guide the reader through the new definitions and theorems.
2. Consistent abstraction of SCMs is an interesting and relevant topic, and establishing results for when an abstraction is possible are needed, as this paper does.

**Q4 Main Weakness:**

1. While the paper fills a gap in the literature, it is arguably (and perhaps also subjectively) a small gap. The necessary and sufficient condition for abstraction in the linear ANM setting boils down to having a direct path from every low-level variable to its corresponding high-level variable, that only passes through low-level variables irrelevant for abstraction.

2. The evidence for experimental superiority of Abs-LiNGAM is not very strong, discounting the computational time, and does not incorporate the fact that pairs of low-level and high-level data would be expensive to gather.

**Q5 Detailed Comments To The Authors:**

**General Comments:**

The introduction lacks clarity, in particular the second paragraph. Words such as  'concrete', 'relevant' and 'arc' that are domain-specific jargon, should ideally be avoided in the introduction, or explained when used. Additionally, the second sentence in paragraph two seems to imply that abstraction itself is the consistency relation, and the consistency relation is then explained by means of abstraction, there is some recursiveness here that should be fixed.

**Specific Comments:**

 It is not clear to me how Lemma 2 and Theorem 1 can be true at the same time. Example 2 is an example of the violation of Theorem 1, but if a theorem is violated it fails to be a theorem.

**Typos/Grammar/Notational Issues:**

1. Section 1, paragraph 2, it should say "vice versa" not "viceversa".
2. Figure 1, the caption can be made more readable by introducing labels (a) and (b), instead of 'left' and 'right'.
3. In Algorithm 1, the dimensions of the low-level adjacency matrix should be $d \times d$.

**Q9 Complying With Reviewing Instructions:**

Yes

---

> ### Author Rebuttal · Authors · 2024-04-04
>
> We thank the reviewer for their feedback. We answer each point in the following.
>
> > While the paper fills a gap in the literature, it is arguably (and perhaps also subjectively) a small gap.
>
> As highlighted in the introduction, our contribution is the first to study the necessary graphical and parametric conditions for causal abstraction. This enables us to tackle the problem of recovering an abstraction function without **any** information on the abstract graph. In comparison, all previous works required either to know the abstract model (Zennaro et al. 2023, Geiger et al. 2023) or the abstract graph (Kekic et al. 2023, Felekis et al. 2023) among other assumptions, as we detailed in Section 6.
>
> > The introduction lacks clarity, in particular the second paragraph. Words such as 'concrete', 'relevant' and 'arc' that are domain-specific jargon, should ideally be avoided in the introduction, or explained when used. Additionally, the second sentence in paragraph two seems to imply that abstraction itself is the consistency relation, and the consistency relation is then explained by means of abstraction, there is some recursiveness here that should be fixed.
>
> We thank the reviewer for their suggestion, we will clarify the use of abstraction specific terms within the introduction to ease readability. To clarify, “arc” is simply a synonym of edge.
>
> > It is not clear to me how Lemma 2 and Theorem 1 can be true at the same time.
>
> Lemma 2 states that in a T-abstraction whenever a T-direct path between relevant variables exists, the corresponding abstract variables are connected by an edge. Theorem 1 states that in a T-abstraction the connection between two abstract variables entails that any relevant variable of the former feeds through a T-direct path towards at least a relevant variable of the latter. The two results are not in contradiction, as the reviewer can easily verify from the proofs respectively in Appendix B.2 and B.5.
>
> > Example 2 is an example of the violation of Theorem 1, but if a theorem is violated it fails to be a theorem.
>
> Theorem 1 states the necessary and sufficient conditions for the existence of an abstract edge whenever T-abstraction holds. From this implication, it directly follows that whenever these conditions are not satisfied, then the models are not in an abstraction relation. Example 2 showcases such violation and the portrayed models are *not* in an abstraction relation. Intuitively, we could sketch the proof of Theorem B.5 as follows: by performing two interventions $i=(X_1, X_2 \gets a, b)$ and $i’ = (X_1, X_2 \gets a’, b)$, with $a \neq a’$, the effect on $X_3$ would be the same as it only depends on the value of $X_2$. However, in the abstract model we would have two distinct abstract interventions $j,j’$ since $\tau_1(a,b)\neq\tau_1(a’,b)$, leading to two distinct values of the abstract variable $Y_2$. Such inconsistency shows the problem with violating the necessary condition of Theorem 1.
>
> > The evidence for experimental superiority of Abs-LiNGAM is not very strong, discounting the computational time, and does not incorporate the fact that pairs of low-level and high-level data would be expensive to gather.
>
> The goal of our experimental section is not to highlight superiority in terms of reconstruction when compared to LiNGAM which, in the limit of infinite data, has identifiability results ensuring the retrieval of the model. Abs-LiNGAM aims instead to produce abstract information from a significantly smaller dataset of high-level observations to speed up the causal discovery of the concrete model. In general, our aim here was to showcase one possible application enabled by our theoretical findings on linear causal abstraction.

---

### Official Review · Reviewer_y7E4 · 2024-03-12

**Q2-1 Originality-Novelty:** 2
**Q2-2 Correctness-Technical Quality:** 3
**Q2-5 Clarity Of Writing:** 3

**Q10 Ethical Concerns:**

No.

**Q1 Summary And Contributions:**

In this work, the authors provide (a) theoretical results on causal abstractions of linear SCMs and (b) a new method for causal discovery on non-Gaussian linear models that learns the causal graph of the concrete variables by first learning the abstraction function and the graph over the abstract variables, and by subsequently using these as constraints for the concrete graph. According to the author’s empirical results, this method outperforms DirectLinGaM in terms of computation time while performing roughly equal in terms of quality of the inference.

**Q2-3 Extent To Which Claims Are Supported By Evidence:**

3: Good: the main claims are supported by convincing evidence (in the form of adequate experimental evaluation, proofs, (pseudo-)code, references, assumptions).

**Q2-4 Reproducibility:**

3: Good: key resources (e.g. proofs, code, data) are available and key details (e.g. proofs, experimental setup) are sufficiently well-described for competent researchers to confidently reproduce the main results.

**Q3 Main Strengths:**

The theoretical discussion is interesting and well-supported by illustrative examples. The second part of the absLinGaM algorithm in which abstract knowledge constraints the concrete search task, is a valuable contribution, and it would be interesting to see whether speed/performance improves further if knowledge of the abstraction map $T$ is also assumed to be known.

**Q4 Main Weakness:**

Some of the underlying assumptions in the theory section could be explicated more clearly, see the detailed comments below. I am also skeptical of the setup of absLinGaM as a whole. The method assumes that the abstract variables as such are already known: it is fed samples of the abstract variables and only the abstraction function needs to be learned. I have a hard time imagining a situation in which one knows that one set of variables is an abstraction of another without knowing the abstraction function itself. Because the abstract variables are known from the get-go, the title of the article is also a misnomer and should be adapted.

**Q5 Detailed Comments To The Authors:**

•	Can the DAG assumption on the low-level model be weakened to include cycles or confounders?

•	Typo: page 2, right column: an hard intervention -> a hard intervention

•	Page 2, right column: adjacency matrix $W$ -> coefficient matrix would be the better terminology here since adjacency matrices have binary entries

•	Page 2, right column: ‘we assume faithfulness and causal sufficiency’. Since you are working on the level of SCMs, it would be beneficial to make clear that this boils down to the independence of noise terms. In particular, it would be good to indicate that you intend to make this assumption for both the low-level and the abstract SCM, since this is the primary reason that you obtain the block decomposition later on.

•	Page 2, right column: Half a sentence, on why $I-W$ is invertible could be added for completion.

•	Page 3: left column: …, or observational when on non-intervened models -> strange grammar.

•	Page 3: Equation (3): Does the set I include the ‘empty intervention’ in the definition of $\tau$-abstraction?

•	Page 3: left column: ‘we assume $J$ to be the set of all possible interventions’. Do you mean by this that (1) $J$ is equal to all interventions on the space $D(Y)$ or (2) that you a priori fix a subset of all hard interventions that you deem possible?

•	Page 3: left column: … consider the set of concrete intervention to be the pre-image $I = ….$ This sentence is a bit confusing since it looks like a definition of $I$ while the map $\omega$ already requires $I$ to be defined. In my view, you should clarify for which interventions you require consistency. It makes a difference whether $I$ consists of all (hard and soft) intervention on $L$, all hard interventions on $L$ or whether $I$ is the preimage of all hard interventions on $H$. After all, a hard intervention on $L$ need not be mapped to a hard intervention on $H$.

•	Lemma 1: The proof of this result crucially relies on the fact that the set $J$ is separating, in the sense that, for every pair of high-level nodes $(Y,Y’)$, $J$ contains a hard intervention on $Y$ that does not affect $Y’$. This is a strong assumption that your paper relies on throughout, so it would be good if this would be made more explicit. This could even make the results a bit more general, since for some results, consistency with a set of hard intervention $J$ with this separation property (instead of all hard interventions) may suffice. On the other hand, learning abstractions that allow for separating interventions, is perhaps at the core of the problem.

•	Corollary 1: The proof in the appendix could have more details, since there are a couple of edge cases, e.g. the relevant variable $X_3$ could still be part of the same macro-variable $Y_1$, that should be explicated.

•	I like the two examples on page 4, very illustrative. Example 2 might even warrant a more elaborate discussion in the supplement, since it sheds some light on what a ‘good’ abstraction should entail.

•	Example 5: Isn’t the fact that the third model is not abstracted by $Y_1 \rightarrow Y_2$ just another instance of what you already pointed out in Example 2?

•	Section 4.1: Can you elaborate why one would have more concrete than abstract samples? I understand why the number of abstract variables should be significantly lower that the number of concrete variables, but why should that be the case for the samples? Are the samples of $D_J$ independent of those in $D_L$?

•	Section 4.2/Algorithm 2: So, from the start, your algorithm already knows what the abstract and concrete variables are but not how they relate? Could you give a real-life example when one would end up in this situation? It seems like usually one would either (1) know only the concrete variables and search for appropriate abstract variables, or (2) know both but also how they relate to each other.

•	Did you check the performance of your method for causal discovery if you do not need to learn $T$ but know it beforehand? This seems more realistic to me and should bolster the performance, no?

•	In your comparison, did you add the samples of $D_J$ that pertain to the concrete variables to $D_L$ when using the baseline method? That would seem a fairer comparison to me, since otherwise your method had a larger sample size ($D_J$  and $D_L$) to work with.

•	What are the false positives/true positives underlying your ROC curve? Falsely/correctly inferred edges?

**Q9 Complying With Reviewing Instructions:**

Yes

---

> ### Author Rebuttal · Authors · 2024-04-04
>
> We thank the reviewer for the interesting insights and extensive feedback.
>
> > Did you check the performance of your method for causal discovery if you do not need to learn but know it beforehand?
>
> Yes, these results are in the Appendix, where all lines labeled as `Abs-LiNGAM-GT` use the ground truth abstraction function and only recover the concrete model. The results show how the time necessary for fitting the abstraction function does not influence the overall performance.
>
> > Your algorithm already knows what the abstract and concrete variables are but not how they relate?
>
> Indeed, our method knows the abstract and causal variables, but not their causal graph nor the abstraction relation between the two models, which is what we learn.
>
> > Could you give a real-life example when one would end up in this situation?
>
> In interpretable ML, causal abstraction is used to model whether a large language model is implementing an abstract causal model (Wu et al. 2024, Geiger et al. 2023b). Our findings would enable identifying such abstract models without any prior knowledge, but through external annotations.
>
> > Can you elaborate why one would have more concrete than abstract samples?
>
> A common assumption is that low-level data are largely available, such as sensor data, and high-level information is derived by a possibly unknown and expensive annotation process. In this context, our setting resembles few-shot learning, where labeled data is in a substantially smaller availability than unlabeled low-level data.
>
> > Are the samples of $D_J$  independent of those in $D_L$?
> > Did you add the samples of $D_J$ that pertain to the concrete variables to $D_L$  when using the baseline method?
>
> Yes to both. All samples are independent and identically distributed (Equations 17-19). The concrete samples from the paired dataset were also used for our baseline DirectLiNGAM.
>
> > What are the false positives/true positives underlying your ROC curve? Falsely/correctly inferred edges?
>
> Thanks for your suggestion, we will add these metrics to our revised version. In the additional table at [this link](https://drive.google.com/file/d/1bJLaQD5Ipu0qXnQHB-4vUxZzeV7CTtIA/view?usp=sharing), we report precision and recall for all our settings, which we already collected but did not include in the manuscript. In line with DirectLiNGAM, most errors are due to a lack of recall.
>
> > Can the DAG assumption on the low-level model be weakened to include cycles or confounders?
>
> An extension to cyclic models and latent confounders is not be trivial, since we currently exploit acyclicity and causal sufficiency in our theoretical results, as well as for LINGAM.
>
> > Since you are working on the level of SCMs, it would be beneficial to make clear that this boils down to the independence of noise terms.
>
> We thank the reviewer for the suggestion. We will clarify in the background that independence of exogenous variables is a necessary condition for causal sufficiency, which we require for both the concrete and abstract models.
>
> >  Does the set I include the ‘empty intervention’?
>
> Yes, the set contains the empty intervention, which maps to the empty intervention on the abstract model.
>
> > ‘we assume J to be the set of all possible interventions’. Do you mean by this that (1) J is equal to all interventions on the space D(Y)
>
> Yes, we mean the first option, i.e., the abstract model should be consistent with any individual or joint intervention for any possible value in their domain.
>
> > Lemma 1: The proof of this result crucially relies on the fact that the set J is separating
>
> While having separating sets might suffice to some of our results, consistency with respect to the set of all possible interventions is required for *strong* causal abstraction, as discussed in Section 2. We will investigate the relationship between abstraction strength and separating interventions in future work.
>
> > The proof in the appendix could have more details, since there are a couple of edge cases
>
> Thanks for the comment, we will clarify this in the paper. Due to the acyclicity of the abstract graph, the scenario $Y_3=Y_1$ or $Y_3=Y_2$ can arise only at the beginning (resp. the end) of the path. In this case, we could take the successive variable until we get one from a different abstract variable, if any. If there is none, then there exists a T-direct path between the relevant variables of $Y_1, Y_2$ and we fallback to the scenario of Lemma 2, implying that $Y_1 \to Y_2$, which entails that $Y_1$ is an ancestor of $Y_2$.
>
> > Isn’t the fact that the third model is not abstracted by just another instance of what you already pointed out in Example 2?
>
> It is indeed the same issue: in Example 2 we show how abstraction breaks due to the graphical condition. We will improve the discussion and add an edge $X_1 \to X_4$ with non-unitary weight to Example 5, so that the graphical conditions are satisfied while the coefficients are not.

---

### Official Review · Reviewer_hfaA · 2024-03-22

**Q2-1 Originality-Novelty:** 2
**Q2-2 Correctness-Technical Quality:** 3
**Q2-5 Clarity Of Writing:** 3

**Q10 Ethical Concerns:**

None.

**Q1 Summary And Contributions:**

The paper considers the problem of learning the linear SCM when the full model can be partitioned into a concrete, lower-level model $\mathcal{H}$ and an abstract, higher-level model $\mathcal{L}$. $\mathcal{H}$ serves as an abstraction of $\mathcal{L}$, where each variable in $\mathcal{H}$ is a linear combination of some variables $\mathcal{L}$, and the coefficients between $\mathcal{L}$ and $\mathcal{H}$ can be translated through linear mappings. The authors provide structural properties of both models that can be used for recovery. Further, the author proposed a casual discovery algorithm assuming the noises to be non-Gaussian. Simulation results show that the proposed algorithm can speed up the recovery.

**Q2-3 Extent To Which Claims Are Supported By Evidence:**

2: Fair: the main claims are somewhat supported by evidence (but the experimental evaluation may be weak, or does not match entirely with the claims, important baselines may be missing, proofs contain important ideas but lack rigor, algorithmic details are only discussed superficially, references are imprecise, assumptions are not sufficiently motivated or explicated, etc.).

**Q2-4 Reproducibility:**

3: Good: key resources (e.g. proofs, code, data) are available and key details (e.g. proofs, experimental setup) are sufficiently well-described for competent researchers to confidently reproduce the main results.

**Q3 Main Strengths:**

1. Clear presentation of the concepts with figures, examples, and text descriptions in Section 3. It would be better if the authors could add one more example explaining the concepts in Section 2 and Equation (3).
2. Clear statement of all theoretical results, and the associated assumptions.

**Q4 Main Weakness:**

The main weakness of the work is that the considered model and the corresponding algorithm may be over-simplified.
1. We may not be able to observe data from $\mathcal{H} $ and $ \mathcal{L}$ at the same time in real-life scenarios. In most cases, we may either want to learn the causal abstractions from data (i.e., data from $\mathcal{L}$ are not observed), or infer the network structure through limited sensors (i.e., data from $\mathcal{H}$ are not observed).
2. The model assumes that the coefficients between the two models are preserved through the mapping, which seems to be a very strong assumption. As described in Theorem 1, this can only happen if there are sufficiently enough connections among variables in the concrete model.
3. The proposed algorithm assumes that the abstract model $\mathcal{H}$ is causally sufficient, which is also enforced in the data-generating process in Section 5. However, as described in Example 4, if we start from the lower-level model (which is the case in most real applications) then this assumption may be violated. In this case, the proposed algorithm would fail to recover the model as DirectLiNGAM requires causal sufficiency.

**Q5 Detailed Comments To The Authors:**

**Questions**:
1. Does the proposed algorithm require faithfulness assumption? The theoretical results in section 3 require faithfulness, but since DirectLiNGAM does not assume faithfulness, the full structure may still be recoverable even if faithfulness is violated (such as the model in Example 1).
2. In the pruning of $\hat{\mathbf{T}}$, how is the threshold selected? is it assumed that each concrete variable only belongs to a single block (i.e., $\hat{\mathbf{T}}$ is block diagonal)?

**Q9 Complying With Reviewing Instructions:**

Yes

---

> ### Author Rebuttal · Authors · 2024-04-04
>
> We thank the reviewer for their thorough analysis and evaluation of our paper.
>
> > We may not be able to observe data from H and L at the same time in real-life scenarios. In most cases, we may either want to learn the causal abstractions from data (i.e., data from
> L are not observed), or infer the network structure through limited sensors (i.e., data from
> H are not observed).
>
> While we agree our assumptions could potentially not hold in some practical examples, we wanted to point out that (i) having data from $\mathcal{H}$ and $\mathcal{L}$ available at the same time is a foundational requirement to recover causal abstractions, (ii) our method requires a small amount of paired samples in the order of the number of the concrete nodes, and (iii) our setting resembles few shot learning, where the high level samples are the labels for the low level samples, which is a common setting in ML.
>
> In particular, our empirical analysis of Abs-LiNGAM highlights how a considerably smaller number of samples from an abstract model substantially improves learning a large concrete model, where we can additionally use a large amount of unpaired samples to improve the accuracy. As we show in Figures 2a, 4, 5, and 6, we extensively study the performance of Abs-LiNGAM for an increasing number of such samples, showing that reliable abstract information can be produced when the number of paired samples is in the order of the number of concrete nodes.
>
> > The model assumes that the coefficients between the two models are preserved through the mapping, which seems to be a very strong assumption. As described in Theorem 1, this can only happen if there are sufficiently enough connections among variables in the concrete model.
>
> As a first step towards a more general theory of graphical and structural conditions to learn causally abstracted models, we started with the notion of exact abstraction. However, we are currently investigating how to relax this requirement and how to connect our theoretical findings to approximate abstraction and how this reflects on the necessary connections between relevant variables.
>
> > The proposed algorithm assumes that the abstract model H  is causally sufficient, which is also enforced in the data-generating process in Section 5. However, as described in Example 4, if we start from the lower-level model (which is the case in most real applications) then this assumption may be violated. In this case, the proposed algorithm would fail to recover the model as DirectLiNGAM requires causal sufficiency.
>
> Indeed, the sufficiency requirement on the abstract model comes from the use of DirectLiNGAM in our overall pipeline. On the other hand, the integration of our findings with LiNGAM constitutes only one of the contributions of our paper and showcases one possible application of having rigorously defined novel theoretical properties on causal abstraction.
>
> > Does the proposed algorithm require faithfulness assumption? The theoretical results in section 3 require faithfulness, but since DirectLiNGAM does not assume faithfulness, the full structure may still be recoverable even if faithfulness is violated (such as the model in Example 1).
>
> Yes, as we have reported in Section 2 we assume causal faithfulness to obtain reliable abstract information. In Example 1, we highlight why our results require this assumption by showing how without faithfulness there is no abstract edge, despite the presence of T-direct paths between relevant variables. If we were to run Abs-LiNGAM on the models of Example 1, we would get the correct abstract graph in the limit of infinite samples. However, since the abstract model does not contain an edge from $Y_1$ to $Y_2$, Abs-LiNGAM would wrongly infer that the variable $X_1$ is not an ancestor of $X_4$, thus inhibiting the correct recovery of the concrete model.
>
> > In the pruning of T, how is the threshold selected? is it assumed that each concrete variable only belongs to a single block (i.e., T is block diagonal)?
>
> Since we do not know beforehand the ordering of the variables, we can not exploit the block-diagonality of $\hat{\mathbf{T}}$ when optimizing the abstraction function. Thresholding the matrix reliably recovers the abstraction function for a sufficient number of joint samples. We treat the threshold as a hyperparameter of our proposal, and we report in Figure 13 performance over the abstract-induced knowledge $\mathbf{K}$ for an increasing number of samples against different threshold values.

---

### Official Review · Reviewer_UVWy · 2024-03-25

**Q2-1 Originality-Novelty:** 4
**Q2-2 Correctness-Technical Quality:** 4
**Q2-5 Clarity Of Writing:** 4

**Q1 Summary And Contributions:**

Roughly, one structural causal model is an *abstraction* of another if the variables in the former  model are "higher-level" or "macro"  (e.g., beliefs) are functions of the variables in the latter "lower-level" or "micro" model (e.g., neural activity).    The paper first characterize the conditions under which a linear SCM is a *linear* abstraction of another linear SCM and how features of the two models are related (e.g., how edges in the micro-model determine edges in the macro one).  In the final section of the paper, these theoretical results are used to propose an algorithm for learning the concrete/micro model from the abstract one, which is likewise inferred from data.  Using simulated data, the authors' algorithm is compared with the performance of Direct-Lingam on the concrete model.  Though Direct-Lingam is typically superior, the authors try to contextualize these results.

**Q2-3 Extent To Which Claims Are Supported By Evidence:**

4: Excellent: all claims are supported by very convincing evidence (in the form of comprehensive experimental evaluation, rigorous mathematical proofs, detailed (pseudo-)code, precise references, well-motivated and realistic assumptions) and the authors deliver what they promise.

**Q2-4 Reproducibility:**

4: Excellent: key resources (e.g. proofs, code, data) are available and key details (e.g. proof sketches, experimental setup) are comprehensively described for competent researchers to confidently and easily reproduce the main results.

**Q3 Main Strengths:**

See ratings above.   From what I can tell, the paper is technically sound; all the mathematical theorems and proofs appear to be correct.  The paper is well-written, and the authors do a nice job explaining how their work is related to (and builds upon) existing work.

**Q4 Main Weakness:**

The paper doesn't mention any real empirical applications for this work, and to be honest, I'm skeptical that many exist.  It seems extremely limiting to assume both that the lower-level model is linear *and* that the abstraction function is linear.  I conjecture, therefore, that the  results in the paper are entirely of theoretical interest.  It'd be helpful if there were some real-world example in which both linearity assumptions are plausible.

**Q5 Detailed Comments To The Authors:**

See discussion of weaknesses.

**Q9 Complying With Reviewing Instructions:**

Yes

---

> ### Author Rebuttal · Authors · 2024-04-04
>
> We thank the reviewer for their evaluation of the paper.
>
> > The paper doesn't mention any real empirical applications for this work, and to be honest, I'm skeptical that many exist. It seems extremely limiting to assume both that the lower-level model is linear and that the abstraction function is linear. I conjecture, therefore, that the results in the paper are entirely of theoretical interest. It'd be helpful if there were some real-world examples in which both linearity assumptions are plausible.
>
> In this work, we addressed the linear scenario as a first step towards defining both the graphical and structural conditions for causal abstraction, which paves the way for more general theories. Additionally, while from the theoretical point of view linearity is a strong assumption, in practice linear models are a core instrument of econometrics and other social sciences to approximate complex and realistically non-linear behaviors. In these cases, being able to find an accurate linear abstraction implies that the behavior of the system can be approximated with a linear model at different levels of detail.
>
> As a potential application of our method we consider the interpretation of LLMs. For example, most of the operators between variables in the hidden space of a Transformer are linear, as are the abstraction functions themselves. Previous work already adopted the abstraction framework for this task while assuming that the ideal abstract model is known (Wu et al. 2024, Geiger et al. 2024). In this context, our method is a first step towards learning one of the many viable abstractions.
>
> Wu, Zhengxuan, et al. "Interpretability at scale: Identifying causal mechanisms in alpaca." Advances in Neural Information Processing Systems 36 (2024).
>
> Geiger, Atticus, et al. "Finding alignments between interpretable causal variables and distributed neural representations." Causal Learning and Reasoning. PMLR, 2024.

---

### Meta-Review · Area_Chair_J7wM · 2024-04-16

All 5 reviewers recommended acceptance, even though some argued that, given the assumptions made by the authors, the results were not too surprising.